# Vid2World: Crafting Video Diffusion Models to Interactive World Models

**Siqiao Huang**[1]*, **Jialong Wu**[1]*, **Qixing Zhou**[2], **Shangchen Miao**[1], **Mingsheng Long**[1]✉

[1]Tsinghua University    [2]Chongqing University

huang-sq23@mails.tsinghua.edu.cn, wujialong0229@gmail.com
mingsheng@tsinghua.edu.cn

## Abstract

World models, which predict future transitions from past observation and action sequences, have shown great promise for improving data efficiency in sequential decision-making. However, existing world models often require extensive domain-specific training and still produce low-fidelity, coarse predictions, limiting their usefulness in complex environments. In contrast, video diffusion models trained on large-scale internet data have demonstrated impressive capabilities in generating high-quality videos that capture diverse real-world dynamics. In this work, we present *Vid2World*, a general approach for leveraging and transferring pre-trained video diffusion models into interactive world models. To bridge the gap, Vid2World systematically explores *video diffusion causalization*, reshaping both the architecture and training objective of pre-trained models to enable autoregressive generation. Additionally, it incorporates a *causal action guidance* mechanism to enhance action controllability in the resulting interactive world models. Extensive experiments across multiple domains, including robot manipulation, 3D game simulation, and open-world navigation, demonstrate that our method offers a scalable and effective pathway for repurposing highly capable video diffusion models into interactive world models. Code and models are available at https://knightnemo.github.io/vid2world/.

## 1 Introduction

World models (Ha & Schmidhuber, 2018; Dawid & LeCun, 2023) have emerged as pivotal components for sequential decision-making, enabling agents to predict future states and plan actions by simulating environment dynamics. Despite their success in numerous domains, including game simulation (Hafner et al., 2020; Alonso et al., 2024), autonomous driving (Wang et al., 2024b), and robotics (Yang et al., 2024), these models conventionally rely solely on **in-domain action-labeled data**, necessitating meticulous and labor-intensive data collection, yet still often yielding relatively coarse predictions with limited physical realism, constraining their applicability in complex environments.

To mitigate this data-hungry nature, recent works (Bar et al., 2025; Wu et al., 2024) have drawn inspiration from the success of foundation models (Bommasani et al., 2021), exploring pre-training on broader, **cross-domain action-labeled data**. While this strategy improves data efficiency and generation quality to some extent, it does not solve the fundamental problem. The high cost of acquiring any form of action-labeled data persists, and the resulting models still struggle to generate visuals with high fidelity and realism. This indicates that merely expanding the scope of action-labeled data is insufficient. A more fundamental paradigm shift is thus imperative to truly unlock the full capabilities of world models.

We argue that the requisite paradigm shift is to leverage the largest yet most overlooked data source: **internet-scale action-free video data**. Abundant, easy to collect, diverse with rich world priors, these data constitute the most prominent part of the *data pyramid for world models* (shown in Figure 1). While prior work (Gao et al., 2025) has explored co-training with such data, we highlight a more

---
*Equal contribution.

Figure 1: **Vid2World repurposes video diffusion models for interactive world modeling**. From the perspective of the *data pyramid for world models*, it leverages vast pre-trained knowledge from internet-scale, action-free video data to achieve high-fidelity, action-conditioned generation across diverse downstream domains with limited interaction data.

direct and cost-efficient path: transferring the physical priors and generative capabilities learned by video diffusion models (Ho et al., 2020; OpenAI, 2024; DeepMind, 2024) into interactive world models. This transition from data-level exploitation to model-level transfer not only avoids the prohibitive cost of training on massive video corpora but also extracts physical priors more smoothly, as non-causal generative modeling could be inherently easier than its causal counterpart.

Despite profound potential, two significant challenges arise in bridging the gap between passive video diffusion models and interactive world models, as shown in Figure 2. The first key challenge lies in enabling *causal generation*. Standard video diffusion models, designed for full-sequence denoising with bidirectional context, inherently introduce non-causal temporal dependencies. This makes them unsuitable for causal rollouts, where future predictions must strictly depend on past information. The second challenge, equally critical, is enforcing *action conditioning*. While causalization enables autoregressive rollout, these models still lack the ability for counterfactual reasoning—predicting how different actions influence future states. This necessitates injecting fine-grained, frame-level action signals into the generation process. Especially in diffusion models, despite that classifier-free guidance (Ho & Salimans, 2021) offers the freedom of balancing sample diversity and fidelity, extending it to action guidance still requires careful algorithmic and architectural designs.

In this paper, we present *Vid2World*, a general approach to effectively transform internet-scale pre-trained video diffusion models into interactive world models capable of autoregressive, action-conditioned generation. To causalize video diffusion models, we systematically explore and discover better weight transfer schemes that adapt temporal attention and convolution layers into their causal counterparts, enabling fine-tuning under a causal training objective (Chen et al., 2024). For action conditioning, we inject action signals into model inputs at corresponding frames and design an extended training objective that supports action guidance during diffusion sampling at each frame in principle. We evaluate Vid2World by transferring an extensively pre-trained, 1.4B-parameter video diffusion model (Xing et al., 2024) to diverse domains, including robot manipulation (Brohan et al., 2023), 3D game simulation (Pearce & Zhu, 2022), and open-world navigation (Shah et al., 2022). Experimental results demonstrate significant improvements over existing transfer approaches as well as state-of-the-art world models.

To summarize, our contributions are: (1) To the best of our knowledge, we are the *first* to systematically explore the problem of transferring full-sequence, non-causal, passive video diffusion models into autoregressive, interactive, action-conditioned world models. (2) We propose Vid2World, a general and effective approach for this problem, featuring novel techniques for the causalization and action conditioning of video diffusion models. (3) State-of-the-art performance of Vid2World across domains establishes new benchmarks for this critical problem and facilitates future research.

## 2 RELATED WORKS

**Diffusion for World Modeling.** Due to the high fidelity offered by diffusion models in image and video generation, utilizing diffusion for world modeling has garnered growing interest. Prior works fall primarily into two categories. The first treats world modeling as a conditional image generation problem, where history observation and action sequences serve entirely as conditions. While these approaches follow an autoregressive framework and have shown promise in domains such as game simulation (Alonso et al., 2024; Decart et al., 2024) and navigation (Bar et al., 2025), they typically rely on a fixed-length context window, limiting their applicability in environments that demand

Figure 2: **Transforming video diffusion models into interactive world models involves two key challenges**: (1) Causal generation: converting full-sequence diffusion models into causal diffusion models; (2) Action conditioning: adapting causal diffusion models into interactive world models.

long-term temporal reasoning. The second category formulates the problem as a full-sequence video generation task (Yang et al., 2024; Yu et al., 2025; Zhou et al., 2024), often achieving better temporal coherence between frames. Yet, these models operate on full video segments, precluding autoregressive rollout, and thus hindering their use in interactive environments.

**Leveraging Foundation Models to World Models.** Foundation models (Bommasani et al., 2021), trained on large-scale and diverse data, have shown revolutionary potential across modalities such as text (OpenAI et al., 2023; Guo et al., 2025), image (Rombach et al., 2022; Betker et al., 2023), and video (OpenAI, 2024; DeepMind, 2024). In the text domain, large language models are prompted to act as world models for spatio-temporal reasoning in agentic tasks (Hao et al., 2023; Gkountouras et al., 2025; Hu et al., 2025). In the video domain, adapting pre-trained generative models into world models typically involves architectural modifications. For instance, He et al. (2025) integrate an action-conditioning module into the generative backbone, while Rigter et al. (2024) introduce an action-aware adapter to modulate the output of a frozen video model. However, these approaches often overlook the critical need for *interactivity* and *temporal causality*, limiting their applicability in sequential decision-making and interactive environments.

## 3 PRELIMINARIES

**World Models.** A world model is an internal model learned by an agent to model the dynamics of its environment. This environment is typically formalized as a (discrete-time) Partially Observable Markov Decision Process (POMDP) (Kaelbling et al., 1998), defined over a tuple $(\mathcal{S}, \mathcal{O}, \phi, \mathcal{A}, p, r, \gamma)$. At each time step $t$, the agent receives an observation $o_t = \phi(s_t)$, where $s_t \in \mathcal{S}$ is the underlying state that satisfies the Markov property. Upon taking an action $a_t \in \mathcal{A}$, the next state is sampled from the transition distribution $p : \mathcal{S} \times \mathcal{A} \to \Delta(\mathcal{S})$, i.e. $s_{t+1} \sim p(\cdot \mid s_t, a_t)$. In the context of world models, the agent learns to estimate this transition function through history observation and action sequence: $p_\theta(o_{t+1}|o_{\leq t}, a_{\leq t})$. While world models can be applied to a wide range of observation modalities, including proprioceptive signals (Yin et al., 2025), text (Wang et al., 2024a; Wu et al., 2025), 3D meshes (Zhang et al., 2024), and pixel-based inputs (Wu et al., 2024; Zhu et al., 2025), here we focus on learning in the pixel space, where observations are defined over $\mathcal{O} = \mathbb{R}^{H \times W \times 3}$.

**Diffusion Models.** Diffusion models (Ho et al., 2020; Song et al., 2021a) are highly expressive generative models that learn to approximate a target data distribution $q(\mathbf{x})$, where $\mathbf{x} \in \mathbb{R}^d$, by progressively denoising a Gaussian noise. At its core, the model makes use of two Markov Chains: a forward process and a backward process, to transport between the noise distribution $\mathbf{x}^K \sim \mathcal{N}(0, \mathbf{I})$ and the distribution of interest $\mathbf{x}^0 \sim q(\mathbf{x})$. The forward (noising) process is defined as:

$$q(\mathbf{x}^k|\mathbf{x}^{k-1}) = \mathcal{N}(\mathbf{x}^k; \sqrt{1 - \beta_k}\, \mathbf{x}^{k-1}, \beta_k\, \mathbf{I}),$$

where $\{\beta_k\}_{k=0}^K$ is a pre-defined noise schedule. Starting from pure noise $\mathbf{x}^K \sim \mathcal{N}(0, \mathbf{I})$, the learned reverse (denoising) process aims to recreate $\mathbf{x}^0 \sim q(\mathbf{x})$ using the following factorization:

$$p_\theta(\mathbf{x}^{k-1}|\mathbf{x}^k) = \mathcal{N}(\mathbf{x}^{k-1}; \boldsymbol{\mu}_\theta(\mathbf{x}^k, k), \gamma_k\, \mathbf{I}).$$

In practice, it is common to reparameterize the objective in terms of noise prediction, i.e., learning to predict $\boldsymbol{\epsilon}^k = (\sqrt{1 - \overline{\alpha}_k})^{-1}\mathbf{x}^k - \sqrt{\overline{\alpha}_k}\, \boldsymbol{\mu}$, where $\alpha_k \triangleq 1 - \beta_k$ and $\overline{\alpha}_k \triangleq \prod_{i=1}^k \alpha_i$. This simplifies to

minimizing the mean square loss:

$$\mathcal{L}(\theta) = \mathbb{E}_{k,\boldsymbol{\epsilon},\mathbf{x}^0}[||\boldsymbol{\epsilon} - \boldsymbol{\epsilon}_\theta(\mathbf{x}^k, k)||^2],$$

where $\mathbf{x}_k = \sqrt{\overline{\alpha}_k}\,\mathbf{x}^0 + \sqrt{1 - \overline{\alpha}_k}\,\boldsymbol{\epsilon}$ and $\boldsymbol{\epsilon} \sim \mathcal{N}(0, \mathbf{I})$. Sampling is performed via iterative denoising through Langevin dynamics: $\mathbf{x}^{k-1} \leftarrow \frac{1}{\sqrt{\alpha_k}}(\mathbf{x}^k - \frac{1-\alpha_k}{\sqrt{1-\overline{\alpha}_k}}\boldsymbol{\epsilon}_\theta(\mathbf{x}^k, k) + \sigma_k\mathbf{w})$.

**Video Diffusion Models.** In video diffusion models (Ho et al., 2022), the sample $\mathbf{x}$ is represented as a sequence of frames $(\mathbf{x}_t^{k_t})_{1 \le t \le T}$, where $t$ denotes the frame index and $k_t$ indicates the noise level at that frame. Conventional approaches (Blattmann et al., 2023) apply one uniformly sampled noise level across all frames, treating each frame identically within the denoising process. To relax this constraint, Chen et al. (2024) propose to sample noise levels independently for each frame, i.e., $k_t \sim \mathcal{U}([0, K])$ during training. Intuitively, this formulation captures a more diverse set of noise level combinations across frames during training, opening up new capabilities. At inference time, the model follows a denoising schedule $\mathcal{K} \in \mathbb{R}^{M \times T}$, where $M$ is the number of denoising steps and each row $\mathcal{K}_m \in \mathbb{R}^T$ specifies the per-frame noise levels at step $M$. By setting $k_t = 0$ for history frames, $k_t = K$ for masked future frames, and progressively denoising the current frame $k_\tau \in \{K, ..., 0\}$, the model is capable of autoregressive generation.

# 4 METHODS

While video diffusion models excel at generating high-fidelity, physically plausible sequences, their default formulation is fundamentally incompatible with interactive world modeling. Concretely, two key transformation barriers stand out:

1. **Inability of causal generation**: Typical video diffusion models generate frames using *bidirectional temporal context*, allowing future frames to influence the past;
2. **Lack of action conditioning**: These models are typically conditioned on coarse, video-level inputs (e.g., text prompts) and lack mechanisms for fine-grained, frame-level action conditioning.

To overcome these transfer barriers, we propose *Vid2World* with two key modifications, contributing to a systematic, multi-level framework for converting video diffusion models into interactive world models, as shown in Figure 2. In Section 4.1, we present the strategy of *video diffusion causalization*, which converts non-causal architectures into temporally causal variants compatible with the post-training objective (Chen et al., 2023), by exploring weight transfer mechanisms to maximally preserve the representations learned during pre-training. In Section 4.2, we extend the training objective to enable *causal action guidance* during inference for step-wise, interactive rollouts.

## 4.1 VIDEO DIFFUSION CAUSALIZATION

To causalize video diffusion models, modifications are required on both *architectures* and *training objectives*. From an architectural standpoint, while bidirectional temporal modules in standard video diffusion models, which allow information flow across all timesteps, are effective for full-sequence generation, they are fundamentally incompatible with autoregressive world modeling, where the current observation must not depend on future observations or actions. This necessitates architectural surgery to enforce temporal causality, specifically in the computation and parameters of temporal attention (HaCohen et al., 2024) or non-causal convolutions (Blattmann et al., 2023; Guo et al., 2024).

**Temporal Attention Layers.** Non-causal temporal attention layers can be converted into their causal counterparts by straightforwardly applying causal masks. Since attention operates through dot products between queries and keys, it is inherently adaptive to variable sequence lengths; therefore, restricting the receptive field to exclude future frames does not alter the underlying computation of inter-token relationships. Consequently, this does not mandate parametric modifications.

**Temporal Convolution Layers.** In contrast, causalizing temporal convolution layers is more challenging. These layers employ symmetric kernels that aggregate features from both past and future frames, and simple adaptations may lead to suboptimal utilization of pre-trained kernel weights. To achieve this, we systematically investigate three different strategies, as detailed below.

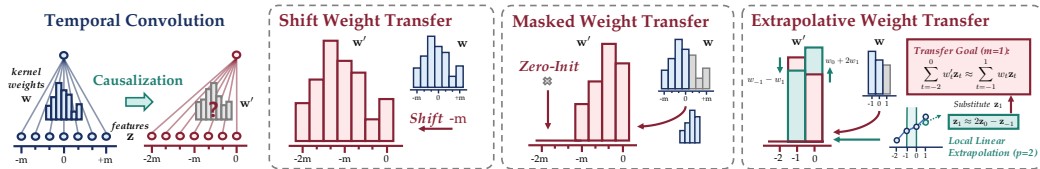

Figure 3: **Illustration of weight transfer mechanisms for temporal convolution layers**: (1) *Shift*: shifts all weights into the past. (2) *Masked*: retains only past weights. (3) *Extrapolative*: leverages local linear feature relationships more in principle(example shown with $m = 1, p = 2$).

A naive approach, which we term **Shift Weight Transfer**, directly reuses the full pre-trained kernel $\{w_t\}_{t=-m}^{m}$ by shifting it $m$ steps into the past, resulting in a new causal kernel $\{w_t'\}_{t=-2m}^{0}$. While this preserves all kernel weights, it introduces *temporal misalignment*: the kernel's $i$-th position now aggregates information at timestep $i - m$, giving no guarantees of producing similar representations.

An alternative strategy, **Masked Weight Transfer**, truncates the kernel by retaining only the weights corresponding to past and present timesteps $\{w_t\}_{t=-m}^{0}$ while setting the rest to zero $\{w_t'\}_{t=-2m}^{-m-1} \equiv 0$. This resembles applying a hard causal mask to the kernel at initialization. Although causality is enforced, it discards potentially useful information encoded in the future-facing weights.

Finally, we propose a more principled and robust mechanism, **Extrapolative Weight Transfer**, based on local linear extrapolation of features along the temporal dimension. Formally, we posit that the feature at a future timestep $\mathbf{z}_{t+k}$ can be linearly approximated over a window of $p$ past timesteps:

$$\mathbf{z}_{t+k} \approx \sum\nolimits_{j=0}^{p-1} \gamma_{k,j} \, \mathbf{z}_{t-j} + \boldsymbol{\beta}_k,$$

where $\boldsymbol{\gamma}_{k,\cdot}, \boldsymbol{\beta}_k$ are determined by a linear extrapolation from the past $p$ features. Our core principle is to maximally preserve the output representation of the original convolution, such that the new causal computation produces a similar result to the original non-causal one:

$$\sum\nolimits_{i=-m}^{m} w_i \mathbf{z}_{t+i} + \mathbf{b} = \sum\nolimits_{j=-2m}^{0} w_j' \mathbf{z}_{t+j} + \mathbf{b}'.$$

This is achieved by re-distributing the weights $\{w_i\}_{i>0}$ that originally acted on future frames, back onto the past part of the kernel, according to the linear feature relationships:

$$w_j' = \mathbf{1}_{[j \geq -m]} \cdot w_j + \mathbf{1}_{[-p+1 \leq j \leq 0]} \cdot \sum\nolimits_{i=1}^{m} \gamma_{i,-j} w_i, \quad \mathbf{b}' = \mathbf{b} + \sum\nolimits_{i=1}^{m} w_i \boldsymbol{\beta}_i.$$

These architectural adaptation strategies are illustrated in Figure 3, with a didactic example of Extrapolative Weight Transfer for $m = 1, p = 2$. A detailed mathematical derivation and analysis of the error bounds are provided in Appendix A.

**Training Objectives for Causal Generation.** Architectural changes alone are insufficient to enable causal generation. In this setting, future frames are predicted step by step, conditioned on previously fully denoised frames, i.e., under noise levels $(k_t)_{t=1}^{T} = (0, 0, \ldots, 0, k)$, $k \in \{0, \ldots, K\}$. Hence, the model must be trained to handle these inference-time noise-level distributions. In conventional video diffusion models, the training procedure follows a *homogeneous noise schedule*, where all frames share the same noise level. This limited subset of noise-level combinations makes them naturally incompetent for noise levels at autoregressive inference. Therefore, it becomes vital to train the model with different noise levels across frames. Here, we adopt *Diffusion Forcing* (Chen et al., 2024), where we sample noise levels to be independent and uniform in different frames, i.e., $k_t \sim \mathcal{U}(0, K), \forall t$. This training scheme exposes the model to the full space of noise-level combinations in the history frames, thereby enabling flexible and robust causal rollouts.

## 4.2 CAUSAL ACTION GUIDANCE [1]

Causal video diffusion models alone are not yet interactive world models, as they still fall short of action-conditioned generation. Prior works (Alonso et al., 2024; Bar et al., 2025) primarily explore

---

[1]While throughout the paper, we primarily use "causal" to denote *temporal causality*, in this section, our mechanism implicitly involves *interventional causality* via action guidance. See Appendix A.5 for details.

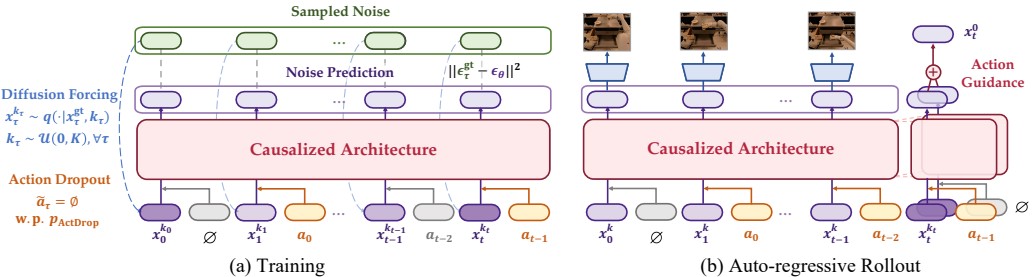

Figure 4: **Training and sampling of Vid2World**, initialized by architecture causalization. (a) During training, we add independently sampled noise levels to each frame, as well as randomly drop out each action with a fixed probability. (b) For autoregressive rollout, we denoise the latest frame while setting history clean. Action guidance is added for the current action. See Appendix B for details.

integrating action condition at the *video level*, where the entire action sequence is encoded to a single embedding, analogous to text embeddings. However, this approach has two major drawbacks: (a) Lacking the ability to perform fine-grained, frame-level action-conditioned predictions; (b) Incompatibility with interactive settings, where actions arrive sequentially in an online fashion during inference. In this section, we introduce both conceptual and methodological advances that extend action conditioning into *causal action guidance*, enabling principled steering of the generative process toward action-aligned predictions in interactive environments.

**Causal Action Injection.**    To address these limitations, we equip the model with *frame-level* conditions through architectural modifications. When predicting $o_t$, the embedding of $a_{t-1}$ is added to the model's latent representation at temporal position $t$. This allows each frame to be conditioned directly on its preceding action in a temporally aligned manner, opening up the potential for precise, fine-grained control in interactive settings. Specifically, this is implemented by feeding the action inputs into the denoising network using a lightweight multi-layer perceptron (Haykin, 1994).

**Training and Sampling with Guidance.**    For more targeted control over the generated dynamics, we adopt classifier-free guidance (Ho & Salimans, 2021) in our autoregressive action-conditioned setting, realizing *Causal Action Guidance*. Classifier-free guidance trains a model to jointly learn a conditional and an unconditional score function, allowing for amplified guidance at inference time by steering the output toward the conditional distribution. In our setup, the score function $\epsilon_\theta([\mathbf{x}_\tau^{k_\tau}], [\mathbf{a}_\tau], [k_\tau])$ takes in a tuple of noised observations $[\mathbf{x}_\tau^{k_\tau}]$, actions $[\mathbf{a}_\tau]$, noise levels $[k_\tau]$ as input, and the conditioned variable is the most recent action. Therefore, the model should be capable of capturing both the conditional score function: $\boldsymbol{\epsilon}_{\text{cond}} = \boldsymbol{\epsilon}_\theta([\mathbf{x}_\tau^{k_\tau}]_{\tau \leq t}, [\mathbf{a}_\tau]_{\tau < t}, [k_\tau]_{\tau \leq t})$, as well as its unconditional counterpart, where the most recent action is masked: $\boldsymbol{\epsilon}_{\text{ucond}} = \boldsymbol{\epsilon}_\theta([\mathbf{x}_\tau^{k_\tau}]_{\tau \leq t}, [\mathbf{a}_{\tau < t-1}, \varnothing], [k_\tau]_{\tau \leq t})$.

To achieve this, we extend our training objective by incorporating an *action dropout* mechanism, where $\tilde{a}_t$ for each timestep $t$ is independently dropped with a fixed probability $p$:

$$\mathcal{L}(\theta) = \mathbb{E}_{[k_\tau], \boldsymbol{\epsilon}, [\mathbf{x}_\tau^0], [\tilde{\mathbf{a}}_\tau]} \left[ \sum_{t=0}^{T} ||\boldsymbol{\epsilon}_t - \boldsymbol{\epsilon}_\theta([\mathbf{x}_\tau^{k_\tau}]_{\leq t}, [\tilde{\mathbf{a}}_\tau]_{< t}, [k_\tau]_{\leq t})||^2 \right], \quad \tilde{\mathbf{a}}_t = \begin{cases} \varnothing, & \text{w.p. } p, \\ \mathbf{a}_t, & \text{otherwise.} \end{cases}$$

At its core, this mechanism compels the model to learn score functions conditioned on all subsets of the action sequences, including the effect of the immediate action on the predicted transition. This, in turn, enables classifier-free guidance for sampling via: $\boldsymbol{\epsilon}_{\text{guided}} = (1 + \lambda) \cdot \boldsymbol{\epsilon}_{\text{cond}} - \lambda \cdot \boldsymbol{\epsilon}_{\text{ucond}}$, where $\lambda \in \mathbb{R}^+$ is the guidance scale. Intuitively, since the score function represents the gradient of the log-probability, this linear composition in the score space corresponds to a multiplicative steering mechanism in the probability space. We formalize this equivalence in the following theorem.

**Theorem 4.1 (Causal Action Guidance as Probability Steering).** *Let $\mathcal{H}_t := ([\mathbf{x}_\tau]_{\tau < t}, [\mathbf{a}_\tau]_{\tau < t-1})$ denote the history context excluding the current action. Under standard score-based formulation,, the proposed score composition is mathematically equivalent to sampling from the following steered*

*posterior distribution, where $\omega \propto (1 + \lambda)$ is a fixed constant:*

$$\tilde{p}(\mathbf{x}_t \mid \mathbf{a}_{t-1}, \mathcal{H}_t) \propto \underbrace{p(\mathbf{x}_t \mid \mathcal{H}_t)}_{\text{History-Consistent Prior}} \cdot \underbrace{\left(\frac{p(\mathbf{x}_t \mid \mathbf{a}_{t-1}, \mathcal{H}_t)}{p(\mathbf{x}_t \mid \mathcal{H}_t)}\right)^{\omega}}_{\text{Action Alignment}} \propto p(\mathbf{x}_t \mid \mathcal{H}_t) \cdot p(\mathbf{a}_{t-1} \mid \mathbf{x}_t, \mathcal{H}_t)^{\omega}.$$

In other words, the guidance term acts as an implicit classifier, steering the generation towards regions aligned with the user's most recent action, while preserving the high-fidelity generation capabilities in sequential modeling. As a result, through varying $\lambda$, the model is equipped with the test-time flexibility of controlling responsiveness towards fine-grained action variations. Ultimately, this transformation better aligns the model with its core objective of world modeling—not merely to capture average behavioral trends, but to reason about an agent's immediate actions. We provide the detailed proof for Theorem 4.1 in Appendix A.4.

**Summary.** Vid2World transfers full-sequence, passive video diffusion models into autoregressive, interactive world models. Through *video diffusion causalization*, we open up the model's capability to perform causal generation, and through *causal action guidance*, we incorporate and strengthen action signals for interactive settings. Pseudocode of our approach is provided in Algorithms 1 and 2.

## 5 EXPERIMENTS

We leverage *DynamiCrafter* (Xing et al., 2024), a state-of-the-art 1.1B U-Net-based video diffusion model pre-trained on internet-scale videos, as our base model. We evaluate Vid2World across multiple domains, spanning real-robot manipulation, 3D game simulation, and open-world navigation. Results show that the transferred models can not only achieve high-fidelity video predictions but also support downstream tasks in decision-making, showcased by real-to-sim policy evaluation.

### 5.1 VID2WORLD FOR ROBOT MANIPULATION

Robot manipulation is an ideal test for world models, demanding action-conditioned predictions that are both visually realistic and causally faithful under real-world physical constraints.

**Setup.** We utilize the RT-1 dataset (Brohan et al., 2023), a collection of real-world robotic experiences spanning diverse manipulation tasks such as picking, placing, and drawer operation. Our base model under extrapolative weight transfer is post-trained for 100k gradient steps ($\sim$ 7 days on 4$\times$ A100 GPUs), with two inference variants: (1) *Vid2World-NAR*, which follows conventional video diffusion models and baseline methods by denoising all frames simultaneously in a non-autoregressive manner, under homogeneous noise levels; and (2) *Vid2World*, which denoises frames autoregressively with proposed action guidance. Evaluation uses standard video generation metrics, including FVD (Unterthiner et al., 2018), FID (Heusel et al., 2017), SSIM, PSNR, LPIPS (Zhang et al., 2018), and DreamSim (Fu et al., 2023). Implementation details can be found in Appendix C.3.1.

**Baselines.** We compare against a variety of baselines introduced by Rigter et al. (2024) that build upon the same base model but utilize different transfer approaches, including Action-Conditioned Fine-tuning, Language-Conditioned Fine-tuning, ControlNet (Zhang et al., 2023), Classifier Guidance, and AVID (Rigter et al., 2024). Details are shown in Appendix C.3.2.

**Results.** As shown in Table 1, Vid2World demonstrates strong quantitative performance across both non-autoregressive and autoregressive settings, outperforming or matching other transfer methods. In the non-autoregressive setting, it delivers superior or comparable results compared to all prior methods. Even in the autoregressive generation setup, where other baselines are not capable of doing so, Vid2World still attains superior performance in FVD and FID, as well as on par performance to previous best methods in other metrics, showcasing its strong capabilities in world modeling.

**Application: Real2Sim Policy Evaluation.** We further conduct Real2Sim Policy Evaluation to demonstrate our method's potential to aid downstream decision-making. Following SIMPLER (Li et al., 2025), our goal is to estimate the performance of a given policy by interacting with the world

Table 1: **World modeling performance across various domains.** Best performances are in **bold**, second best are underlined. Dash (-) indicates the metric was not originally evaluated for that dataset. [*]Autoregressive prediction. [†]Non-autoregressive prediction. [‡]One-step prediction.

| Model | FVD ↓ | FID[1] ↓ | SSIM ↑ | LPIPS ↓ | PSNR ↑ | DreamSim ↓ |
|---|---|---|---|---|---|---|
| | | 🤖 *Robot Manipulation: RT-1* | | | | |
| Pre-trained Base Model[†] | 237.6 | 5.432 | 0.712 | 0.228 | 20.6 | - |
| Classifier Guidance[†] | 213.1 | 6.005 | 0.683 | 0.250 | 19.8 | - |
| ControlNet[†] | 27.1 | 3.248 | 0.836 | 0.148 | 24.5 | - |
| Action-Conditioned[†] | 24.2 | 2.965 | 0.852 | **0.134** | 25.6 | - |
| Language-Conditioned[†] | 33.7 | 3.511 | 0.812 | 0.177 | 22.1 | - |
| AVID[†] | 39.3 | 3.436 | 0.842 | 0.142 | 25.3 | - |
| Vid2World-NAR[†] | 18.7 | 5.871 | **0.856** | 0.140 | **25.8** | **0.048** |
| Vid2World[*] | **18.5** | 5.806 | 0.842 | 0.152 | 24.6 | 0.054 |
| | | 🎮 *3D Game Simulation: CS:GO* | | | | |
| DIAMOND-Fast[*] | 577.1 | 115.6 | 0.449 | 0.547 | 18.2 | 0.2817 |
| DIAMOND-HQ[*] | 368.5 | 87.2 | 0.447 | 0.510 | 18.3 | 0.2416 |
| Vid2World[*] | **106.6** | **17.5** | **0.481** | **0.404** | **18.7** | **0.135** |
| | | 🧭 *Open-World Navigation: RECON* | | | | |
| NWM (1B)[‡] | **31.2** | **34.1** | 0.389 | **0.295** ± 0.002 | 15.343 ± 0.060 | **0.091** ± 0.001 |
| NWM + Ego4D (1B)[‡] | 41.0 | 34.9 | 0.361 | 0.368 ± 0.003 | 14.072 ± 0.075 | 0.138 ± 0.002 |
| Vid2World[*] | 59.4 | 42.9 | **0.481** | 0.3236 | **16.10** | 0.108 |

model, rather than the real world. This requires the model to perform autoregressive rollouts and faithfully predict the diverse outcomes induced from different policies. The procedure is summarized in Algorithm 3. We evaluate on the task of closing drawers and consider three policy checkpoints from RT-1 (Brohan et al., 2023): *RT-1 (Begin)*, *RT-1 (15%)*, and *RT-1 (Converged)*, representing different stages of training. Human evaluation is used as the verifier to annotate trajectory success. As shown in Figure 5, Vid2World reliably reflects the performance gap among policies, closely tracking their real-world success trends. Further details can be found in Appendix C.4.

## 5.2 VID2WORLD FOR 3D GAME SIMULATION

Game simulation is a key application for world models and *neural game engines* (Bamford & Lucas, 2020; Bruce et al., 2024; Decart et al., 2024) have attracted growing attention. It is particularly challenging due to complex temporal dynamics and strong action dependence, involving rapid viewpoint shifts, contact-rich interactions, and fine-grained motion patterns that demand reasoning over causally entangled visual-temporal cues.

**Setup.** We evaluate *Vid2World* on the celebrated 3D video game *Counter-Strike: Global Offensive* (CS:GO) using the online gameplay dataset from Pearce & Zhu (2022) (5.5M frames, 95 hours), with the exact 0.5M-frame holdout set from DIAMOND (Alonso et al., 2024) for testing. DIA-MOND (Alonso et al., 2024), a state-of-the-art autoregressive world model, generates the next frame conditioned on a fixed number of previous observations and actions. Following its setup with 4 conditioning frames, we initialize with four history frames, and autoregressively generate frames until a sequence length of 16. Evaluation metrics are the same as Section 5.1, computed on predicted frames excluding conditioning frames. More details are listed in Appendix C.5.

**Results.** As shown in Table 1, Vid2World outperforms both configurations of DIAMOND across all evaluation metrics with a significant margin, including a 79.9% relative performance improvement

---

[1]In the publicly released code of AVID (Rigter et al., 2024), the FID scores are computed without setting the Inception model to evaluation mode, making it artificially lower. These results are shown in gray accordingly.

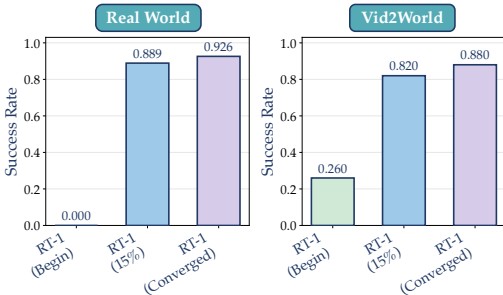
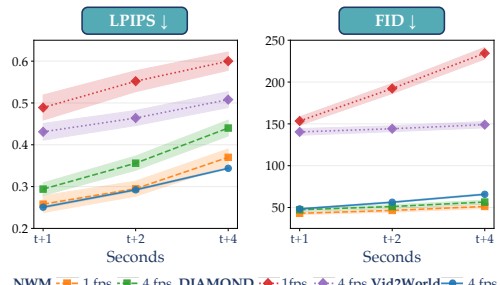

Figure 5: Vid2World for real2sim policy evaluation, validated by real-world evaluation.

Figure 6: Comparison of autoregressive rollout for open-world navigation.

Table 2: **Ablation study on two components of our proposed method**: the choice of Weight Transfer (WT) mechanisms and the use of Action Guidance (AG).

| Model | WT | AG | FVD ↓ | FID ↓ | SSIM ↑ | LPIPS ↓ | PSNR ↑ |
|---|---|---|---|---|---|---|---|
| Vid2World | Shift | | 29.9 | 7.85 | 0.799 | 0.185 | 21.5 |
| Vid2World | Masked | | 29.4 | 7.07 | 0.824 | 0.169 | 22.9 |
| Vid2World | Extrapolative | | 28.6 | 7.52 | 0.832 | 0.162 | 23.4 |
| Vid2World | Masked | ✓ | 25.8 | 6.84 | **0.840** | **0.159** | **23.9** |
| Vid2World | Extrapolative | ✓ | **22.4** | **6.16** | 0.839 | **0.159** | **23.9** |

in FID and a 71.1% performance gain in FVD compared to the best baseline configuration. These results demonstrate the superior visual fidelity and semantic consistency of our method, showcasing potential for leveraging video diffusion models to interactive neural game engines.

## 5.3 VID2WORLD FOR OPEN-WORLD NAVIGATION

Open-world navigation is a fundamental capability for autonomous agents, with broad applications to autonomous driving (Yuan et al., 2025) and robotics (Shah et al., 2022).

**Setup.** We evaluate on the RECON dataset (Shah et al., 2022), which uses a 3D $(x, y, \text{yaw})$ action space. Comparisons are made against two leading baselines, Navigation World Model (NWM) (Bar et al., 2025) and DIAMOND (Alonso et al., 2024). We also include NWM (+Ego4D), a variant of NWM co-trained with action-free videos, aiming for out-of-domain generalization. Unlike our model, which is restricted to sequential, autoregressive generation, NWM explicitly conditions on the prediction timestep $t$, allowing single-step prediction of a distant future frame. Accordingly, we evaluate against both baseline setups: single-step prediction and autoregressive rollout, at the dataset's native 4 fps rate. Especially in the autoregressive setup, by comparing our model to well-established baselines such as NWM and Diamond at different timesteps, this offers clear insight into our model's robustness against error accumulation. More details are listed in Appendix C.6.

**Results.** In the single-step prediction setting (Table 1), Vid2World achieves performance on par with NWM and surpasses NWM (+Ego4D) for 4 out of 6 evaluated metrics, even under error accumulation from autoregressive rollouts. This is a particularly surprising result since NWM is a state of the art model, and trained with significantly more computation and bypassing error accumulation via single-step prediction in this setup. It's also worth noting that our model's prediction horizon of 16 frames combined with a history length of 4 results in a total context length of 20—exceeding the training horizon of 16—demonstrating strong temporal generalization. In the autoregressive rollout setup (Figure 6), our model consistently produces predictions that are superior or comparable to NWM, while significantly outperforming DIAMOND baselines. Taken together, these results highlight the effectiveness of Vid2World in leveraging rich priors from action-free video data, obviating the prohibitive data requirements and training costs associated with pre-training on cross-domain action-labeled datasets.

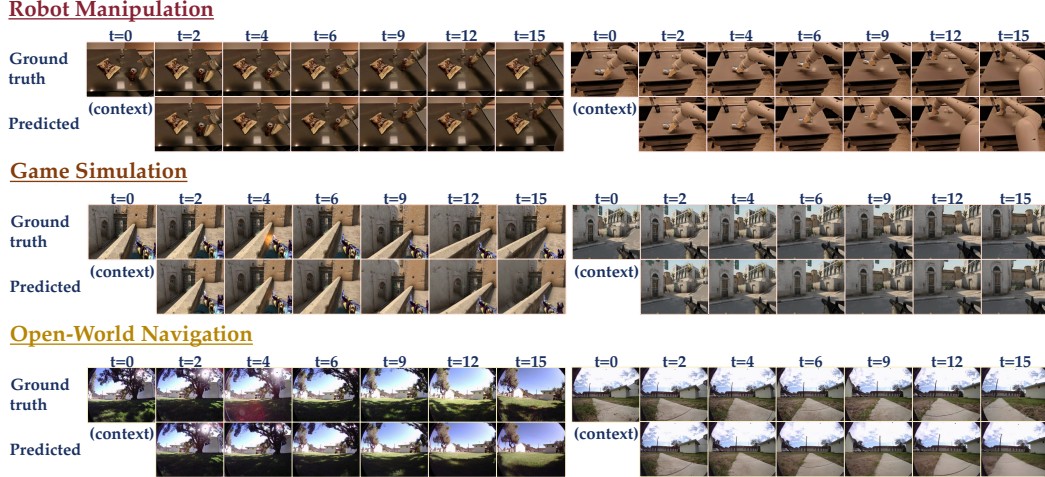

Figure 7: **Qualitative evaluation of Vid2World across various domains.** Zoom in for details. Extended examples can be found in Appendix D.

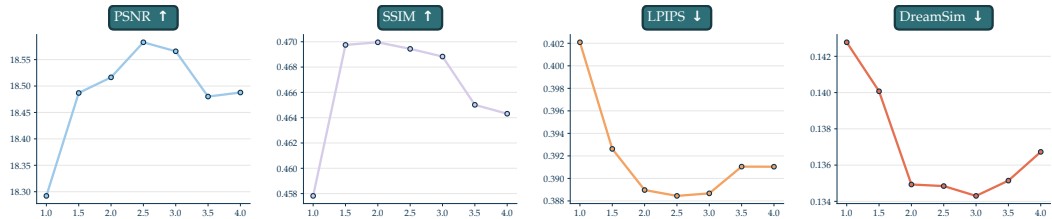

Figure 8: **Video Prediction Metrics as a function of Causal Action Guidance Scale ($\lambda$) in the CS:GO environment.** While increasing $\lambda$ initially improves performance by enforcing action alignment, excessive guidance leads to degradation due to over-sharpening artifacts.

## 5.4 ABLATION STUDY

To verify the effectiveness of components in our method, we focus on two questions: *(1) How critical is action guidance? (2) Which weight transfer mechanisms do best transfer?* Due to limited computational budgets, all models are trained for 30k gradient steps. Regarding the first question, as shown in Table 2, we observe that for both Extrapolative Weight Transfer and Masked Weight Transfer, enforcing action guidance yields better performance compared to their counterpart, which have never dropped out action in training and inference. We further validate the impact of guidance scale $\lambda$ on generation quality in Figure 8, detailed in Appendix D.5. Regarding the second question, both Masked and Extrapolative Weight Transfer yield better performance than Shift Weight Transfer, and utilizing Extrapolative Weight Transfer yields slightly better outcomes compared to Masked Weight Transfer. Hence, both techniques play a dominant role in the superior performance of Vid2World.

## 6 CONCLUSION

In this work, we transform passive video diffusion models into interactive world models. We propose Vid2World, introducing two key mechanisms—video diffusion causalization and causal action guidance—to support autoregressive, action-conditioned generation. Extensive experiments demonstrate that Vid2World achieves state-of-the-art performance in world modeling tasks and also effectively supports downstream decision-making. While this work marks a successful first attempt, it leaves plentiful space for further exploration. First, due to computational resource constraints, we are limited to employing a relatively lightweight video diffusion model as the base model. We envision that exploring larger-scale models (NVIDIA et al., 2025; Peng et al., 2025) may lead to better performance. Second, the training process remains relatively time-consuming. We look forward to future methods that can achieve comparable or even superior performance with fewer training steps.

ETHICS STATEMENT

This work complies with the ICLR Code of Ethics. Human evaluations were conducted with informed consent from volunteers. All datasets and pretrained models are publicly available and used under their respective licenses. We have carefully designed our experiments to minimize risks of bias or discrimination. We uphold the principles of transparency and integrity throughout the work.

REPRODUCIBILITY STATEMENT

We have taken concrete steps to ensure that our results are reproducible. All datasets and models used in this work are publicly available and do not require special access. We have included detailed descriptions of our experimental setup, including detailed algorithmic descriptions (Appendix B.1), training and inference procedures (Appendix B), hyperparameter configurations (Table 3), and evaluation protocols (Appendix C). We have released the full source code, configuration files, and trained model checkpoints at https://knightnemo.github.io/vid2world/ to ensure full reproducibility and future extensions.

ACKNOWLEDGMENTS

The authors would like to thank Chaoyi Deng, Ningya Feng, Xingzhuo Guo and Yuchen Zhang for helpful discussions and instructive suggestions. This work was supported by the Natural Science Foundation of China (U2342217), the Fundamental and Interdisciplinary Disciplines Breakthrough Plan of the Ministry of Education of China (JYB2025XDXM803), the Beijing Scholar Program, and the National Engineering Research Center for Big Data Software.

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

## A  THEORETICAL JUSTIFICATIONS AND EXTENDED DISCUSSION

### A.1  CONSTRUCTION OF COUNTER-EXAMPLE FOR SHIFT WEIGHT TRANSFER

We start by showing that even when the input sequence $\mathbf{z}_t \triangleq f(t)$ is $L$-smooth, Shift Weight Transfer (SWT) can still yield outputs with arbitrarily large error. Denote

$$\mathbf{y}_t^{\text{SWT}} = \sum\nolimits_{i=-m}^{m} w_i \mathbf{z}_{t+i-m} + \mathbf{b}.$$

**Counter-example.**  Consider the one-dimensional input $f(t) = \alpha t$, where $\alpha > 0$ is a scaling parameter. Clearly $f$ is $L$-smooth for any finite $L$ (since $f''(t) = 0$). The original convolution output at time $t$ should be

$$\mathbf{y}_t = \sum\nolimits_{i=-m}^{m} w_i \mathbf{z}_{t+i} + \mathbf{b}.$$

The error term now becomes:

$$||\mathbf{y}_t^{\text{SWT}} - \mathbf{y}_t|| = ||\sum\nolimits_{i=-m}^{m} w_i(\mathbf{z}_{t+i-m} - \mathbf{z}_{t+i})||$$

$$= \alpha m \cdot ||\sum\nolimits_{i=-m}^{m} w_i||.$$

**Implication.**  Even though $f$ is perfectly smooth (indeed linear), the approximation error of SWT grows endlessly as $\alpha \to \infty$. Hence, the error is *not controlled* by the smoothness constant $L$ alone. This shows that Shift Weight Transfer may catastrophically fail, motivating the more principled extrapolative construction.

### A.2  DETAILED DERIVATION FOR EXTRAPOLATIVE WEIGHT TRANSFER

From first principles, we posit that $\mathbf{z}_{t+k}$ can be linearly approximated over a window of $p$ past timesteps. Specifically, we perform linear regression on $\{(\mathbf{z}_\tau, \tau)\}_{\tau=t-p+1}^{t}$ and predict $\mathbf{z}_{t+k}$ based on the regression result $(\hat{\mathbf{z}}_{t+k}, t+k)$.

$$\mathbf{z}_{t+k} \approx \sum\nolimits_{j=0}^{p-1} \gamma_{k,j}\, \mathbf{z}_{t-j} + \boldsymbol{\beta}_k,$$

where $\boldsymbol{\gamma}_{k,\cdot}, \boldsymbol{\beta}_k$ are determined by a linear extrapolation (OLS) from the past $p$ features. Concretely, let $\tau_j \triangleq t - j$ and define the empirical mean and variance of the timestamps:

$$\mu = \frac{1}{p}\sum_{j=0}^{p-1}\tau_j = t - \frac{p-1}{2}, \qquad S = \sum_{j=0}^{p-1}(\tau_j - \mu)^2.$$

Then the regression prediction admits a closed form:

$$\hat{\mathbf{z}}_{t+k} = \sum_{j=0}^{p-1}\left(\frac{1}{p} + \frac{(t+k-\mu)(\tau_j-\mu)}{S}\right)\mathbf{z}_{t-j}.$$

Thus, the coefficients are explicitly

$$\gamma_{k,j} = \frac{1}{p} + \frac{(t+k-\mu)(\tau_j-\mu)}{S}, \qquad \boldsymbol{\beta}_k = \mathbf{0}.$$

Note that $\sum_j \gamma_{k,j} = 1$, hence the intercept $\boldsymbol{\beta}_k$ vanishes automatically, and the extrapolation is expressed as a weighted combination of past features $\mathbf{z}_{t-j} \in \mathbb{R}^d$.

Keeping in mind the design principle of maximally preserving the output representation of the original convolution, such that the new causal computation produces a similar result to the original non-causal one:

$$\sum\nolimits_{i=-m}^{m} w_i \mathbf{z}_{t+i} + \mathbf{b} = \sum\nolimits_{j=-2m}^{0} w'_j \mathbf{z}_{t+j} + \mathbf{b}'.$$

We rewrite the left-hand side as

$$\sum_{i=-m}^{m} w_i \mathbf{z}_{t+i} + \mathbf{b} = \sum_{i=-m}^{0} w_i \mathbf{z}_{t+i} + \sum_{i=1}^{m} w_i \mathbf{z}_{t+i} + \mathbf{b}$$

$$\approx \sum_{i=-m}^{0} w_i \mathbf{z}_{t+i} + \sum_{i=1}^{m} w_i \left( \sum_{j=0}^{p-1} \gamma_{i,j} \, \mathbf{z}_{t-j} + \boldsymbol{\beta}_i \right) + \mathbf{b}$$

$$= \sum_{i=-m}^{0} w_i \mathbf{z}_{t+i} + \sum_{j=0}^{p-1} \left( \sum_{i=1}^{m} \gamma_{i,j} w_i \mathbf{z}_{t-j} \right) + \sum_{i=1}^{m} w_i \boldsymbol{\beta}_i + \mathbf{b}.$$

Rearranging the terms with respect to $\mathbf{z}$ gives us:

$$w_j' = \mathbf{1}_{[j \geq -m]} \cdot w_j + \mathbf{1}_{[-p+1 \leq j \leq 0]} \cdot \sum_{i=1}^{m} \gamma_{i,-j} w_i, \quad \mathbf{b}' = \mathbf{b} + \sum_{i=1}^{m} w_i \boldsymbol{\beta}_i.$$

In the specialized case of $m = 1, p = 2$, $\mathbf{z}_{t+k}$ satisfies:

$$\mathbf{z}_{t+k} \approx (k+1)\mathbf{z}_t - k\mathbf{z}_{t-1}$$

Since $m = 1$, we can explicitly write out the three terms:

$$w_j' = \mathbf{1}_{[j \geq -m]} \cdot w_j + \mathbf{1}_{[-p+1 \leq j \leq 0]} \cdot \sum_{i=1}^{m} \gamma_{i,-j} w_i$$

$$= \begin{cases} w_0 + 2w_1, & j = 0 \\ w_{-1} - w_1, & j = -1 \\ 0, & j = -2 \end{cases}.$$

Also, $\mathbf{b}' = \mathbf{b}$. Since all temporal convolution layers in DynamiCrafter (Xing et al., 2024) have a kernel size of 3, we are restricted to using the formulation for $m = 1, p = 2$. However, we anticipate that extrapolating with higher terms may lead to better performance as well as more complicated technical designs, which we leave for future work.

## A.3 EXTRAPOLATIVE WEIGHT TRANSFER ERROR BOUND

**Proposition 1.** *Assuming the input sequence $\mathbf{z}_t \triangleq f(t)$ is generated by a twice-differentiable L-smooth function $f(t)$, the approximation error of the Extrapolative Weight Transfer (EWT) can be bounded by:*

$$\left\| \mathbf{y}^{\text{orig}} - \mathbf{y}^{\text{EWT}} \right\|_2 \leq \frac{L}{2} \sum_{i=1}^{m} |w_i| \left( i^2 + \frac{6p^2}{p+1} i + \frac{(p-1)(p-2)}{6} \right).$$

*Proof.* The total error is the weighted sum of the per-term extrapolation errors:

$$\left\| \mathbf{y}^{\text{orig}} - \mathbf{y}^{\text{EWT}} \right\|_2 \leq \sum_{i=1}^{m} |w_i| \cdot \left\| \mathbf{z}_{t+i} - \tilde{\mathbf{z}}_{t+i} \right\|_2. \tag{1}$$

We derive a complete bound for the per-term error $\|\mathbf{z}_{t+i} - \tilde{\mathbf{z}}_{t+i}\|_2$. Let $l^*(x) = f(t) + (x - t)f'(t)$ be the true tangent line at $t$, and $l(x)$ be the OLS fitted line. The error is $\|f(t+i) - l(t+i)\|_2$. We use the triangle inequality to decompose this error into three distinct sources:

$$\|f(t+i) - l(t+i)\|_2 \leq \|f(t+i) - l^*(t+i)\|_2 + \|l^*(t+i) - l(t+i)\|_2$$

$$\leq \underbrace{\|f(t+i) - l^*(t+i)\|_2}_{\text{(A) Taylor Error}} + \underbrace{\|f(t) - l(t)\|_2}_{\text{(B) Intercept Error at } t} + \underbrace{\|i(f'(t) - \hat{\mathbf{s}})\|_2}_{\text{(C) Propagated Slope Error}}.$$

We now bound each of these three terms.

**(A) Bounding the Taylor Error** By Taylor's theorem, there exists some $\xi \in (t, t+i)$ such that $f(t+i) - l^*(t+i) = \frac{i^2}{2} f''(\xi)$. Given $\|f''(\cdot)\|_2 \leq L$, this term is bounded by:

$$\|f(t+i) - l^*(t+i)\|_2 \leq \frac{L}{2} i^2.$$

**(B) Bounding the Intercept Error** This term, $\|f(t) - l(t)\|_2$, represents the error of the OLS prediction at time $t$. The prediction is $l(t) = \tilde{\mathbf{z}}_t = \sum_{j=0}^{p-1} \gamma_{0,j} \mathbf{z}_{t-j}$. We analyze the error component-wise using a Taylor expansion of $f(t-j)$ around $t$:

$$l(t) = \sum_{j=0}^{p-1} \gamma_{0,j} f(t-j) = \sum_{j=0}^{p-1} \gamma_{0,j} \left[ f(t) - jf'(t) + \frac{j^2}{2} f''(\xi_j) \right].$$

For the coefficient sums for $\gamma_{0,j} = \frac{1}{p} + \frac{(t-\mu)(\tau_j - \mu)}{S}$, we have:

$$\sum_{j=0}^{p-1} \gamma_{0,j} = 1, \tag{2}$$

$$\sum_{j=0}^{p-1} j\gamma_{0,j} = \frac{1}{p} \sum j + \frac{t-\mu}{S} \sum j(\tau_j - \mu) = \frac{p-1}{2} + \frac{(p-1)/2}{S}(-S) = 0. \tag{3}$$

Thus, the error simplifies to:

$$f(t) - l(t) = f(t) - \left( f(t) - 0 \cdot f'(t) + \sum_{j=0}^{p-1} \gamma_{0,j} \frac{j^2}{2} f''(\xi_j) \right) = -\frac{1}{2} \sum_{j=0}^{p-1} j^2 \gamma_{0,j} f''(\xi_j).$$

Taking the norm and the bound $\|f''(\cdot)\|_2 \le L$:

$$\|f(t) - l(t)\|_2 \le \frac{L}{2} \left| \sum_{j=0}^{p-1} j^2 \gamma_{0,j} \right|.$$

The sum can be calculated in closed form:

$$\begin{aligned}
\sum_{j=0}^{p-1} j^2 \gamma_{0,j} &= \frac{1}{p} \sum j^2 + \frac{t-\mu}{S} \sum j^2 (\tau_j - \mu) \\
&= \frac{(p-1)(2p-1)}{6} + \frac{(p-1)/2}{S} \left( -\frac{p(p-1)^2(p+1)}{12} \right) \\
&= \frac{(p-1)(2p-1)}{6} - \frac{(p-1)/2}{S} S(p-1) = -\frac{(p-1)(p-2)}{6}.
\end{aligned}$$

Therefore, the intercept error is bounded by:

$$\|f(t) - l(t)\|_2 \le \frac{L}{2} \frac{(p-1)(p-2)}{6}.$$

**(C) Bounding the Propagated Slope Error** This can be achieved by bounding the slope-estimation error:

$$\Delta_i \triangleq \|f'(t) - \hat{\mathbf{s}}\|,$$

for the OLS fit on uniformly spaced timestamps $\tau_j = t - j$. The OLS slope estimator is given by

$$\hat{\mathbf{s}} = \frac{\sum_{j=0}^{p-1}(\tau_j - \mu) f(\tau_j)}{\sum_{j=0}^{p-1}(\tau_j - \mu)^2} = \frac{1}{S} \sum_{j=0}^{p-1} (\tau_j - \mu) f(\tau_j).$$

Using the Taylor expansion $f(\tau_j) = f(t) - jf'(t) + \frac{j^2}{2} f''(\xi_j)$, we can distribute the sums into:

$$\hat{\mathbf{s}} = \frac{1}{S} \left( f(t) \sum (\tau_j - \mu) - f'(t) \sum j(\tau_j - \mu) + \frac{1}{2} \sum (\tau_j - \mu) j^2 f''(\xi_j) \right).$$

Using the properties in Equation 3 gives us

$$\hat{\mathbf{s}} = \frac{1}{S} \left( \mathbf{0} - f'(t)(-S) + \frac{1}{2} \sum (\tau_j - \mu) j^2 f''(\xi_j) \right) = f'(t) + \frac{1}{2S} \sum (\tau_j - \mu) j^2 f''(\xi_j).$$

Taking vector norm and using $\|f''(\cdot)\|_2 \leq L$ we get the uniform bound

$$\Delta_i = \|\hat{\mathbf{s}} - f'(t)\|_2 \leq \frac{1}{2S} \sum_{j=0}^{p-1} \left|(\tau_j - \mu)j^2\right| \cdot \|f''(\xi_j)\|^2 \leq \frac{L}{2S} \sum_{j=0}^{p-1} \left|(\tau_j - \mu)j^2\right|.$$

As

$$\sum_{j=0}^{p-1} \left|(\tau_j - \mu)j^2\right| \leq p^2 \sum_{j=0}^{p-1} |(\tau_j - \mu)| \leq p^3 \cdot \frac{p-1}{2},$$

therefore

$$\Delta_i \leq \frac{L}{2S} \cdot \frac{1}{2}(p-1)p^3 = \frac{L}{2} \cdot \frac{6p^2}{p+1}.$$

**Combining All Terms** Combining the three terms, we get the complete per-term error bound:

$$\|\mathbf{z}_{t+i} - \tilde{\mathbf{z}}_{t+i}\|_2 \leq \frac{L}{2}i^2 + \frac{L}{2} \cdot \frac{6p^2}{p+1}i + \frac{L}{2}\frac{(p-1)(p-2)}{6}.$$

Substituting this back into Equation 1:

$$\left\|\mathbf{y}^{\text{orig}} - \mathbf{y}^{\text{EWT}}\right\|_2 \leq \sum_{i=1}^{m} |w_i| \left(\frac{L}{2}i^2 + \frac{L}{2} \cdot \frac{6p^2}{p+1}i + \frac{L}{2}\frac{(p-1)(p-2)}{6}\right)$$

$$= \frac{L}{2} \sum_{i=1}^{m} |w_i| \left(i^2 + \frac{6p^2}{p+1}i + \frac{(p-1)(p-2)}{6}\right).$$

This completes the full proof. $\qquad\square$

### A.4 CAUSAL ACTION GUIDANCE AS PROBABILITY STEERING

In this section, we provide the proof for Theorem 4.1, formally demonstrating that our Causal Action Guidance mechanism is mathematically equivalent to sampling from a sharpened posterior distribution, effectively steering the generation process.

**Theorem 4.1** (Causal Action Guidance as Probability Steering). *Let $\mathcal{H}_t := ([\mathbf{x}_\tau]_{\tau<t}, [\mathbf{a}_\tau]_{\tau<t-1})$ denote the history context excluding the current action. Under standard score-based formulation,, the proposed score composition is mathematically equivalent to sampling from the following steered posterior distribution, where $\omega \propto (1 + \lambda)$ is a fixed constant:*

$$\tilde{p}(\mathbf{x}_t \mid \mathbf{a}_{t-1}, \mathcal{H}_t) \propto \underbrace{p(\mathbf{x}_t \mid \mathcal{H}_t)}_{\text{History-Consistent Prior}} \cdot \underbrace{\left(\frac{p(\mathbf{x}_t \mid \mathbf{a}_{t-1}, \mathcal{H}_t)}{p(\mathbf{x}_t \mid \mathcal{H}_t)}\right)^\omega}_{\text{Action Alignment}} \propto p(\mathbf{x}_t \mid \mathcal{H}_t) \cdot p(\mathbf{a}_{t-1} \mid \mathbf{x}_t, \mathcal{H}_t)^\omega.$$

*Proof.* We begin by establishing the relationship between the score functions and the probability densities. In the context of diffusion models (Ho et al., 2020; Song et al., 2021b), the trained noise prediction network $\epsilon_\theta(\mathbf{x}_t, \cdot)$ estimates the score of the data distribution $-\nabla_{\mathbf{x}_t} \log p(\mathbf{x}_t \mid \cdot)$ (up to a scaling constant).

Let $\mathcal{H}_t := ([\mathbf{x}_\tau]_{\tau<t}, [\mathbf{a}_\tau]_{\tau<t-1})$ denote the history context excluding the current action. We identify the two score components used in our method as:

$$\epsilon_{\text{cond}} \propto -\nabla_{\mathbf{x}_t} \log p(\mathbf{x}_t \mid \mathbf{a}_{t-1}, \mathcal{H}_t),$$
$$\epsilon_{\text{ucond}} \propto -\nabla_{\mathbf{x}_t} \log p(\mathbf{x}_t \mid \mathcal{H}_t).$$

Applying Bayes' rule to the conditional density $p(\mathbf{x}_t \mid \mathbf{a}_{t-1}, \mathcal{H}_t)$:

$$\log p(\mathbf{x}_t \mid \mathbf{a}_{t-1}, \mathcal{H}_t) = \log p(\mathbf{x}_t \mid \mathcal{H}_t) + \log p(\mathbf{a}_{t-1} \mid \mathbf{x}_t, \mathcal{H}_t) - \log p(\mathbf{a}_{t-1} \mid \mathcal{H}_t).$$

Since the term $\log p(\mathbf{a}_{t-1} \mid \mathcal{H}_t)$ is independent of $\mathbf{x}_t$, taking the gradient with respect to $\mathbf{x}_t$ yields the following decomposition of the conditional score:

$$\nabla_{\mathbf{x}_t} \log p(\mathbf{x}_t \mid \mathbf{a}_{t-1}, \mathcal{H}_t) = \nabla_{\mathbf{x}_t} \log p(\mathbf{x}_t \mid \mathcal{H}_t) + \nabla_{\mathbf{x}_t} \log p(\mathbf{a}_{t-1} \mid \mathbf{x}_t, \mathcal{H}_t),$$
$$\Leftrightarrow \boldsymbol{\epsilon}_{\text{cond}} = \boldsymbol{\epsilon}_{\text{ucond}} - \gamma \nabla_{\mathbf{x}_t} \log p(\mathbf{a}_{t-1} \mid \mathbf{x}_t, \mathcal{H}_t).$$

where $\gamma$ is a proportionality constant absorbed into the guidance scale. During sampling, the update rule for Causal Action Guidance can now be rewritten as :

$$\begin{aligned}
\boldsymbol{\epsilon}_{\text{guided}} &= (1 + \lambda)\boldsymbol{\epsilon}_{\text{cond}} - \lambda\boldsymbol{\epsilon}_{\text{ucond}} \\
&= (1 + \lambda)\left(\boldsymbol{\epsilon}_{\text{ucond}} - \gamma \cdot \nabla_{\mathbf{x}_t} \log p(\mathbf{a}_{t-1} \mid \mathbf{x}_t, \mathcal{H}_t)\right) - \lambda\boldsymbol{\epsilon}_{\text{ucond}} \\
&= \boldsymbol{\epsilon}_{\text{ucond}} - \gamma \cdot (1 + \lambda)\nabla_{\mathbf{x}_t} \log p(\mathbf{a}_{t-1} \mid \mathbf{x}_t, \mathcal{H}_t).
\end{aligned}$$

Recalling that $\boldsymbol{\epsilon}_{\text{guided}} \propto -\nabla_{\mathbf{x}_t} \log \tilde{p}(\mathbf{x}_t \mid \mathbf{a}_{t-1}, \mathcal{H}_t)$, we equate the gradients:

$$\nabla_{\mathbf{x}_t} \log \tilde{p}(\mathbf{x}_t \mid \mathbf{a}_{t-1}, \mathcal{H}_t) = \nabla_{\mathbf{x}_t} \log p(\mathbf{x}_t \mid \mathcal{H}_t) + \gamma \cdot (1 + \lambda)\nabla_{\mathbf{x}_t} \log p(\mathbf{a}_{t-1} \mid \mathbf{x}_t, \mathcal{H}_t).$$

Denoting $\omega := \gamma \cdot (1 + \lambda)$, integrating both sides with respect to $\mathbf{x}_t$ recovers the log-density, up to constant $C$:

$$\log \tilde{p}(\mathbf{x}_t \mid \mathbf{a}_{t-1}, \mathcal{H}_t) = \log p(\mathbf{x}_t \mid \mathcal{H}_t) + \log p(\mathbf{a}_{t-1} \mid \mathbf{x}_t, \mathcal{H}_t)^{\omega} + C.$$

Exponentiation on both sides yields the steered posterior distribution:

$$\tilde{p}(\mathbf{x}_t \mid \mathbf{a}_{t-1}, \mathcal{H}_t) \propto p(\mathbf{x}_t \mid \mathcal{H}_t) \cdot p(\mathbf{a}_{t-1} \mid \mathbf{x}_t, \mathcal{H}_t)^{\omega}.$$

This completes the proof. $\qquad\square$

### A.5 Clarification on Causal Terminology and Connections to Interventional World Models

In this work, we bridge concepts from video generation and world modeling, domains where the term "causal" carries distinct nuances. Throughout the paper, our usage of "causal" primarily refers to *Temporal Causality*, focusing on the sequential evolution of environment dynamics. However, specifically within our proposed *Causal Action Guidance*, the mechanism also implicitly embodies *Interventional Causality*, i.e. $p(\mathbf{s}_{t+1} \mid \mathbf{s}_t, \text{do}(\mathbf{a}_t))$. In this section, we clarify these two interpretations and discuss their connections to the broader causal inference literature.

**Temporal Causality .**    In the deep learning community, particularly in autoregressive sequence modeling, "causal" typically refers to the *temporal structure* or non-anticipative dependency. A model is temporally causal if the prediction at time $t$, $\mathbf{x}_t$, depends only on the past context $\mathbf{x}_{<t}$ and is independent of future information $\mathbf{x}_{>t}$. Our *Video Diffusion Causalization* (Section 4.1) strictly follows this definition. Standard video diffusion models utilize bidirectional attention, violating the arrow of time required for online interaction. Our method repurposes these architectures to strictly enforce temporal causality, establishing the structural foundation for autoregressive rollout.

**Interventional Causality.**    While our framework is built on temporal causality, our *Causal Action Guidance* (Section 4.2) naturally extends into the realm of *interventional causality* (Pearl, 2009). In causal inference, this refers to reasoning about the effect of an intervention, i.e., estimating $p(\mathbf{s}_{t+1} \mid \mathbf{s}_t, \text{do}(\mathbf{a}_t))$. By injecting the action signal $\mathbf{a}_t$ and utilizing classifier-free guidance to amplify its likelihood, we are effectively performing an intervention $\text{do}(\mathbf{a}_t)$. This forces the model to generate the specific counterfactual future resulting from that action, rather than a generic future based solely on observational correlations.

In Vid2World, we establish *temporal causality* as the necessary architectural prerequisite to enable online rollout, while leveraging *action guidance* to implicitly enforce *interventional causality*. Together, these components enable controllable and action-conditioned world simulation.

### A.6 Extended Discussion on Data Distribution Mismatch during Video Generation and World Modeling

While Vid2World is motivated by leveraging the broad visual experience and rich physical priors encoded in the pretrained video diffusion models, it is natural that the distribution of internet-scale pretraining video data differs from the agent-centric interaction data used in world modeling. To this end, we highlight two key dimensions of this data distribution gap:

1. **Scene Composition and Object Distribution**. Internet videos span a wide variety of environments, objects, and scene layouts, whereas world-model training datasets generally contain more controlled, task-centric settings with a narrower range of objects and interactions. Although this introduces a distribution mismatch, the underlying compositional regularities (e.g. object permanence, contact dynamics, spatial relations), are shared across all sources and serve as transferable priors.

2. **Motion Scale and Interaction Granularity**. Pretraining videos typically capture large-scale, coarse global motions, while world models focus on fine-grained, local interactions between agents and objects. Despite this difference in motion granularity, low-level physical regularities, such as temporal continuity, acceleration smoothness and occlusion dynamics, remain consistent, supporting transfer of motion priors into action-conditioned rollouts.

Even with these mismatches, we argue that by large-scale pretraining on diverse visual experience, the video diffusion model captures strong and broadly applicable visual and physical priors. This is especially true in our Vid2World approach. Building upon DynamiCrafter (Xing et al., 2024), an image animation model fine-tuned from the general-purpose video generation model VideoCrafter (Chen et al., 2023), Vid2World inherits both the broad visual experiences from internet-scale video pretraining and the structured motion priors essential for producing physically coherent dynamics, properties that are directly beneficial for action-conditioned generation.

### A.7 EXTENDED DISCUSSION ON UTILIZING VID2WORLD FOR DOWNSTREAM TASKS

While Vid2World demonstrates a successful first step at transferring video diffusion models to world models, due to the large parameter size of the pretrained model and the iterative process in diffusion, the model does not enjoy fast inference speed, compared to counterparts trained using teacher forcing, see Appendix E for further details. However, we fully acknowledge that the world model's performance in downstream tasks is a critical factor for evaluating the world model's applicability, especially for applications in domains such as embodied artificial intelligence.

We conduct the downstream task of Real2Sim Policy Evaluation in the RT-1 environment (Section 5.1), a challenging task validating the model's effectiveness in discriminating the performance of different policies as well as serving as a reference of the policy's real world success rate. Due to limited computation resources, we did not explore training agents via reinforcement learning with our world models, as reinforcement learning is notoriously known as being sample inefficient.

As the model scale grows increasingly larger, the computation cost grows higher accordingly. Even at industry-level world models, such as Genie 3 (Google, 2025), V-JEPA 2 (Meta, 2025) and PAN (PAN-Team, 2025), downstream tasks especially reinforcement learning are not conducted. We believe this is a collective effort where multiple domain may require breakthroughs:

1. **Novel world modeling methods with faster inference speed**: Current high-fidelity world models are dominated by diffusion models. With recent advances in one-step and few-step generative models showing feasibility in domains such as text to image generation (Song et al., 2023; Boffi et al., 2025; Geng et al., 2025), we anticipate world modeling methods that generate high-fidelity future predictions with improved inference latency.

2. **Sample-Efficient Methods for Model-Based Reinforcement Learning**: Reinforcement Learning is notoriously known for being sample inefficient. We strongly believe that designing model-based reinforcement learning algorithms with improved sample efficiency (both for offline and online reinforcement learning) may be crucial to fully unlock the potential of training policy model inside world models with reinforcement learning.

3. **Hardware Acceleration Methods for Inference Speedup**: Aside from algorithmic innovations, we strongly believe hardware speedup methods are necessary to unleash the full capabilities of world models in downstream tasks, especially planning, control and reinforcement learning. Such improvement may include novel methods for implementing KV Cache in the context of world modeling (Yin et al., 2024; Huang et al., 2025; Yang et al., 2025), better utilization of GPU locality, and developing hardware-friendly software packages that are easier to use.

## B VID2WORLD IMPLEMENTATION DETAILS

### B.1 ALGORITHM PSEUDO-CODE

In this subsection, we provide the pseudo-code for training and autoregressive inference of Vid2World.

---

**Algorithm 1** Vid2World Training

1: **Input**: Model $\theta$, Trajectory dataset $\mathcal{D}$.
2: **loop**
3:     Sample trajectory $[\mathbf{x}_t^{\text{gt}}, \mathbf{a}_t^{\text{gt}}]_{0:T}$ from $\mathcal{D}$
4:     **for** $t = 0, ..., T$ **do**
5:         $\mathbf{x}_t^{k_t} \sim q(\cdot \mid \mathbf{x}_t^{\text{gt}}, k_t), k_t \sim \mathcal{U}[0, K]$
6:         $\tilde{\mathbf{a}}_t = \begin{cases} \varnothing, & \text{w.p. } p, \\ \mathbf{a}_t^{\text{gt}}, & \text{o/w.} \end{cases}$
7:         $\boldsymbol{\epsilon}_t = \dfrac{\mathbf{x}_t^{k_t} - \sqrt{\bar{\alpha}_{k_t}} \mathbf{x}_t^{\text{gt}}}{\sqrt{1 - \bar{\alpha}_{k_t}}}$
8:         $\hat{\boldsymbol{\epsilon}}_t = \boldsymbol{\epsilon}_\theta([\mathbf{x}_\tau^{k_\tau}]_{\tau \le t}, [\tilde{\mathbf{a}}_\tau]_{\tau < t}, [k_\tau]_{\tau \le t})$
9:     **end for**
10:     $\mathcal{L} = \text{MSELoss}([\hat{\boldsymbol{\epsilon}}_1, ..., \hat{\boldsymbol{\epsilon}}_n], [\boldsymbol{\epsilon}_1, ..., \boldsymbol{\epsilon}_n])$
11:     Backprop with $\mathcal{L}$ and update $\theta$
12: **end loop**
13: **Return** Model $\theta$.

---

**Algorithm 2** Auto-Regressive Sampling

1: **Input:** Model $\theta$, Initial observation $x_0$, Action sequence $[\mathbf{a}_t]_{0:T-1}$, Action guidance scale $\lambda$.
2: **Initialize** $\mathbf{x}_t \sim \mathcal{N}(\mathbf{0}, \sigma_K^2 \mathbf{I}), \forall t \in 1, ..., T$.
3: **for** $t = 1, ..., T$ **do**
4:     **for** $k = K, ..., 0$ **do**
5:         $\hat{\boldsymbol{\epsilon}} = \boldsymbol{\epsilon}_\theta([\mathbf{x}_\tau]_{\tau \le t}, [\mathbf{a}_\tau]_{\tau < t}, [0, ..., 0, k])$
6:         **if** $\lambda \neq 1$ **then**
7:             $\boldsymbol{\epsilon}_{\text{uc}} = \boldsymbol{\epsilon}_\theta([\mathbf{x}_\tau]_{\tau \le t}, [\mathbf{a}_{\tau < t-1}, \varnothing], [0, ..., 0, k])$
8:             $\hat{\boldsymbol{\epsilon}} \leftarrow (1 + \lambda)\hat{\boldsymbol{\epsilon}} - \lambda \boldsymbol{\epsilon}_{\text{uc}}$
9:         **end if**
10:         $\mathbf{w} \sim \mathcal{N}(\mathbf{0}, \mathbf{I})$
11:         $x_t \leftarrow \frac{1}{\sqrt{\alpha_k}}(\mathbf{x}_t - \frac{1 - \alpha_k}{\sqrt{1 - \bar{\alpha}_k}}\hat{\boldsymbol{\epsilon}}) + \sigma_k \mathbf{w}$
12:     **end for**
13: **end for**
14: **Return** $\mathbf{x}_{0:T}$.

---

### B.2 MODEL DETAILS

**Base Model Details.** The pre-trained model DynamiCrafter (Xing et al., 2024) is a state-of-the-art latent video diffusion model conditioned on text and image, with its full-sized version ranking high on the VBench leaderboard (Huang et al., 2024). It builds on the Stable Diffusion variational autoencoder (Rombach et al., 2022) and trains a 3D U-Net for video generation using web-scale video data. Specifically, starting from the pre-trained VideoCrafter T2V model (Chen et al., 2023), DynamiCrafter introduces a dual-stream conditional image injection paradigm: in one stream, CLIP (Radford et al., 2021) image encoder embeddings are fed into the U-Net via cross-attention; in the other, images are encoded into VAE latents, which are then replicated along the channel dimension for the full video length and concatenated with the initial noise latents. This mechanism simultaneously injects text-aligned semantic representations and fine-grained visual details, improving video quality. For the noise level $k$, the model injects such information into the diffusion network by firstly using sinusoidal embedding to transform it into a vector, which is subsequently fed into a two-layer MLP, obtaining a learned embedding. The embedding is then added to the convolutional features to provide the noise level condition. Since the base model only contains temporal convolution layers with kernel size 3, our Extrapolative Weight Transfer Method is applied using hyperparameters $m = 1, p = 2$.

**Image Preprocessing.** For all of our experiments, we use the publicly released DynamiCrafter model at $320 \times 512$ resolution, which has 1.1B trainable parameters. During data preprocessing, we resize the shorter side to 320 px while preserving the aspect ratio. After resizing, if the longer side remains below 512 px, we pad with black borders up to 512 px; otherwise, we take the other approach: resizing the longer edge to 512 px, and pad with black borders on the height dimension. This setup is used in both training and inference. For evaluation metrics calculation, we resize the model output to the baseline method's resolution. For instance, in CS:GO, we calculate the metrics by firstly cropping out the black paddings in the model output, followed by resizing to $150 \times 280$ resolution.

**Noise-level Conditioning.** The structure of noise level embedding layers naturally supports the transformation to different noise scales at different frames. Specifically, instead of broadcasting the identical noise level sinusoidal embedding along the temporal axis, we use the independently sampled noise level at each frame, stacking it in the temporal dimension.

**Action Conditioning.** For action conditioning, we inject frame-level action conditions into the base model, similar to the injection of noise levels. For cases where actions are discrete, we alter the first layer of the noise conditioning network into a learned embedding layer. For cases where the action space is continuous, we simply switch the first layer to a linear projection. The embedding obtained through action conditioning is later integrated with the noise conditioning through element-wise addition.

## B.3 TRAINING DETAILS

We use the $320 \times 512$ version of DynamiCrafter (Xing et al., 2024) as the base model for all experiments. For robot manipulation, game simulation as well as open-world navigation tasks, we train for 100k gradient steps; for ablation studies, all models are trained for 30k steps. The training is conducted using $4 \times 40$GB NVIDIA A100 GPUs.

## B.4 INFERENCE DETAILS

During autoregressive rollout, we denoise the current frame by fixing noise levels at the history frames to be zero, whereas denoising the current frame using DDIM (Song et al., 2021a). In practice, following diffusion forcing (Chen et al., 2024), we add a small noise $k_{\text{small}}$ uniformly to history frames. Under all settings in this paper, concerning action guidance, we apply a guidance scale of 2.5 for our experiments, as well as a guidance rescale factor (Lin et al., 2024) of 0.7. We believe that the optimal values of these hyperparameters are related to domains, and an extensive hyperparameter search can lead to even better performance. A detailed list of hyperparameters regarding the model architecture, training, and inference process is shown in Table 3.

Table 3: Hyperparameters for Vid2World

| Hyperparameter | Value |
|---|---|
| **Architecture** | |
| *Base Model:* | |
| Resolution | $320 \times 512$ |
| Latent Diffusion | True |
| Downsample Ratio $f$ | 8 |
| $z$-shape | $32 \times 32 \times 4$ |
| U-Net Chaneels | 320 |
| *Noise level Conditioning:* | |
| Embedding dimension | 1024 |
| *Action Conditioning:* | |
| Embedding dimension | 1024 |
| *Other Conditioning:* | |
| Language condition | Empty Sequence |
| FPS condition | 3 |
| Image condition | First frame |
| **Training** | |
| Learning rate | $1.0 \times 10^{-5}$ |
| # training steps | 100k |
| Batch size per GPU | 2 |
| # GPUs | 4 |
| Accumulate gradient batches | 2 |
| GPU-type | A100-40GB |
| *Diffusion Setup:* | |
| Diffusion steps $K$ | 1000 |
| Noise schedule | Linear |
| $\beta_0$ | 0.00085 |
| $\beta_K$ | 0.0120 |
| Noise level along Temporal Axis | iid. samples |
| *Data Processing:* | |
| Input video length | 16 |
| Normalize | [-1,1] |
| Input resize | Resize, Center-Crop |
| Brightness | [0.9,1.1] |
| Contrast | [0.9,1.1] |
| Saturation | [0.9,1.1] |
| Hue | [-0.05,0.05] |
| *Causalization:* | |
| Mixed weight transfer | True |
| Causal Mask for Temporal attention | True |
| *Action Conditioning:* | |
| Dropout rate $p$ | 0.2 |
| Sampling along Temporal Axis | iid. samples |
| **Sampling** | |
| Sampler | DDIM |
| Steps | 50 |
| Timestep spacing | Uniform trailing |
| Action Guidance scale | 2.5 |
| Guidance rescale | 0.7 |
| $k_{\text{small}}$ | 20 |

## C EXPERIMENTAL DETAILS

### C.1 DATASET DETAILS

**RT-1 Details.** RT-1 (Brohan et al., 2023) is a widely used dataset consisting of real-world robot experiences, spanning multiple robot manipulation tasks, including opening drawers, closing drawers, picking and placing. Each episode is sampled at an fps of 3, with the embodiment, a robot arm, performing certain tasks. In addition to video frames, it also records action sequences as well as annotated language prompts. In our setup, we use the observations obtained by RGB cameras, as well as the action sequence.

**CS:GO Details.** We use the publicly released dataset collected by Pearce & Zhu (2022). It contains different subsets of human players interacting with the CS:GO maps, spanning from expert-level to novice players. Here, we use the largest subset in their dataset, `dataset_dm_scraped_dust2`, which contains 5.5M frames (95 hours) of online human gameplay from the map *Dust II*. The dataset is created by scraping user behaviors on online servers, offering a diverse set of interactions from policies of all sorts. For each timestep, the actions are represented as an array of discrete values.

**RECON Details.** RECON (Shah et al., 2022) is a well-known open-world navigation dataset. It consists of 40 hours across 9 open-world environments, collected using a Clearpath Jackal UGV platform. The dataset is collected at a fps of 4, and The action space is defined as a 3D vector $a_t = (x, y, \text{yaw})$, where $(x, y) \in \mathbb{R}^2$ denotes translation along the forward/backward and left/right axes, and $\text{yaw} \in \mathbb{R}$ denotes the change in rotation angle. Formally, each action is given by the proprioceptive state difference between timesteps, i.e., $a_t = s_{t'} - s_t$, where $s_t$ is the agent's proprioceptive state and $t'$ denotes either the next timestep or a future timestep of interest (as in NWM).

### C.2 METRICS FOR VIDEO PREDICTION

For Robot Manipulation, Game Simulation and Open-World Navigation tasks, we adopt commonly used video prediction metrics for image or video generation tasks. These metrics measure either the pixel-level or the semantic-level similarity between the generated videos and the ground truth videos. For metrics calculated on each image, the values are obtained by extracting all frames and treating them as independent images for feature extraction and statistical estimation.

Next, we provide a description for each metric:

**FID.** We compute the Fréchet Inception Distance (FID) introduced by Heusel et al. (2017). FID measures the Fréchet distance between two multivariate Gaussians fitted to Inception-v3 activations of real and generated frames. Specifically, let $\mu_r, \Sigma_r$ and $\mu_g, \Sigma_g$ denote the empirical means and covariances of these activations for real and generated frames, respectively. FID is defined as:

$$\text{FID}(P_r, P_g) = \|\mu_r - \mu_g\|_2^2 + \text{Tr}\left(\Sigma_r + \Sigma_g - 2\left(\Sigma_r \Sigma_g\right)^{1/2}\right).$$

**FVD.** *Fréchet Video Distance* (FVD), introduced by Unterthiner et al. (2018), generalizes FID by embedding entire video clips via a pre-trained Inflated 3D ConvNet (I3D) and computing the Fréchet distance between the resulting feature distributions of real and generated videos. Concretely, let $P_r$ and $P_g$ be the distributions of I3D activations for real and generated videos, respectively, with empirical means $\mu_r, \mu_g$ and covariances $\Sigma_r, \Sigma_g$. FVD is then defined as:

$$\text{FVD}(P_r, P_g) = \|\mu_r - \mu_g\|_2^2 + \text{Tr}\left(\Sigma_r + \Sigma_g - 2\left(\Sigma_r \Sigma_g\right)^{1/2}\right).$$

**SSIM.** Structural Similarity Index Measure (SSIM) (Wang et al., 2004) quantifies perceptual similarity by jointly comparing the luminance, contrast, and structural information between two image patches. Given a pair of patches $x$ and $y$, let $\mu_x, \mu_y$ be their mean intensities, $\sigma_x^2, \sigma_y^2$ their variances, and $\sigma_{xy}$ their covariance. The SSIM index is calculated using:

$$\text{SSIM}(x, y) = \frac{(2\mu_x \mu_y + C_1)(2\sigma_{xy} + C_2)}{(\mu_x^2 + \mu_y^2 + C_1)(\sigma_x^2 + \sigma_y^2 + C_2)},$$

where $C_1 = (K_1 L)^2$ and $C_2 = (K_2 L)^2$ are stability constants with $L$ the pixel dynamic range. For our purpose, we compute SSIM over an $11 \times 11$ Gaussian-weighted sliding window and average the local SSIM values to obtain a mean SSIM (MSSIM) per frame; the final video-level SSIM score is the average MSSIM across all sampled frames.

**LPIPS.** Learned Perceptual Image Patch Similarity (LPIPS) (Zhang et al., 2018) measures perceptual similarity by comparing deep feature activations of real and generated frames across multiple layers of a pre-trained network. Specifically, let $\hat{f}^l(x)$ and $\hat{f}^l(y)$ be the unit–normalized activations at layer $l$ for inputs $x$ and $y$, and $w_l$ the learned channel-wise weights. LPIPS is computed via:

$$\text{LPIPS}(x, y) = \sum_l \frac{1}{H_l W_l} \sum_{h=1}^{H_l} \sum_{w=1}^{W_l} \left\| w_l \odot \left( \hat{f}_{h,w}^l(x) - \hat{f}_{h,w}^l(y) \right) \right\|_2^2. \tag{4}$$

It is worth noting that for evaluation in RT-1 and CS:GO, we use VGG (Simonyan & Zisserman, 2015) as the feature extraction network, whereas in RECON, we use AlexNet (Krizhevsky et al., 2012) as the network, following baselines.

**PSNR.** Peak Signal-to-Noise Ratio (PSNR) (Hore & Ziou, 2010) quantifies pixel-level fidelity by comparing the maximum possible pixel intensity to the mean squared error (MSE) between two frames. PSNR is defined as:

$$\text{PSNR}(x, y) = 10 \log_{10} \frac{L^2}{\text{MSE}(x, y)},$$

where $L$ is the maximum pixel value (e.g. 255 for 8-bit images).

**DreamSim.** DreamSim (Fu et al., 2023) is a relatively new metric for measuring perceptual image similarity, which aims to evaluate perceptual similarity. This is accomplished by comparing deep features from a neural network. The resulting metric is better aligned with human perception.

## C.3 Details of Vid2World for Robot Manipulation.

### C.3.1 Implementation

We make use of the RT-1 (Brohan et al., 2023) dataset. To align with the baseline evaluation methods, we randomly split 4361 episodes as the holdout set, using the remaining 82851 episodes as the training set. In this case, since the action space is continuous, we use a linear layer as the first layer to add the action condition, as described in Appendix B.

Following baseline (Rigter et al., 2024), we train the model for up to 100k gradient steps on $4 \times$ A100, which takes less time (6.4 days) than the seven days reported for training baseline methods. During training, the model inputs are video and action sequence segments of length 16. At test time, we randomly sample 1024 episodes from the evaluation set, and sample a segment of 16 frames for each episode. The model is provided with the first frame of the segment as well as the action sequence, and the metric is calculated on all 16 frames, the same as baseline methods.

### C.3.2 Baselines

We compare Vid2World with several baselines, all utilizing the same base model (Dynamicrafter (Xing et al., 2024), resolution $320 \times 512$), while differing in their transfer methods. It is worth noting that for all baseline methods in this setting, the model is transferred without enforcing causality, neglecting the need for interactiveness; i.e., the models are still trained and sampled with homogeneous noise levels in all frames and the model is still architecturally non-causal. Therefore, the transferred models are unable to perform autoregressive rollout. During testing, the models generate videos in a non-autoregressive manner. Next, we provide a brief introduction to each baseline method:

**Action-Conditioned Fine-tuning.** In this approach, all parameters of the pre-trained model are fine-tuned on the action-conditioned dataset. For each timestep $t$ of the noisy video $\mathbf{x}$, the corresponding action $\mathbf{a}^t$ is embedded to compute the action embedding $\mathbf{e_a}^t$ using an embedding table for discrete

actions or a linear layer for continuous actions. For RT1, action embeddings are both concatenated with and added to the corresponding timestep embeddings.

**Language-Conditioned Fine-tuning.** Language-Conditioned Fine-tuning fine-tunes the pre-trained model using a textual description of each video. Each description is embedded via CLIP (Radford et al., 2021) and incorporated through cross-attention following the approach of the original model.

**ControlNet (Zhang et al., 2023).** ControlNet freezes the parameters of the pre-trained model and creates a trainable copy of its UNet encoder. The trainable branch is conditioned on the action signal and connected to the original decoder via zero-initialized convolutions. In this work, ControlNet is employed with the aim of incorporating action-conditioning into the diffusion process.

**Classifier Guidance (Dhariwal & Nichol, 2021)** A classifier $f_\phi(a \mid x_i)$ is trained on noisy images $x_i$ to predict actions. With weight $w$, this classifier steers the diffusion sampling process toward samples that are consistent with the specified actions. The resulting noise prediction is

$$\bar{\epsilon}_{\text{final}}(x_i, a, i, x_0) = \epsilon_{\text{pre}}(x_i, i, x_0) - \sqrt{1 - \overline{\alpha_t}} w \nabla_{x_i} \log f_\phi(a|x_i).$$

## C.4 Details of Real2Sim Policy Evaluation

Real2Sim Policy Evaluation (Li et al., 2025) aims to evaluate policies using simulation as a surrogate for the real world, serving as an indicator of the performance of different policies. This interaction between the policy and the simulation environment requires world models to generate images in an interactive manner. A well-performing model should be capable of distinguishing successful trajectories from failure cases by autoregressively simulating the outcomes of different policy actions.

We employ *Vid2World* as the world model to evaluate three policies: RT-1 (Begin), RT-1 (15%), RT-1 (Converged), taken for different stages of RT-1 (Brohan et al., 2023) training. Specifically, we sample $N$ trajectories from the RT-1 dataset for the given task and extract their initial frames. These frames are provided to each RT-1 policy to generate actions, which are then fed into the world model to simulate the next frame. The policy continues to act on these imagined frames in an iterative manner.

For the first $L$ frames, new frames are generated autoregressively based on all previously observed frames. Beyond this point, each subsequent frame is generated based on a sliding window of the most recent $L$ frames. This process continues until a sequence of length $H$ is produced. We then employ a verifier to determine whether each trajectory is successful, and compute the overall success rate accordingly. In our experiments, we sample trajectories from the "close drawer" task in the RT-1 dataset. For each policy, we use sample number $N = 50$, sliding window length $L = 10$, and rollout horizon $H = 40$. For simplicity, we use human evaluation as the verifier $\psi$.

The complete procedure is described in Algorithm 3.

---

**Algorithm 3** Real2Sim Policy Evaluation

**Require:** World model $P(\mathbf{o}_{t+1}|\mathbf{o}_{\leq t}, \mathbf{a}_{\leq t})$, policy $\pi(\mathbf{a}_t|\mathbf{o}_t)$, task $\kappa$, initial frame set $\mathcal{D}_\kappa$, trajectory success verifier $\psi(\mathbf{o}_{0:H}) \to \{0, 1\}$.
1: **Init** success_count $\leftarrow 0$
2: **for** $n = 0, ..., N$ **do**
3:     **Sample** initial frame $\mathbf{o}_0$ from $\mathcal{D}_\kappa$
4:     **for** $t = 0, ..., H$ **do**
5:         **Sample** $\mathbf{a}_t \sim \pi(\cdot \mid \mathbf{o}_t)$
6:         **if** $t < L$ **then**
7:             $\mathbf{o}_{t+1} \sim P(\mathbf{o}_{t+1}|\mathbf{o}_{\leq t}, \mathbf{o}_{\leq t})$
8:         **else**
9:             $\mathbf{o}_{t+1} \sim P(\mathbf{o}_{t+1}|\mathbf{o}_{t-L:t}, \mathbf{a}_{t-L:t})$
10:         **end if**
11:     **end for**
12:     success_count $\leftarrow$ success_count $+ \psi(\mathbf{o}_{0:H})$
13: **end for**
14: **Return** success_rate $= \frac{1}{N} \cdot$ success_count

---

We provide the instructions for human verification below:

---

**Instruction for Human Verification**

Watch each clip of the robot attempting to close a drawer and decide if the attempt succeeds or fails: label Success when, by the final frame, the drawer face sits flush with the cabinet frame (no visible gap and no rebound); label Failure when any gap remains, the drawer re-opens after contact, the robot jams or stops short, or the view prevents you from confirming full closure.

---

### C.5 DETAILS OF VID2WORLD FOR GAME SIMULATION

#### C.5.1 IMPLEMENTATION

We utilize the largest subset in the CS:GO dataset (Pearce & Zhu, 2022). Following DIAMOND (Alonso et al., 2024), we use exactly the same holdout set of 0.5M frames (corresponding to 500 episodes, or 8 hours) for testing. As actions are discrete values in this domain, the first layer in the action is injected via a learned embedding layer. For training and evaluation purposes, we use segments of 16 frames. For evaluation, since DIAMOND (Alonso et al., 2024) requires 4 frames as history context, we autoregressively generate frames from four consecutive history frames, until a sequence length of 16 is reached. In this experiment, the metrics are calculated only on the predicted frames, excluding frames used for conditioning. Since the output of the baseline method, DIAMOND, is in a resolution of $150 \times 280$, we downsampled our generated image to match this corresponding resolution. For our model, we train for 100k steps.

#### C.5.2 BASELINES

We use DIAMOND (Alonso et al., 2024), a state-of-the-art autoregressive world model as the baseline. It treats the world modeling task as an image generation problem, which learns an image diffusion model based on the previous four observations and actions. In practice, the input image diffusion model is downsampled, and a separate upsampler is learned to upsample the diffusion model's output to higher resolutions. Here, we use the publicly released checkpoints of DIAMOND, which contain both the diffusion model and the upsampler. We evaluated both sampling configurations provided by the authors, namely:

1. DIAMOND-Fast: Under this configuration, the model generates images with lower fidelity in exchange for faster inference speed, necessary for interactive gaming.

2. DIAMOND-HQ: This is the configuration where the generated images have higher fidelity, coming at the cost of slower inference speed.

We test our model's performance with baseline performances using exactly the same test set. Additional generation results can be viewed in Appendix D.2.

### C.6 DETAILS OF VID2WORLD FOR OPEN-WORLD NAVIGATION

#### C.6.1 IMPLEMENTATION

We utilize the RECON dataset (Shah et al., 2022), a well-celebrated dataset for open-world navigation. Following baseline implementations, we split the data into two parts: 9,468 videos for training and 2367 videos for evaluation, using exactly the same data split. Since the action space is continuous, we use linear projection as the first layer for injecting actions. During training, we preprocess the image into $320 \times 512$ resolutions by padding $320 \times 320$ with black borders. During evaluation, we cut out the black borders and downsample the image to $224 \times 224$, making it comparable with baselines. During training, we use a context length of 16, with no downsampling in the temporal fps. Since the dataset is collected at 4 fps, for our evaluations into 4s into the future, the model is provided with a history of 4 frames (following baselines) and predicts a sequence of 16 frames, creating a total context length of 20 frames, which is longer than the training horizon. In our experiments, we are focused on two setups: Single-Step Prediction and autoregressive Prediction.

**Single-Step Prediction.** The Single-Step Prediction set contains 500 video segments. Since NWM is capable of single-step prediction of a future timestep within its training horizon, the model is evaluated given 4 frames of history context and asked to single-shotedly predict the observations at

4s into the future. Our model, however, must generate predictions in a sequential manner; hence, our evaluations are done using autoregressive inference. It is worth noting that this makes the problem significantly harder, as the model's prediction will degrade with respect to the rollout horizon due to error accumulation. The results are shown in Table 1. For image generation metrics (i.e., all metrics except FVD), we report the results at the predicted frame, different from RT-1 and CS:GO evaluations, where we report means across all predicted frames. For FVD, we report the metric acquired by evaluating the video sequence of all 16 predicted frames. For LPIPS, PSNR and Dreamsim, we take the results from of NWM directly from the reported numbers in their paper, whereas for FVD, FID and SSIM, we report our evaluated numbers, which are obtained by building on the official implementation of NWM.

**autoregressive Predictions.** In this setup, the evaluation set consists of 150 video segments. Here, baseline methods as well as Vid2World conduct inference via autoregressive rollout. For baseline methods, except for the normal version of predicting future frames at 4 fps, there is also a downsampled version, which predicts future frames at 1 fps. This results in fewer autoregressive rollout steps, potentially leading to less error accumulation. For results in this domain, following baselines, we evaluate our model for 5 parallel runs using different random seeds, and report the mean and std.

### C.6.2 BASELINES

Here we consider two state-of-the-art baselines: Navigation World Models (NWM) (Bar et al., 2025) and DIAMOND (Alonso et al., 2024).

**NWM.** Navigation World Model (Bar et al., 2025) is a state-of-the-art model, built on a novel architecture CDiT. At its core, the model takes in action as well as the predicted timestep as conditions, and the backbone, follows the architecture of DiT (Peebles & Xie, 2023) used in image generation. This equips the model with the ability to single-step predict a timestep in the future. Here we use CDiT-XL, a 1B model trained on various action-labeled cross-domain data, leveraging 4 history frames as context. The original autoregressive setup is 4 fps, and 1 fps denotes the autoregressive rollout by predicting the future 1 second from the current time. We also consider a model variant: NWM+Ego4D, which was co-trained with action-free video data to improve out-of-distribution generalization. It is worth noting that the NWM model is trained on 8 nodes, each with 8 Nvidia H100 GPUs, trained for 100k (NWM) / 200k (NWM+Ego4D) gradient steps using a batch size of 1024. This is significantly more computationally expensive than Vid2World's setup, with is trained on 4 Nvidia A100 40GB GPUs for 100k gradient steps using a batch size of 32.

**DIAMOND.** DIAMOND (Alonso et al., 2024) is also the baseline we used for CS:GO. In this setup, following NWM, the model is trained from various cross-domain data, and inference is done using autoregressive generation. Additionally, we include DIAMOND (1fps), which is a model trained using observations and actions at intervals of 1 second.

We use exactly the same training and test split, and the same evaluation samples; showcases of generation results are included in Appendix D.3.

## D    ADDITIONAL EXPERIMENTS AND VISUALIZATION RESULTS

In this section, we provide additional visualization results for our proposed Vid2World model. Generated results from our model are obtained by autoregressive rollout. In Section D.1, we include visual results for the RT-1 dataset in the video prediction task. In Section D.2, we provide generated results in the CS:GO environment under the video prediction task. In Section D.3, we include showcases of Vid2World generation in RECON environment. In Section D.4, we provide some generated examples for the Real2Sim Policy Evaluation experiments.

### D.1    GENERATION RESULTS OF RT-1

We provide additional visualization results for Vid2World on the RT-1 Dataset in Figure 10. As shown in the figure, our model makes video predictions that accurately represent the environment dynamics. Our world model generates physically plausible frame sequences with high fidelity, offering great potential in video prediction tasks.

However, limitations still exist. We provide two examples of such limitations in Figure 9. These fall into two categories:

1. **Failing to predict fine-grained control**: In the upper case of Figure 9, the model predicts the moving directions of the robot arm successfully, but fails to capture the gripper's control over the green bag.

2. **Regressing to more familiar scenes**: In the lower case of Figure 9, although the robot movement is mostly correct, the grasped object changes to a more often seen object.

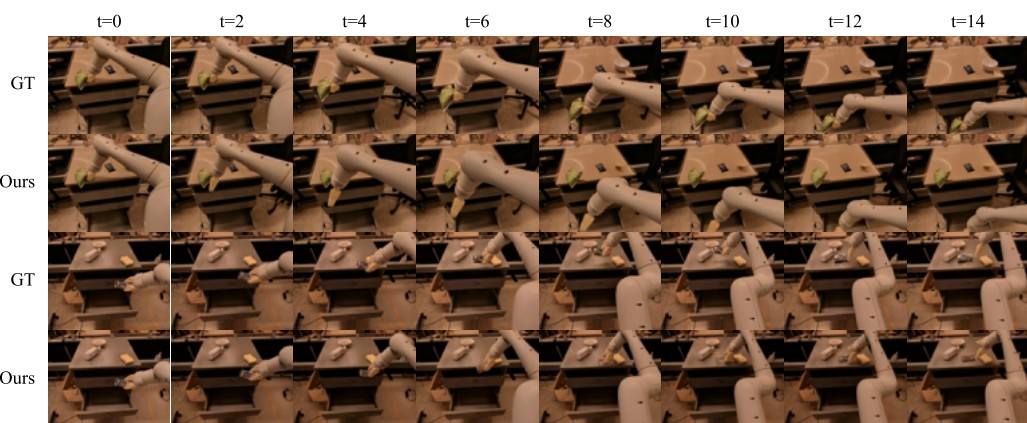

Figure 9: Failure cases for RT-1 dataset

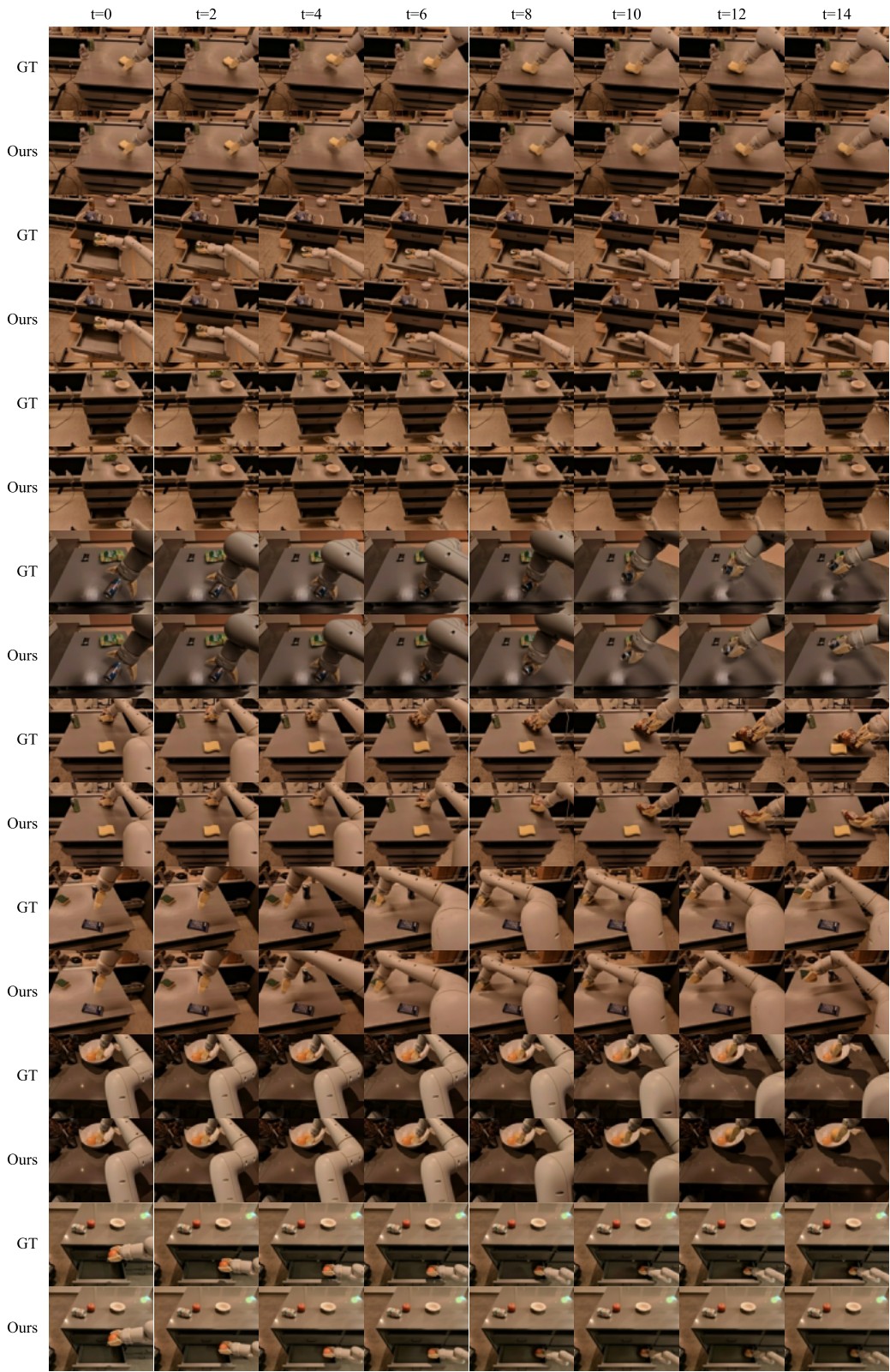

Figure 10: Comparison between ground truth and generated videos by Vid2World in RT-1 Environment. The first frame is provided as context.

## D.2 GENERATION RESULTS OF CS:GO

We provide generation results of Vid2World compared to baseline methods (DIAMOND (Alonso et al., 2024)) in the CS:GO environment. We observe several interesting phenomenon, demonstrating the characteristics, both in strength and in limitations, of our model. We provide the discussion below.

**Error Accumulation.** A common challenge for autoregressive models (for example, the baseline model DIAMOND) in multi-step prediction is performance degradation due to error accumulation, which is especially pronounced when consecutive frames exhibit large variation. In Figure 11, we compare the qualitative predictions of Vid2World and DIAMOND under rapid viewpoint changes. By contrast, DIAMOND's frames become progressively blurred; Vid2World maintains sharpness and closely follows the ground truth trajectory.

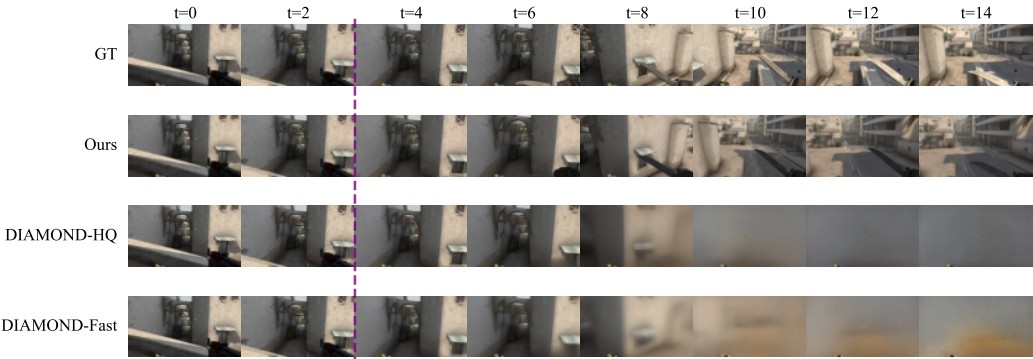

Figure 11: Error Accumulation in CS:GO. While DIAMOND's fidelity degrades significantly during rollout, Vid2World maintains high-quality generation with strong physical accuracy.

**Action Alignment.** The reliability of a world model, to a large extent, depends on how well its predictions align with the input actions. As shown in Figure 12, Vid2World accurately reflects the *aim-down-sights* action in its predicted video, whereas DIAMOND fails to manifest this action.

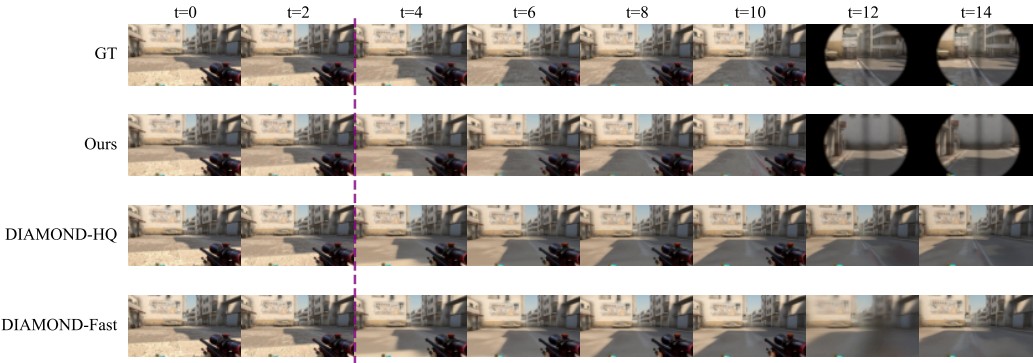

Figure 12: Action alignment in CS:GO. Vid2World truthfully reflects the *aim-down-sights* action in its predicted video, while DIAMOND fails to follow the action.

**Failure Cases.** Despite substantially reducing the accumulated error and preserving action alignment, Vid2World still encounters failure cases, as demonstrated in Figure 13. In this figure, neither Vid2World nor DIAMOND matches the ground truth. Although the model's capability is one important factor leading to failure, the environment's randomness, in this case, the place for the player's respawn, also adds to the difficulty.

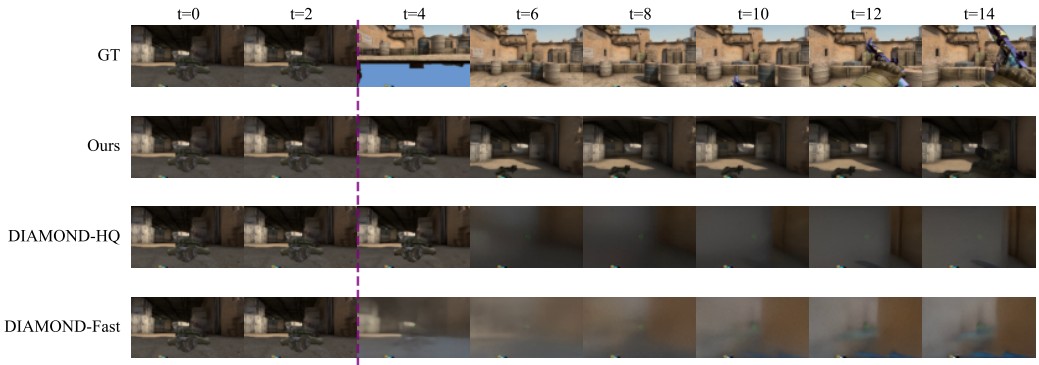

Figure 13: Failure Cases in CS:GO environments.

**Action influence on generated sequence.** For world models, it is important to do so-called *counterfactual reasoning* with the current action, instead of predicting trends based solely on past observations. In Figure 14, we showcase the capability of our model to perform generation based on action sequences. All trajectories start from the same observation, but lead to completely different generated frame sequences due to different action sequences.

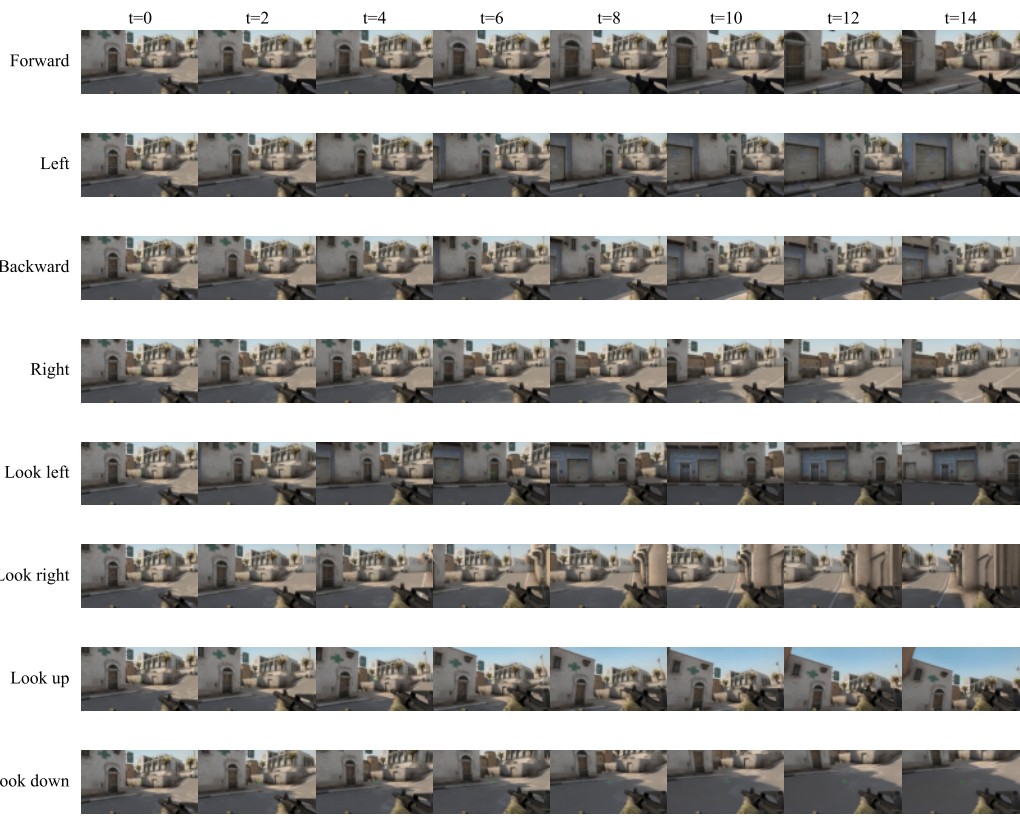

Figure 14: Effect of different actions on generated videos in CS:GO for Vid2World. Trajectories start with the same initial observation, diverging drastically as a result of different action sequences.

## D.3 GENERATION RESULTS FOR OPEN-WORLD NAVIGATION

We provide generation results of Vid2World in the open-world navigation video prediction task, as shown in Figure 15.

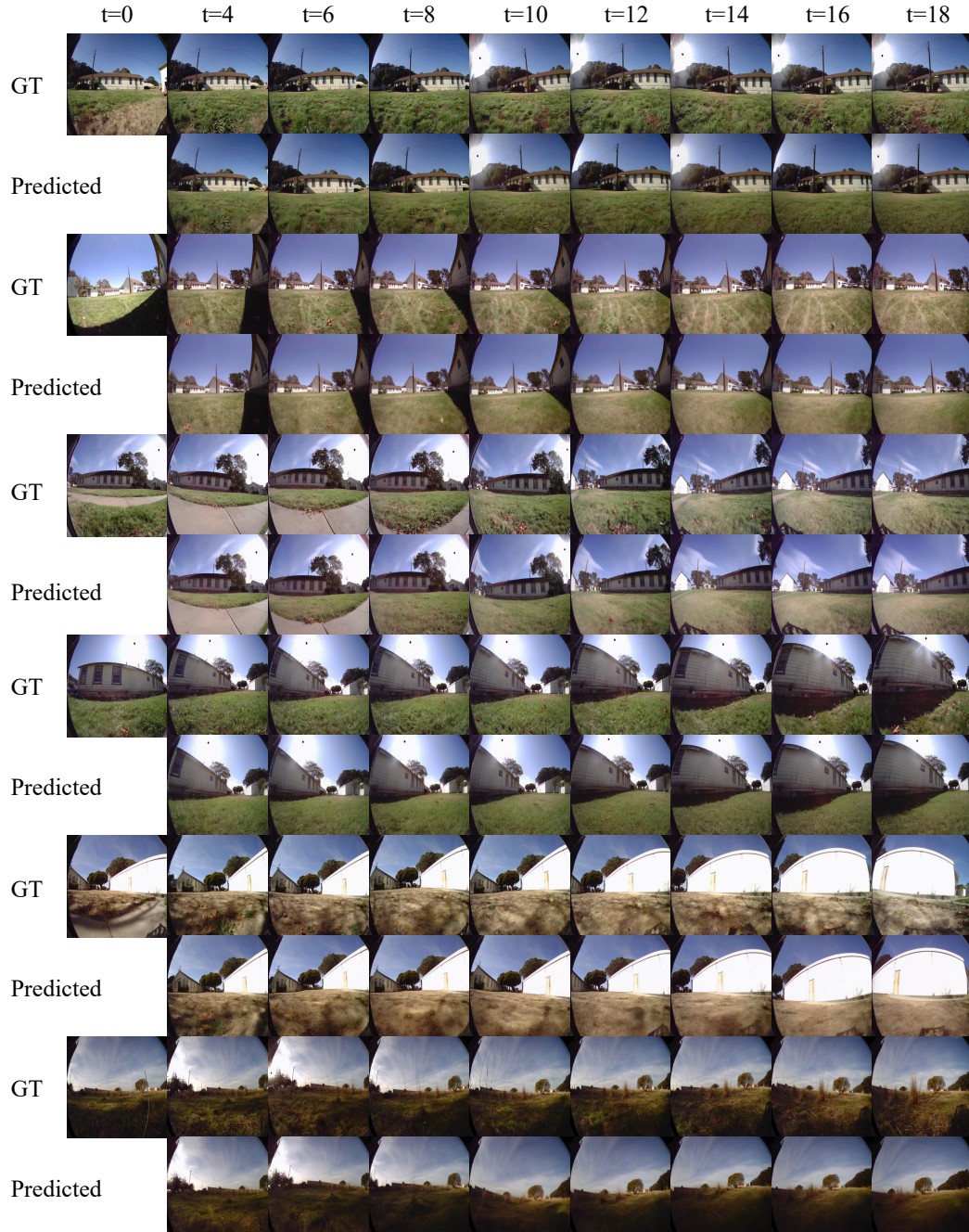

Figure 15: Comparison between ground truth and generated videos by Vid2World in RECON Environment. The first four frames are provided as context.

## D.4 GENERATION RESULTS OF REAL2SIM POLICY EVALUATION

We provide generation results of Vid2World in the Real2Sim Policy Evaluation task, as shown in Figure 16.

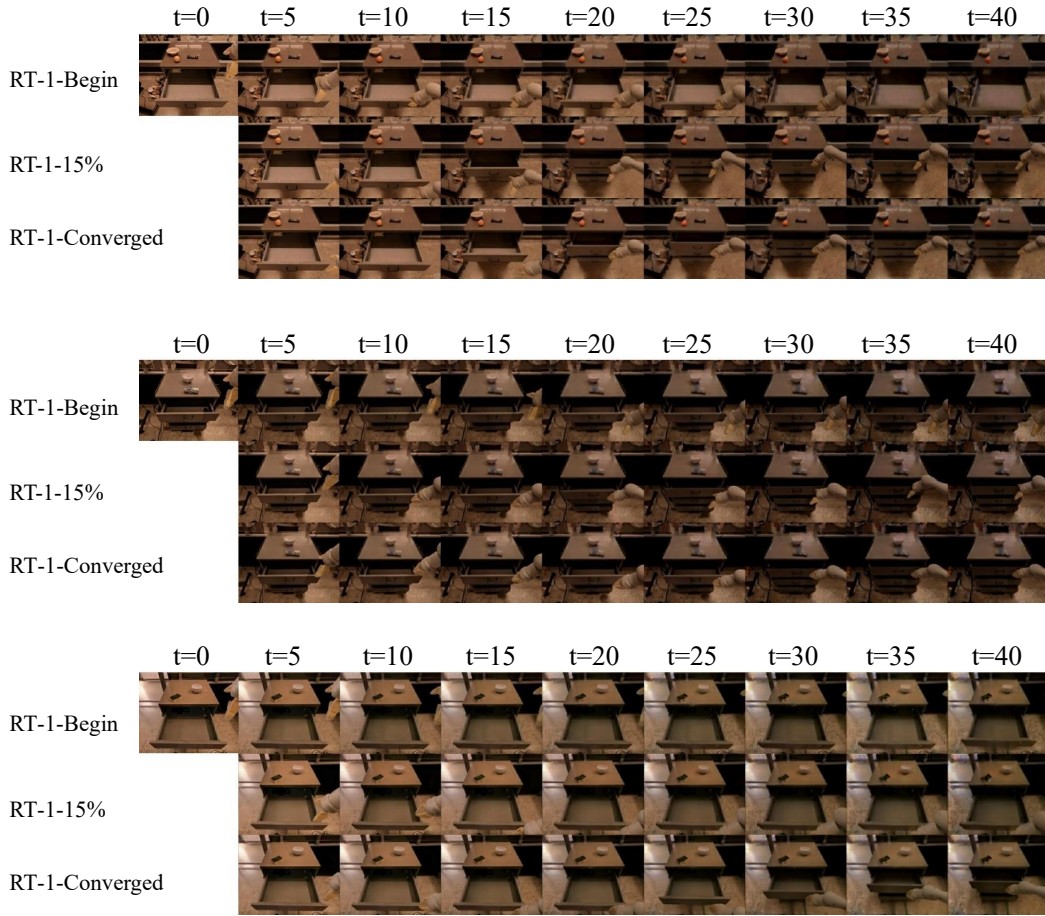

Figure 16: Generation Results for Vid2World in Real2Sim Policy Evaluation experiments.

### D.5    IMPACT OF CAUSAL ACTION GUIDANCE SCALE ON GENERATION QUALITY

To further validate the efficacy of our proposed *Causal Action Guidance* mechanism, we analyze the impact of the guidance scale $\lambda$ on generation quality using 3D Game Simulation.

**Setup.**    We randomly subsample a subset of 50 trajectories from the validation set in the CS:GO environment, and evaluate Vid2World's performance across a range of scales $\lambda \in [1.0, 1.5, 2.0, 2.5, 3.0, 3.5, 4.0]$, reporting the following four metrics: PSNR, SSIM, LPIPS, and DreamSim. Following the setup in Section 5.2, the video prediction metrics are calculated on the predicted frames, conditioned on four initial history frames given as ground truth.

**Results.**    As shown in Figure 8, we observe a consistent trend of improvement followed by degradation with respect to increasing guidance scale $\lambda$ in the generation quality across all evaluated metrics. This is aligned with our intuition of probability steering: While absence or insufficiency of guidance scale results in suboptimal metric scores due to poor action adherence, overshoot of guidance scale leads to over-sharpened distributions and visual artifacts. Through varying causal action guidance scale $\lambda$, our mechanism offer the test-time flexibility of trading off *action alignment* with *visual fidelity*, leading to improved world modeling capabilities. These results, taken together with ablation studies in Section 5.4 (which confirm the necessity of guidance) and theoretical justifications in Appendix A.4, provide a holistic validation of our method, spanning theoretical arguments, intuitive explanations and empirical validations.

### D.6    LONG-HORIZON ROLLOUT AND ZERO-SHOT GENERALIZATION

To validate Vid2World's robustness to *temporal extrapolation*, we conduct a long-horizon rollout experiment in the CS:GO environment. Initialized with 9 ground truth frames, the model performs autoregressive generation conditioned on a sliding window of the 9 most recent observations. The action sequence is sampled uniformly from $\{\texttt{W}, \texttt{A}, \texttt{S}, \texttt{D}\}$, with each action held constant for 10 consecutive frames to induce significant movement. We extend the rollout to a total of 100 generated frames, significantly exceeding the training horizon of 16. As illustrated in Figure 17, while artifacts such as softened textures and drift in spatial layout occur as the rollout horizon increases, the generation maintains relatively high fidelity and physical realism, demonstrating strong robustness against error accumulation.

In addition to in-domain temporal extrapolation, we further evaluate the model's *zero-shot generalization* on a completely out-of-distribution (OOD) dataset: the tactical shooter game *Valorant* (Riot Games, 2020). Crucially, during the Vid2World training process, the model was exposed *exclusively to CS:GO data*. Therefore, any cross-domain generalization observed in this setting stems entirely from the robust *visual priors* preserved from the pre-trained video diffusion backbone during our transformation process. We perform autoregressive rollouts up to 50 frames. As shown in Figure 18, although the visual quality degrades much faster compared to the in-domain setting, the model surprisingly retains rudimentary capabilities of *temporal consistency* and *action responsiveness*. This zero-shot generalization experiment further validates that Vid2World enables interactivity while preserving and channeling the video diffusion model's inherent generalization capabilities.

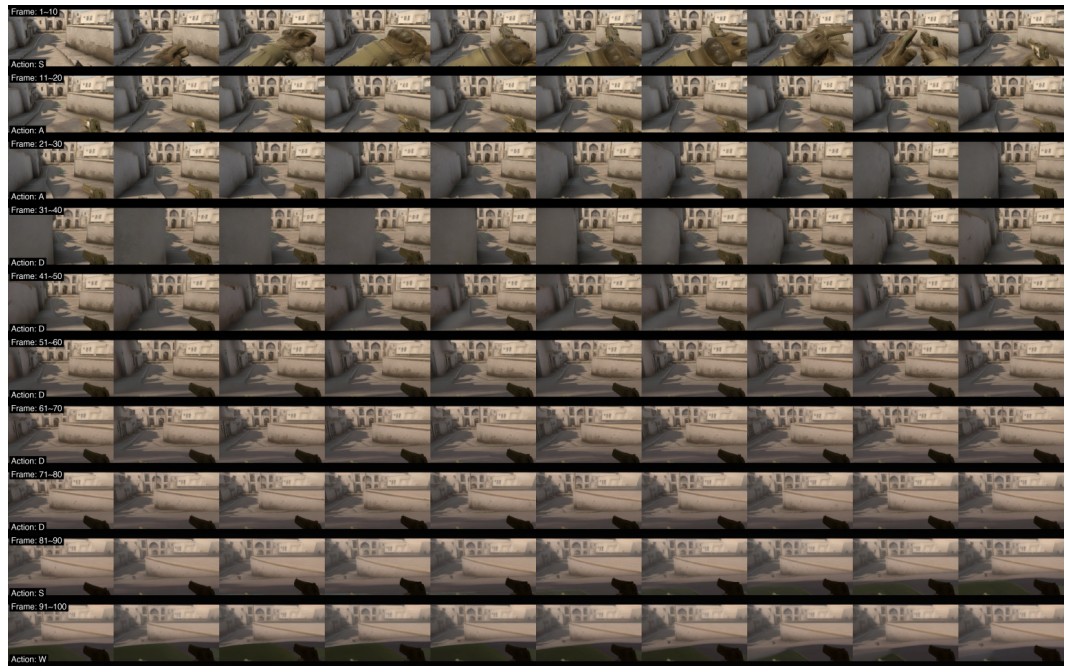

(a) Rollout Sample 1

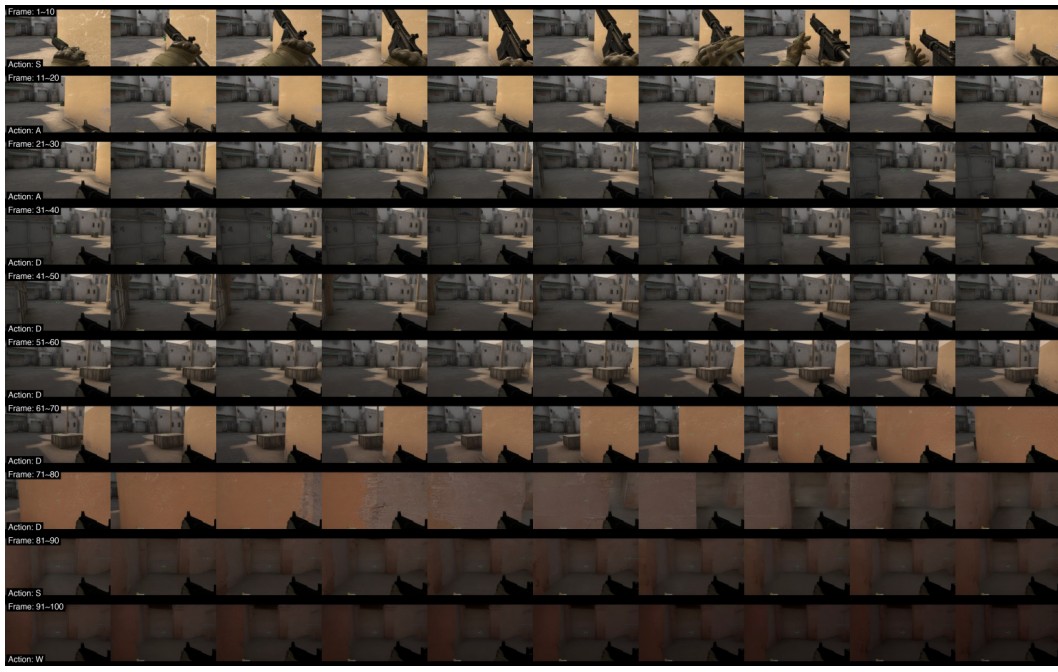

(b) Rollout Sample 2

Figure 17: **Temporal Extrapolation via Long-Horizon Rollout (Part I)**. Initialized with 9 history frames, Vid2World autoregressively generates 100 frames conditioned on random action sequences in the CS:GO environment (over 6x the training horizon). Frame indices and corresponding actions are annotated at the start of each row. *Zoom in for details. Continued on next page.*

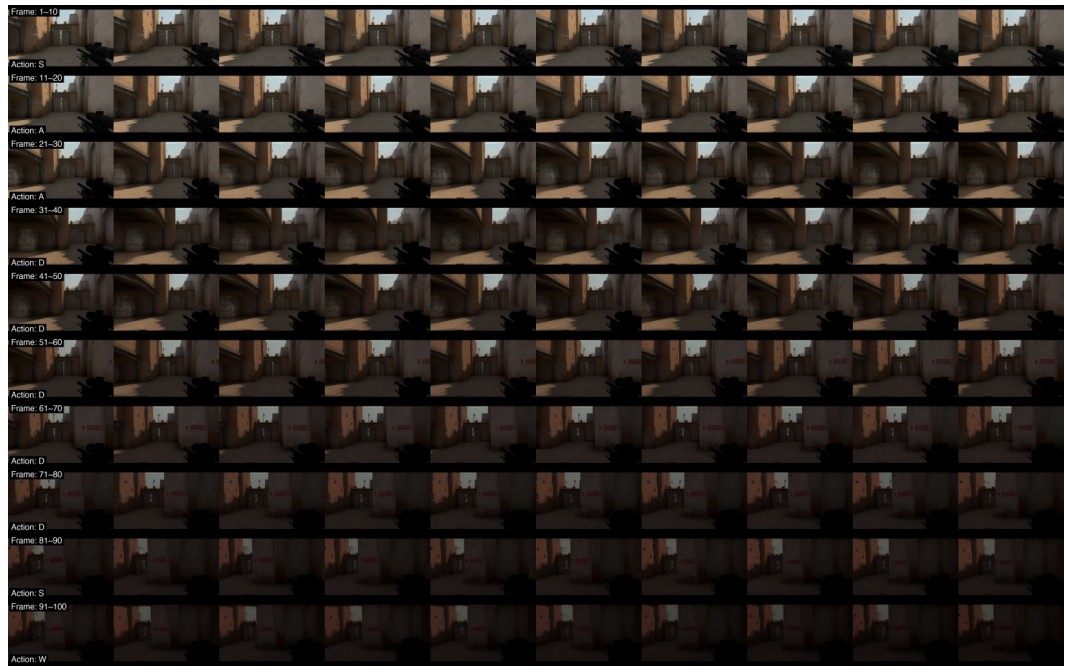

(c) Rollout Sample 3

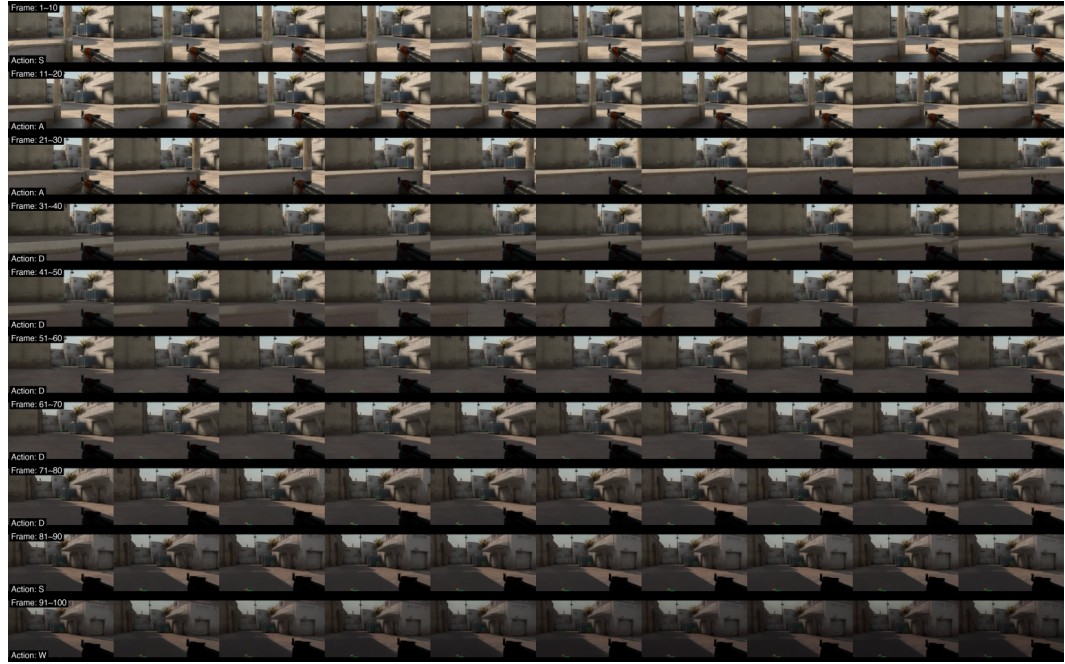

(d) Rollout Sample 4

Figure 17: **Temporal Extrapolation via Long-Horizon Rollout (Part II)**. Initialized with 9 history frames, Vid2World autoregressively generates 100 frames conditioned on random action sequences in the CS:GO environment (over 6x the training horizon). Frame indices and corresponding actions are annotated at the start of each row. *Zoom in for details. Continued on next page.*

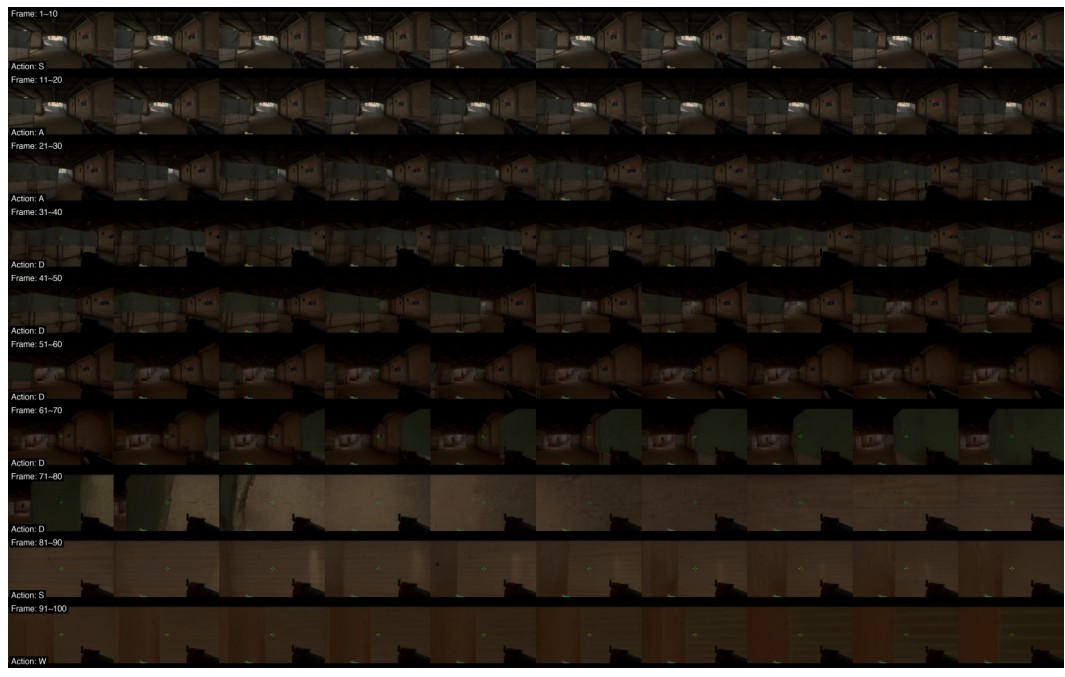

(e) Rollout Sample 5

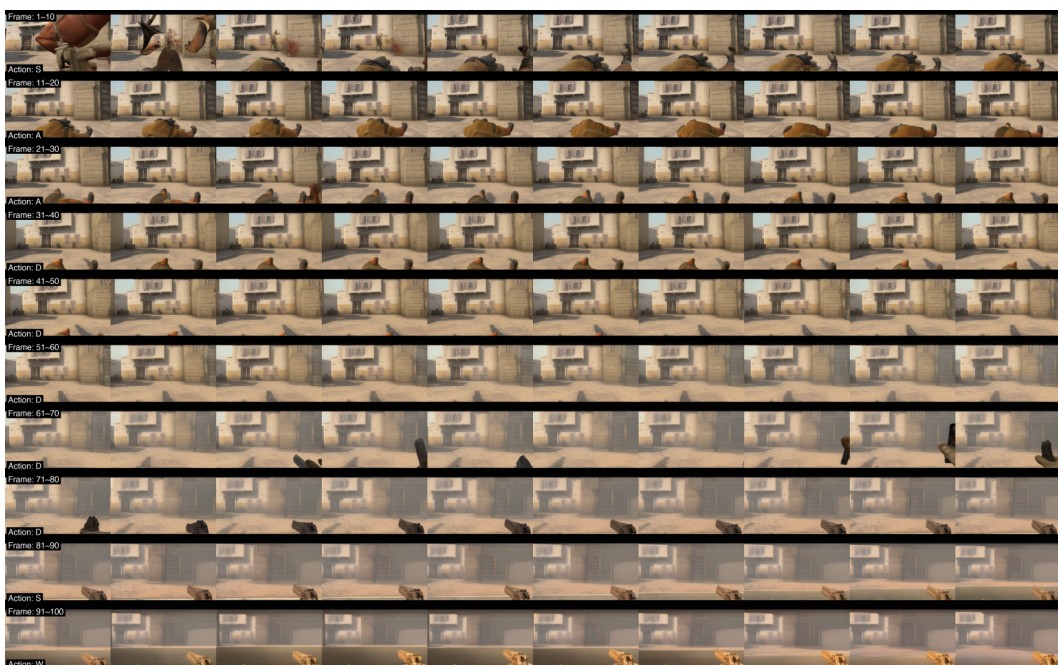

(f) Rollout Sample 6

Figure 17: **Temporal Extrapolation via Long-Horizon Rollout (Part III)**. Initialized with 9 history frames, Vid2World autoregressively generates 100 frames conditioned on random action sequences in the CS:GO environment (over 6x the training horizon). Frame indices and corresponding actions are annotated at the start of each row. *Zoom in for details.*

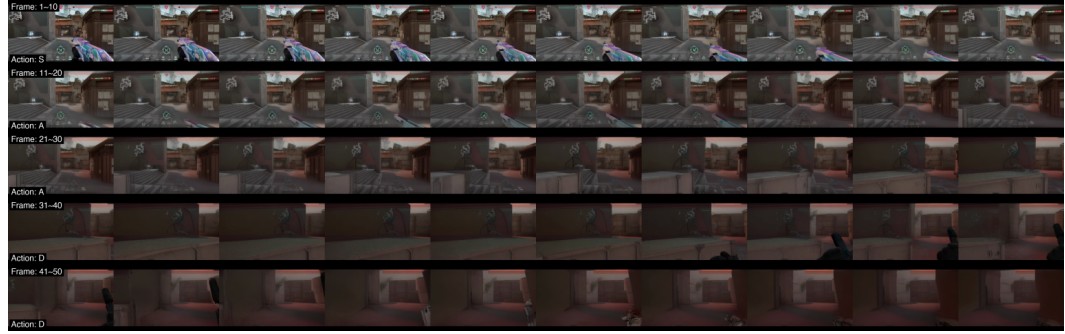

(a) Zero-Shot Rollout Sample 1

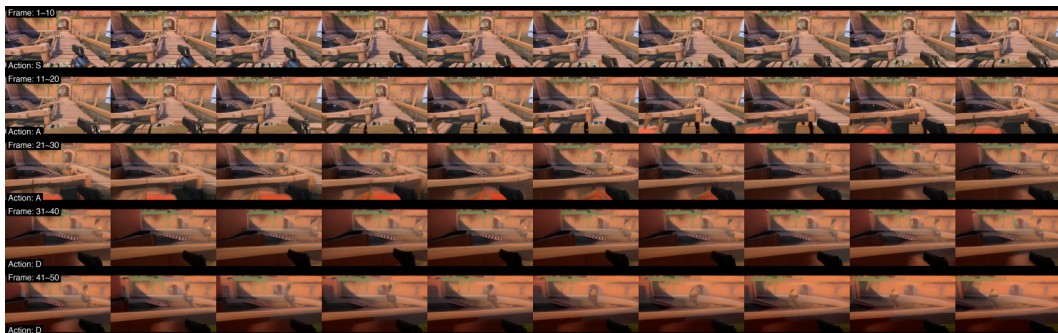

(b) Zero-Shot Rollout Sample 2. *Notice the movement of the railing from Frame 21 to 30, moving right due to the character moving left, showcasing action responsiveness.*

Figure 18: **Zero-Shot Generalization on Valorant (OOD)**. Despite being trained exclusively on CS:GO, Vid2World autoregressively generates 50-frame rollouts in the unseen game *Valorant* while suprisingly retaining rudimentary temporal consistency and action responsiveness, demonstrating the robust visual priors preserved from the pre-trained backbone. *Zoom in for details.*

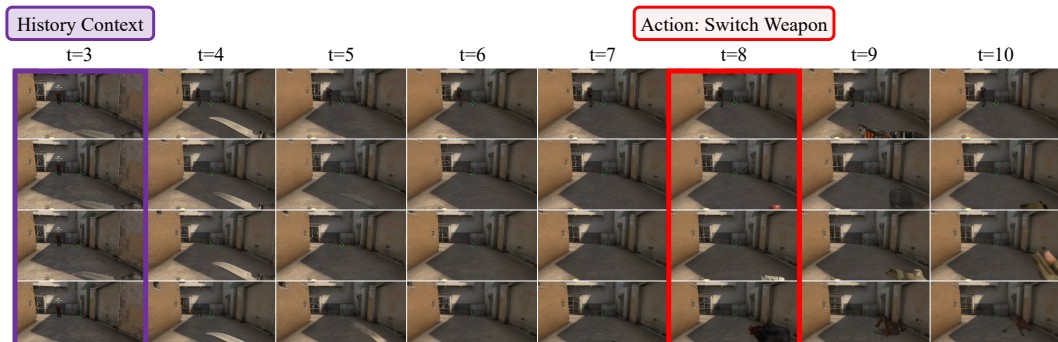

Figure 19: **Capturing Environment Stochasticity.** Conditioned on 4 history frames, Vid2World predict the future in auto-regressive rollout. On timestep t=8, given the "switch weapon" action, the model generates the previously unseen weapon. Since the specific weapon is unobserved in the history, the model produces **diverse plausible outcomes** with various shapes and colors across samples, effectively capturing the stochastic nature of the environment. *Zoom in for details.*

## D.7 CAPTURING ENVIRONMENT STOCHASTICITY

To further investigate whether Vid2World captures the complex stochastic nature of the underlying dynamics, rather than merely predicting a single-modal distribution, we conduct a qualitative analysis in the domain of CS:GO. Specifically, we fix the provided history (4 frames) and the future action

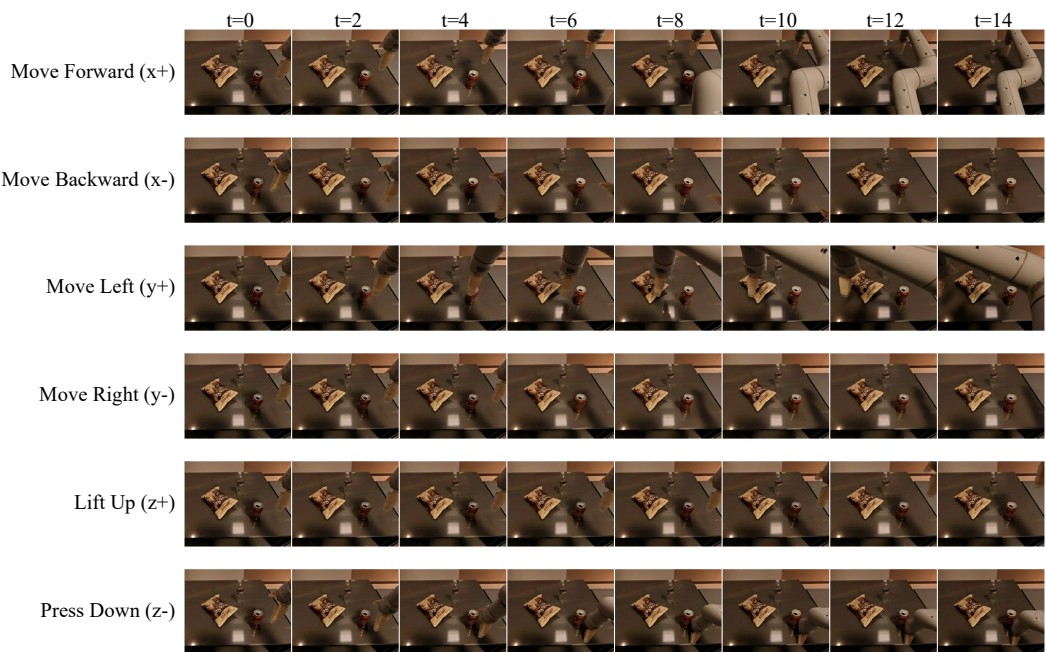

Figure 20: **Following Action Semantics.** Conditioned on the same initial frame, Vid2World predicts diverging futures under different action conditions, faithfully following action semantics.

sequence (triggering "switch weapon" at $t = 8$), while sampling multiple trajectories with our Vid2World model. As illustrated in Figure 19, the model produces diverse plausible outcomes of the potential weapon, varying in shapes and colors. Since the secondary weapon is not visible in the conditioning context, the model is destined to draw the weapon from a multi-modal distribution. This diversity confirms that our model effectively learns the joint distribution of the data, capturing the inherent stochasticity of the environment rather than collapsing to a single mode.

### D.8    FOLLOWING ACTION SEMANTICS

To validate Vid2World genuinely follows action semantics, rather than predicting observatory trends based solely on history frames, we start from the same initial frame, sampling different action sequences for video prediction. In addition to validating this capability in the CS:GO environment (as shown in Figure 14), we conduct the experiment with similar spirits using the RT-1 environment.

**Setup.**    We randomly sample an initial frame from the validation set, using it as the history context. Since in the RT-1 environment, the action space represents the end effector position of the robot arm, we therefore condition generation on six canonical action sequences: *Move Forward*, *Move Backward*, *Move Left*, *Move Right*, *Lift Up*, and *Press Down*, corresponding to translations of the end effector along the x, y and z axes. Using the same configurations as Section 5.1, we autoregressively rollout trajectories until a total number of 16 frames is reached.

**Results.**    As shown in Figure 20, Vid2World's predictions exhibit clear, directionally coherent end-effector motion that aligns with the semantics of the conditioned action sequence. Notably, the model correctly anticipates challenging phenomena such as *end effector exiting the camera view* (for Move Backward and Move Right) and *physical contact with the table* (for Press Down). Taken together with the action influence on generated trajectory experiment in Figure 14 and the Real2Sim Policy Evaluation Experiment in Section 5.1, these results validate Vid2World's capability of generating future predictions that faithfully reflect the semantics of actions.

Table 4: **Quantitative Results of Interactive Metrics on CS:GO environment.** After generating video predictions based on ground truth actions as well as random actions, we calculate the normalized delta metrics $\Delta_N - \mathcal{M}$. Best results are shown in **bold**.

| Model | $\Delta_N$-**FVD** ↓ | $\Delta_N$-**FID** ↓ | $\Delta_N$-**SSIM** ↑ | $\Delta_N$-**LPIPS** ↓ | $\Delta_N$-**PSNR** ↑ | $\Delta_N$-**Dreamsim** ↓ |
|---|---|---|---|---|---|---|
| Diamond-Fast | -48.43 | -5.01 | 44.84 | -20.95 | **52.94** | -44.16 |
| Diamond-HQ | -55.85 | -14.76 | **47.04** | **-25.55** | 48.78 | **-53.35** |
| Vid2World | **-77.06** | **-29.44** | 22.40 | -20.32 | 17.60 | -41.81 |

## D.9 QUANTITATIVE RESULTS ON INTERACTIVE METRICS

To quantitatively validate Vid2World's capability of performing counterfactual reasoning based on the action sequence, following Bruce et al. (2024), we evaluate this "action semantics following" capability by comparing the generation quality conditioned on ground truth and conditioned on random action input.

**Metrics.** We generalize the $\Delta_t$-PSNR proposed by Bruce et al. (2024) to video generation metrics, introducing *normalized delta metrics* ($\Delta_N$-$\mathcal{M}$) for corresponding video generation metric $\mathcal{M}$. Specifically, given the ground truth video sequence $\mathbf{x}_{0:T}$, generated sequence conditioned on ground truth action $\tilde{\mathbf{x}}_{0:T}$ and generated sequence conditioned on random actions $\tilde{\mathbf{x}}'_{0:T}$, as well as a video generation metric $\mathcal{M} : \mathbb{R}^{T \times d} \times \mathbb{R}^{T \times d} \to \mathbb{R}$, the metric is defined as:

$$\Delta_N\text{-}\mathcal{M}(\mathbf{x}_{0:T}, \tilde{\mathbf{x}}_{0:T}, \tilde{\mathbf{x}}'_{0:T}) := \frac{\mathcal{M}(\mathbf{x}_{0:T}, \tilde{\mathbf{x}}_{0:T}) - \mathcal{M}(\mathbf{x}_{0:T}, \tilde{\mathbf{x}}'_{0:T})}{\mathcal{M}(\mathbf{x}_{0:T}, \tilde{\mathbf{x}}'_{0:T})}.$$

Intuitively, this metric measures how much the video generations differ when conditioned on ground truth action sequences compared to actions sampled from a random distribution, normalized by the scale of the metric $\mathcal{M}$. For metric $\mathcal{M}$ such that higher is better, $\Delta_N$-$\mathcal{M}$ also satisfies higher is better, as vice versa.

**Setup.** We evaluate our model's action following capability under the CS:GO environment. Following Section 5.2, we initialize with 4 history frames and corresponding ground truth actions, and predict on the validation set (500 trajectories) autoregressively until a total frame number of 16 is reached. For randomly sampled actions, we uniformly sample from all valid keyboard inputs. We report the calculated Delta Normalized Metrics $\Delta_N$-$\mathcal{M}$, are shown in Table 4.

**Results and Discussion.** As shown in Table 4, Vid2World outperforms baseline methods in $\Delta_N$-FVD and $\Delta_N$-FID, showing Vid2World's capability of following action semantics. However, we notice clear trends of generation quality degradation when sampling DIAMOND with random actions, whereas our model does not. Hence, we raise the concern of "hacking" the metric, where a model can acquire high quantitative performance on these delta normalized metrics by generating low-quality videos when conditioned on random action distribution. This inability to distinguish *action following capabilities* with *generation quality degrade under ood action distributions*, contributes at least partially to Vid2World's limited performance on these metrics. We anticipate futher work to come up with better evaluation metrics for measuring interactivity capabilities.

## D.10 THE ROLE OF LARGE-SCALE PRETRAINING

In this section, we focus on a key question, highlighting the role of large-scale pretraining on this transfer method:

**RQ** : Is the model's success attributed to channeling visual priors from the pre-trained video diffusion model or merely the novel architectural changes?

To ablate the role of large-scale video pretraining, we randomly initialize the model parameters, following the exact architectural choice as Vid2World, and train the entire model from scratch based solely on the RT-1 dataset, using 30k gradient steps for fair comparison with other ablated models.

Table 5: **Ablation study on the role of video pretraining**: To validate Vid2World truly transfers priors from the pretrained video diffusion model, we train an additional model from scratch in the RT-1 environment, maintaining the exact architecture as Vid2World but randomly initializing the parameters. Best results in **bold**, worst in *italics*.

| Model | WT | AG | FVD ↓ | FID ↓ | SSIM ↑ | LPIPS ↓ | PSNR ↑ |
|---|---|---|---|---|---|---|---|
| Vid2World | Shift | | 29.9 | 7.85 | 0.799 | 0.185 | 21.5 |
| Vid2World | Masked | | 29.4 | 7.07 | 0.824 | 0.169 | 22.9 |
| Vid2World | Extrapolative | | 28.6 | 7.52 | 0.832 | 0.162 | 23.4 |
| Vid2World | Masked | ✓ | 25.8 | 6.84 | **0.840** | **0.159** | **23.9** |
| Vid2World | Extrapolative | ✓ | **22.4** | **6.16** | 0.839 | **0.159** | **23.9** |
| From Scratch | Does not Apply | ✓ | *1768.8* | *469.4* | *0.065* | *0.8177* | *8.3* |

As shown in Table 5, even with the identical architectural and algorithmic designs, training the world model from scratch leads to significant performance drop compared to all other Vid2World models that utilize the pre-existing video diffusion model across all metrics. This indicates that during Vid2World, the generation priors encapsulated in video diffusion model due to large-scale video pretraining truly transfers to the transformed interactive world model.

## D.11 EMPIRICAL VALIDATION OF CAUSAL WEIGHT TRANSFER

To validate the functioning our causal weight transfer mechanism, we conduct additional experiment, investigating whether representation similarity is perserved during transformation. Specifically, we focus on the following research question:

**RQ**: Does Weight Transfer preserve representations close to the original video diffusion model?

**Setup.** To measure the similarity of the transformed feature of our transformed world model and the original pretrained video diffusion model, we utilize the metric of **Cosine Similarity**. Cosine Similarity measures the similarity between two non-zero vectors in a multi-dimensional space by calculating the cosine of the angle between them, resulting in values in $[-1, 1]$. The higher the score, the stronger the correlation. We randomly sample 50 trajectories in the validation set of RT-1, passing the clean video sequence as input. Actions are dropped out for world models, and noise scale is set to zero across of frames. We measure the mean cosine similarity between latent representations extracted after the first convolutional layer of our transformed world model and the original pretrained video diffusion model. For all models, we employ training on 30k gradient steps, following ablation setup (Section 5.4). The results are shown in Table 6.

Table 6: **Cosine similarity between features of transformed world models and original video diffusion model.** Taken the same video sequence as input, we measure the mean cosine similar of features taken after the first convolution layer of both models. Values are scaled by 100 for better visualization. Best results are shown in **bold**.

| Model | WT | AG | Cosine Similarity ↑ |
|---|---|---|---|
| Vid2World | Shift | ✓ | 71.04 |
| Vid2World | Masked | ✓ | 71.09 |
| Vid2World | Extrapolative | ✓ | **71.13** |

**Results.** As shown in Table 6, all three weight transfer methods of Vid2World achieve high cosine similarity scores (>0.7), demonstrating that Vid2World preserves the representational structure of the original video diffusion model. The extrapolative variant achieves the highest similarity, providing empirical evidence of the functioning of linear extrapolation, effectively repurposing generative video priors for interactive world modeling.

## E    COMPUTATION COSTS

In this section, we provide a detailed breakdown of the computational costs required for Vid2World. As a first step toward repurposing large-scale video diffusion models into interactive world models, our primary focus has been on establishing visual fidelity and physical realism rather than optimizing for inference latency.

**Training Computation Costs.**    We conduct training on 4 Nvidia 40GB A100 GPUs, using the hyperparameters provided in Table 3. The configured training completes 100k gradient steps in 7 days, utilizing a memory footprint of 37GB of GPU VRAM.

**Inference Computation Costs.**    Since our model is capable of autoregressive generation conditioned on a flexible number of history frames and ddim sample steps, our inference speed depends on the configuration of such hyperparameters. We provide a realistic setup below: On a single NVIDIA A100 (40GB) GPU, the model is configured with a context window of 10 history frames and utilizes 50-step DDIM sampling with causal action guidance scale greater than 1. Under these settings, with a batch size of 2, the model's inference latency is approximately 20.0 seconds per generated frame, with memory footprint of approximately 16 GB of GPU VRAM.

**Potential Speedup Methods.**    While the current inference latency reflects the heavy computational cost of high-fidelity diffusion sampling, we identify several promising avenues of speeding up inference: (a.) Decreasing the number of sampling steps; (b.) Conditioning on shorter history; (c.) Software optimizations for instance JIT compiling the model; (d.) Implementing (rolling) KV-Cache for autoregressive generation (Yin et al., 2024; Huang et al., 2025). We anticipate further work that builds upon these insights, achieving real-time action-conditioned video generation.

## F    THE USE OF LARGE LANGUAGE MODELS (LLMS)

Large Language Models (LLMs) were employed with the single purpose of polishing the writing of this manuscript. Their use was strictly limited to enhancing language quality, improving readability, and ensuring clarity throughout the paper. In particular, LLM assisted in rephrasing sentences, checking grammar, and identifying typos.

It is important to note that LLMs were not involved in the conception and realization of research ideas, the design of methodologies, or the execution of experiments. All research contributions—including conceptual development, methodological choices, and data analysis—were carried out independently by the authors. LLMs' role was strictly confined to linguistic refinement, without influencing the scientific substance of the work.

The authors take full responsibility for the entire content of the manuscript, including text that was modified using LLM-based assistance. We have taken great care to ensure our use of LLMs adheres to ethical guidelines and does not give rise to plagiarism or scientific misconduct.

