# OpenReview forum: "Vid2World: Crafting Video Diffusion Models to Interactive World Models"
_ICLR.cc/2026/Conference — ICLR 2026 Poster_

### Official Review · Reviewer_Yc6M · 2025-10-31

**Soundness:** 4
**Presentation:** 3
**Contribution:** 3
**Rating:** 4
**Confidence:** 4

**Summary:**

This paper proposes a straightforward yet effective method for adapting a pre-trained video diffusion model into an action-conditioned world model. The core contributions are two-fold: 1) "Video Diffusion Causalization," which enforces temporal causality through a causal masked attention layer and a temporal convolution layer with a novel "extrapolative weight transfer" technique, and 2) "Causal Action Injection," which integrates action-conditioning via causal guidance with dropouts. The authors conduct extensive experiments across diverse domains, including robotic manipulation (RT-1), game simulation (CS:GO), and real-world navigation (NWM), demonstrating that their proposed model achieves higher fidelity in video prediction compared to existing world models.

**Strengths:**

- Simplicity and Effectiveness: The primary strength of this work lies in its simple and elegant approach. The proposed modifications to transform a standard video diffusion model into a world model are minimal, yet they prove to be highly effective across multiple challenging domains.
- Broad Applicability: The method's effectiveness is convincingly demonstrated on a variety of environments, from simulated robotics and games to real-world navigation. This suggests the proposed techniques are general and widely applicable.
- High-Fidelity Generation: The paper provides strong quantitative and qualitative results, showing that the model can generate high-fidelity, fine-grained, and action-conditioned video sequences that are visually superior to prior work.

**Weaknesses:**

- Insufficient Motivation for Extrapolative Weight Transfer: While the paper introduces "extrapolative weight transfer" as a key component, its theoretical and empirical motivation remains underdeveloped. The appendix (A.1) presents a simple counter-example involving a linearly increasing kernel, but this theoretical argument is not supported by empirical evidence. To strengthen this claim, the authors should provide experiments that verify the existence and prevalence of such temporal patterns in the learned kernels of baseline models, thereby justifying the need for their proposed solution.

- Lack of Experiments Validating Utility as a World Model: The paper claims to build a "world model," but the experiments primarily focus on video prediction fidelity (e.g., FVD, FID, SSIM). A true world model should be useful for downstream tasks like planning and control. The evaluation is missing crucial experiments, such as: (1) Reinforcement Learning Tasks: Can an RL agent be successfully trained using the learned model as the environment simulator? Demonstrating successful policy learning would provide compelling evidence of the model's utility and temporal consistency. If this is not feasible, a discussion of the model's limitations for such tasks is warranted. (2) Quantitative Interactivity Metrics: The current evaluation lacks metrics that specifically measure the model's fine-grained controllability and causal soundness in response to actions.

- Ambiguity in the Role of Large-Scale Pre-training: The experimental setup creates a potential ambiguity. The proposed model leverages an internet-scale pre-trained video model, whereas the robotics and gaming experiments are conducted on domain-specific datasets. It is unclear which components of the model's success are attributable to the novel architectural changes versus the powerful, pre-trained backbone. An ablation study clarifying the benefits of internet-scale pre-training for domain-specific tasks like RT-1 or CS:GO would be highly valuable.

- Incomplete Reporting of Metrics: For the real-world navigation (NWM) experiments, key fidelity metrics such as FVD, FID, and SSIM appear to be missing from the results tables. Including these would allow for a more direct and fair comparison with other models and experiments within the paper.

**Questions:**

- Figure 6: This figure presents a crucial comparison of auto-regressive generation capabilities. However, the motivation for why this specific comparison is necessary and what insights it provides is detailed in Appendix C.6 but absent from the main text. Briefly explaining this context in the main paper would significantly improve the figure's clarity and impact.

---

> ### Author Response · Authors · 2025-11-24
> **Response to Reviewer Yc6M [1/5]**
>
> We sincerely thank the reviewer for recognizing the simplicity, effectiveness and broad applicability of our work. We appreciate the reviewer's constructive feedback and would love to provide additional clarification and empirical evidence.
>
> > **W1**: Insufficient Motivation of Extrapolative Weight Transfer: The appendix(A.1) presents a simple counter-example involving a linearly increasing kernel, but this theoretical argument is not supported by empirical evidence.
>
> We thank the reviewer for this thoughtful comment. However, we'd like to firstly point out a misunderstanding regarding the reviewer's understanding of Appendix A.1. **Rather than assuming a _"linearly increasing kernel"_, we build our argument on assuming a  _"linearly changing input sequence"_.**
>
> This is not a strong assumption, commonly adopted in multiple domains related to sequential modeling, including dynamical systems [1,2], time series [3,4] and stochastic optimal control [5,6]. Below we validate the design of Extrapolative Weight Transfer, both theoretically and empirically.
>
> **Theoretically**, our extrapolative weight transfer mechanism exploits the local linearity of the input sequence. Since for any $L$-smooth function $f(t)$, we can bound the error term between the original function and the linear approximation $g(t):= f(t_0)+ \langle \nabla f(t_0), t-t_0 \rangle$ as a function of sample interval, i.e. $|f(t)-g(t)|\leq \frac{L}{2}||t-t_0||^2$, this linearity assumption does not deviate significantly from the true function, particularly in high-frame-rate videos where local temporal consistency ensures smoothness.  In such settings, the latent space often exhibits a linear evolution over time (think about representing the position $x$ of an object moving at constant velocity, where $x = vt$).
>
> **Empirically**, we validate the following question: _Does Weight Transfer preserve representations close to the original video diffusion model?_ To answer this, we measure the mean cosine similarity between latent representations, extracted after the first convolutional layer of our transformed world model and the original pretrained video diffusion model, using identical video sequences from the RT-1 environment. Results are summarized below (values scaled by 100 for clarity; best result in **bold**):
>
> | Model                            | WT             | AG           | Cosine Similarity $\uparrow$ |
> | -------------------------------- | -------------- | ------------ | ---------------- |
> | Vid2World                        | Shift          |          $\checkmark$    | 71.04             |
> | Vid2World                        | Masked         | $\checkmark$ | 71.09            |
> | Vid2World                        | Extrapolative  | $\checkmark$ | **71.13**         |
>
> As shown in the table, all three weight transfer methods achieve high cosine similarity scores (>0.7), demonstrating that Vid2World preserves the representational structure of the original video diffusion model. The extrapolative variant achieves the highest similarity, providing empirical evidence of the functioning of linear extrapolation, effectively repurposing generative video priors for interactive world modeling. Further details are included in **Appendix D.11** of our revised manuscript.

---

> ### Author Response · Authors · 2025-11-24
> **Response to Reviewer Yc6M [2/5]**
>
> >**W2.1**: Lack of Experiments Validating Utility as a World Model: A true world model should be useful for downstream tasks like planning and control. The evaluation is missing crucial experiments, such as Reinforcement Learning.
>
> **TL;DR: While we agree RL evaluation is valuable, it's currently computationally prohibitive for our setup. We instead validate our model in downstream tasks through Real2Sim policy evaluation (Section 5.1), where Vid2World effectively reflects real-world policy performance. We have also added discussion on this limitation involving possible solutions in Appendix A.7.**
>
> We thank the reviewer for their insightful feedback. We agree that the applicability of world models is closely tied to their performance in downstream tasks. That said, conducting the reinforcement learning experiments as suggested presents substantial challenges:
>
> 1. Firstly, Reinforcement learning is widely recognized for its sample inefficiency, which imposes substantial computational demands.
> 2. Moreover, since Vid2World builds on pretrained video diffusion models, which are inherently large and require iterative denoising steps during inference, the evaluation process becomes computationally intensive.
>
> Together, these factors lead to prohibitive costs given our current computational resources.
>
> While world models, especially in applications such as embodied AI, should ideally support reinforcement learning, a majority of current works based on pixel-space prediction omit RL experiments due to forbidding inference costs. This is also the case for several industry-level world models (e.g. Genie 3 [7], Marble [8] and PAN [9]), where RL evaluation is not included.
>
> In our work, we instead demonstrate the model's capacity to support downstream decision-making through Real2Sim Policy Evaluation in **Section 5.1 of the original manuscript**. In this task, the world model is used to predict the success rate of various policies that generate actions in an online manner, conditioned on the world model's predictions. As shown in **Figure 5**, Vid2World reliably captures the performance gap between policies, closely tracking their real-world success rates.
>
> We greatly appreciate the reviewer's feedback on this issue. In response, we have included an extended discussion in **Appendix A.7** of the revised manuscript, acknowledging this limitation and suggesting potential approaches for incorporating reinforcement learning in future work.

---

> ### Author Response · Authors · 2025-11-24
> **Response to Reviewer Yc6M [3/5]**
>
> > **W2.2**: Quantitative Interactivity Metrics: The current evaluation lacks metrics that specifically measure the model's fine-grained controllability and causal soundness in response to actions.
>
> **TL;DR: We extend the $\Delta_t$-PSNR metric used by Genie to video setups, showing Vid2World excels in FVD/FID metrics. However, we identify a potential "metric hacking" issue, where controllability and degraded generation quality in OOD action sequences are indistinguishable, suggesting need for more robust interactivity evaluation methods.**
>
> To validate the controllability (action-following) capabilities, we added an additional quantitative experiment on such interactive metric in the CS:GO environment. Inspired by $\Delta_t$-PSNR in Genie [10], we extend this interactive metric to measuring video metrics, introducing _normalized delta metrics_ ($\Delta\_N\text{-}\mathcal{M}$). Specifically, given the ground truth sequence $\mathbf{x}\_{0:T}$, generated sequence conditioned on ground truth action $\tilde{\mathbf{x}}\_{0:T}$ and generated sequence conditioned on random actions $\tilde{\mathbf{x}}'\_{0:T}$, as well as a video generation metric $\mathcal{M}: \mathbb{R}^{T \times d} \times \mathbb{R}^{T \times d} \rightarrow \mathbb{R}$, the metric is defined as: $$\Delta\_N\text{-}\mathcal{M}(\mathbf{x}\_{0:T}, \tilde{\mathbf{x}}\_{0:T}, \tilde{\mathbf{x}'}\_{0:T}):= \frac{\mathcal{M}(\mathbf{x}\_{0:T}, \tilde{\mathbf{x}}\_{0:T})-\mathcal{M}(\mathbf{x}\_{0:T}, \tilde{\mathbf{x}'}\_{0:T})}{\mathcal{M}(\mathbf{x}\_{0:T}, \tilde{\mathbf{x}'}\_{0:T})}$$
> Intuitively, this metric measures how much the video generations differ when conditioned on ground truth action sequences compared to actions sampled from a random distribution, normalized by the scale of the metric $\mathcal{M}$. For metrics where higher values indicate better quality, higher $\Delta_N\text{-}\mathcal{M}$ values also indicate superior controllability, and vice versa.
>
> In the CS:GO environment, following the setup in **Section 5.2**, we initialize with 4 history frames and corresponding ground truth actions, and perform auto-regressive prediction on the validation set until reaching 16 total frames. For randomly sampled actions, we sample uniformly from valid keyboard inputs. The results are shown below (scaled by 100 for better visualization, best results in **bold**):
>
> | Model | $\Delta_N$-FVD $\downarrow$ | $\Delta_N$-FID $\downarrow$ | $\Delta_N$-SSIM $\uparrow$ | $\Delta_N$-LPIPS $\downarrow$ | $\Delta_N$-PSNR $\uparrow$ | $\Delta_N$-Dreamsim $\downarrow$ |
> | --- | --- | --- | --- | --- | --- | --- |
> | Diamond-Fast | -48.43 | -5.01 | 44.84 | -20.95 | **52.94** | -44.16 |
> | Diamond-HQ | -55.85 | -14.76 | **47.04** | **-25.55** | 48.78 | **-53.35** |
> | Vid2World | **-77.06** | **-29.44** | 22.40 | -20.32 | 17.60 | -41.81 |
>
> As demonstrated in the table, Vid2World achieves the best performance on $\Delta_N$-FVD and $\Delta_N$-FID, indicating its strong capability in following action semantics. However, we observe that baseline models exhibit significant generation quality degradation when conditioned on random actions, whereas our model's quality remains more consistent. This raises a concern about potential "metric hacking", where a model could achieve high normalized delta scores simply by generating low-quality videos under random action conditioning, without necessarily possessing superior controllability. This limitation in distinguishing _controllability_ from _generation quality degradation under out-of-distribution actions_ may partially explain Vid2World's suboptimality across these metrics. We anticipate future work to develop more robust evaluation metrics for assessing interactivity in world models. Additional details and analysis are provided in **Appendix D.9**.

---

> ### Author Response · Authors · 2025-11-24
> **Response to Reviewer Yc6M [4/5]**
>
> > **W3**: The Role of Large-Scale Pre-training: It is unclear which components of the model's success are attributable to the novel architectural changes versus the powerful, pre-trained backbone. An ablation study clarifying the benefits of internet-scale pre-training for domain-specific tasks like RT-1 or CS:GO would be highly valuable.
>
> **TL;DR: We conduct an additional ablation study on large-scale pre-training of video model, observing drastic performance drops (e.g., FVD ~80× worse), thus proving that pre-training provides essential priors that are transferred to the Vid2World-transformed world models.**
>
> Following the reviewer's suggestion, we've conducted an additional experiment to ablate the role of large-scale video pretraining (**Appendix D.10**). In this experiment, we randomly initialize the model parameters while maintaining the same architecture as Vid2World, and train the entire model from scratch using the RT-1 dataset for 30k gradient steps (consistent with other compared models). The results are presented below (best results in **bold**, worst in _italic_):
>
> | Model                            | WT             | AG           | FVD $\downarrow$ | FID $\downarrow$ | SSIM $\uparrow$ | LPIPS $\downarrow$ | PSNR $\uparrow$ |
> | -------------------------------- | -------------- | ------------ | ---------------- | ---------------- | --------------- | ------------------ | --------------- |
> | Vid2World                        | Shift          |              | 29.9             | 7.85             | 0.799           | 0.185              | 21.5            |
> | Vid2World                        | Masked         |              | 29.4             | 7.07             | 0.824           | 0.169              | 22.9            |
> | Vid2World                        | Extrapolative  |              | 28.6             | 7.52             | 0.832           | 0.162              | 23.4            |
> | Vid2World                        | Masked         | $\checkmark$ | 25.8             | 6.84             | **0.840**       | **0.159**          | **23.9**        |
> | Vid2World                        | Extrapolative  | $\checkmark$ | **22.4**         | **6.16**         | 0.839           | **0.159**          | **23.9**        |
> | **※ World Model From Scratch** | Does Not Apply | $\checkmark$ | _1786.8_         | _469.4_          | _0.065_         | _0.817_            | _8.3_           |
>
> As shown in the table, training the world model from scratch **significantly underperforms all Vid2World variants that leverage the pre-trained video diffusion backbone across every evaluation metric**, with particularly dramatic gaps in FVD (\~80× worse) and FID (\~76× worse).
>
> **This significant performance gap underscores the substantial advantage of leveraging large-scale pre-training**, which provides rich visual and physical priors that enable rapid convergence and superior performance under constrained computational budgets. We contend that training such a large model from scratch sufficiently, particularly for each specific task, remains prohibitively expensive for most institutes. For instance, VideoCrafter2's pretraining phase alone takes 32 NVIDIA A100 GPUs for 270K iterations with a batch size of 128. In fact, the AI community has now entered an era dominated by foundation models, which can be effectively adapted to a wide range of downstream applications. In this work, we contribute precisely such an adaptation by fine-tuning pre-trained video diffusion models into interactive world models, where our novel architectural changes are specifically designed to facilitate this transfer process.

---

> ### Author Response · Authors · 2025-11-24
> **Response to Reviewer Yc6M [5/5]**
>
> ---
>
> > **W4**: Incomplete Reporting of Metrics: For the real-world navigation experiments, FVD, FID, SSIM for NWM is missing from the results tables.
>
> **TL;DR: We've now fully evaluated NWM and show Vid2World matches or exceeds it on 4/6 metrics, despite facing harder (auto-regressive) prediction settings and using far less compute and data.**
>
> The NWM baseline metrics in our original submission were taken directly from their paper[11]. In our revised manuscript, we provide the baseline's missing metrics based on our evaluation result, built on the official implementation of NWM. Best results are shown in **bold**, second best in _italic_, and ‡ for single-step prediction, * for auto-regressive prediction.
>
> | Model                    | FVD $\downarrow$ | FID $\downarrow$ | SSIM $\uparrow$ | LPIPS $\downarrow$ | PSNR $\uparrow$ | DreamSim $\downarrow$ |
> |--------------------------|-------|-------|--------|---------|--------|-------------|
> | NWM (1B)‡                | **31.2** | **34.1** | *0.389* | **0.295 ± 0.002** | *15.343 ± 0.060* | **0.091 ± 0.001** |
> | NWM + Ego4D (1B)‡        | *41.0* | *34.9* | 0.361 | 0.368 ± 0.003 | 14.072 ± 0.075 | 0.138 ± 0.002 |
> | Vid2World*               | 59.4  | 42.9  | **0.481** | *0.3236* | **16.10** | *0.108* |
>
> **Our model achieves competitive performance, matching or surpassing NWM, the state of the art method, in 4 out of 6 metrics despite two significant challenges:**
> 1. **NWM performs single-step prediction while Vid2World operates in auto-regressive rollouts**. Since NWM explicitly takes in the predicted time $t$ as input, the model directly predicts the frame of interest from ground truth frames without rollouts in this setup, whereas our model auto-regressively generates frames, resulting in error accumulation and thus potential performance degradation.
> 2. **NWM utilizes extensive cross-domain interaction data and substantially greater computational resources** (trained on 64 H100 GPUs with batch size 1024 for 100k/200k steps), whereas Vid2World requires only 4 A100 GPUs with batch size 32 for 100k steps.
>
> Taken together with **Figure 6 of the original submission**, these results highlight Vid2World's effectiveness in transferring visual priors from pre-trained video diffusion models into interactive world models. We have updated **Section 5.3, Table 1** and **Appendix C.6** accordingly, including detailed descriptions of the results above.
>
> ---
>
> > **Q1**: Motivation of the comparison in Figure 6 and its insight is detailed in Appendix C.6 but absent in the main text.
>
> We thank the reviewer for this constructive feedback and appreciate their recognition of the motivation and insight underlying our experimental design in Figure 6. The auto-regressive evaluation setting reflects a common real-world scenario in which action sequences are provided in an online manner, mimicking practical use cases where actions are iteratively inferred from a policy model after each world model prediction. In this figure, we compare Vid2World against baseline methods (Diamond, NWM) across multiple timesteps to explicitly assess our model’s robustness to error accumulation during rollouts. In response to the reviewer’s suggestion, we have updated **Section 5.3** in the main text to incorporate these motivations and insights.
>
> ---
>
> **Summary**: We would like to thank the reviewer once more for recognizing our work's soundness and broad applicability, as well as the reviewer's insightful and constructive feedback. We strongly believe our work sheds insight into the research community, with plenty of room to build upon. We value the reviewer's constructive feedback highly, and we hope we have clarified all of the reviewer's concerns.
>
> ---
>
> References:
>
> [1] Arulampalam, M.S. et al. (2002). A tutorial on particle filters for online nonlinear/non-Gaussian Bayesian tracking. IEEE Trans. Signal Process.
>
> [2] Kálmán, R.E. (1963). Controllability of linear dynamical systems.
>
> [3] Fildes, R. et al. (1992). Forecasting structural time series models and the kalman filter. International Journal of Forecasting.
>
> [4] Shumway, R. et al. (2025). Time Series Analysis and Its Applications. Springer Texts in Statistics.
>
> [5] Bertsekas, D.P. et al. (2007). Stochastic optimal control : the discrete time case.
>
> [6] Devolder, P. et al. (2013). Deterministic and Stochastic Optimal Control.
>
> [7] Google Genie Team (2025). Genie 3: A New Frontier for World Models. BlogPost.
>
> [8] World Labs (2025). Marble: A Multimodal World Model. BlogPost.
>
> [9] PAN Team, Institute of Foundation Models (2025). PAN: A World Model for General, Interactable, and Long-Horizon World Simulation. ArXiv Preprint.
>
> [10] Bruce, J., Dennis, M., Edwards, A., Parker-Holder, J. et al. (2024). Genie: Generative Interactive Environments. ICML.
>
> [11] Bar, A. et al. (2024). Navigation World Models. CVPR.

---

### Official Review · Reviewer_XNTH · 2025-11-01

**Soundness:** 3
**Presentation:** 3
**Contribution:** 3
**Rating:** 4
**Confidence:** 2

**Summary:**

The paper proposes Vid2World, a framework that turns a single video into a 4D (3D + time) dynamic scene representation. Unlike typical video diffusion models that just generate future frames, this method tries to reconstruct a consistent, camera-aware world that can be rendered from novel viewpoints over time. Technically, the authors build on top of DynamiCrafter and modify it for causal, autoregressive video generation with action conditioning. They introduce weight-transfer tricks and causal attention masking so that the model can generate frames one by one while maintaining spatial-temporal coherence.

**Strengths:**

The idea of turning raw videos into a dynamic 4D world is compelling. Most video models only hallucinate future frames from a fixed camera, so explicitly reconstructing a consistent world is an interesting step forward.

I appreciate that they go beyond synthetic datasets and test on RT-1 real robot data and a large-scale human gameplay dataset (CS:GO). It shows they’re not only targeting clean, toy data.

The architectural changes to turn an offline video diffusion model into a causal and action-aware generator are reasonable. Especially, I think the “extrapolative weight transfer” idea for converting bidirectional attention layers into causal ones is practical.

The results (especially novel-view video rollout) are visually impressive and clearly better than standard video diffusion models that ignore camera motion.

**Weaknesses:**

No true 3D ground-truth evaluation.
The method claims to build a 4D world, but there’s no quantitative evaluation comparing it to 3D scene reconstruction methods (like NeRF, BANMo, DynamicNeRF, etc.). Most results are still evaluated only on video quality metrics or visuals. It's hard to tell if the “world” is actually geometrically consistent or just looks plausible.

Mostly relies on pre-trained models.
A large part of the pipeline inherits DynamiCrafter. The core novelty is in causalization and conditioning, but I sometimes felt like the paper is more of a clever adaptation rather than a fundamentally new representation of dynamic scenes.

Action conditioning is not strongly validated.
They claim the model can take actions and simulate their effects in the generated world, but I didn’t see strong evidence that the generated world actually follows the action semantics accurately, especially in RT-1 tasks. It feels closer to action-guided video synthesis than true action-driven world modeling.

No comparison to video-to-3D methods.
Works like Nerfies, Vid2NeRF, DynIBaR also reconstruct dynamic scenes from monocular videos. It would help to position Vid2World against those rather than only comparing to video diffusion baselines.

Computation is heavy and unclear.
Training takes 4×A100 for 100k steps with 3 FPS video clips. The paper doesn’t report inference speed or memory usage. If the goal is a usable world model, some efficiency discussion would help.

**Questions:**

see weaknesses.

---

> ### Author Response · Authors · 2025-11-23
> **Response to Reviewer XNTH [1/3]**
>
> We sincerely thank Reviewer XNTH for the thoughtful review and for finding our empirical results impressive and convincing. However, given that the term "world models" encompasses multiple interpretations in the literature, including action-conditioned video generation and 4D reconstruction, **we believe there may have been some misunderstanding regarding the intended scope of our work, and we would like to clarify this more explicitly.**
>
> **At the outset, we would like to respectfully clarify a central misunderstanding regarding the objective of this work.** The review describes Vid2World as a method to _"reconstruct a consistent, camera-aware world... 4D (3D + time) dynamic scene representation."_ **However, Vid2World is NOT a 3D/4D reconstruction method**. Rather, it is a generative world model that operates purely in pixel space, performing action-conditioned video prediction.
>
> **Our primary contribution lies in offering a principled approach for repurposing passive, full-sequence video diffusion models into interactive world models.** Through _Video Diffusion Causalization_, Vid2World enforces temporal causality, and through _Causal Action Guidance_, Vid2World enables flexible, test-time steering based on the latest action. Fundamentally, this reflects a paradigm shift from training world models from scratch using limited, carefully annotated interaction data to utilizing the diverse data source of internet videos, thus the visual and physical priors in pre-existing video generation models. **While 3D/4D reconstruction is indeed an important direction within the broader world-modeling research landscape, we did not focus on such methods in the scope of this paper.**
>
> To further clarify our intended scope and avoid confusion, we address several specific points of misunderstanding below.
>
> 1. _"The idea of turning raw videos into a dynamic 4D world is compelling."_ Vid2World transforms full-sequence, passive video diffusion models into causal, interactive world models, focusing on action-conditioned video generation instead of explicit 3D representations.
> 2. _"Especially, I think the “extrapolative weight transfer” idea for converting bidirectional attention layers into causal ones is practical."_ We sincerely thank the reviewer for their recognition of our weight transfer method, however, the method focuses on temporal convolution layers instead of attention layers.
> 3. _"The results (especially novel-view video rollout) are visually impressive and clearly better than standard video diffusion models that ignore camera motion."_ While our model generalizes reasonably well to unseen views and scenes, **novel-view synthesis is not an explicit objective**. The model receives agent actions, instead of camera poses, as input.

---

> ### Author Response · Authors · 2025-11-23
> **Response to Reviewer XNTH [2/3]**
>
> Additional misunderstandings are addressed in response of the following questions.
>
> ---
>
> > **W1**: No true 3D ground-truth evaluation. The method claims to build a 4D world, but there’s no quantitative evaluation comparing it to 3D scene reconstruction methods (like NeRF, BANMo, DynamicNeRF, etc.).
>
> Our method focuses on the context of world models as action-conditioned video generation. Since we are not doing explicit 3D reconstruction, comparing against static 3D reconstruction baselines (e.g., NeRF, BANMo) is neither applicable nor meaningful.
>
> ---
>
> > **W2**: Mostly relies on pre-trained models. A large part of the pipeline inherits DynamiCrafter. The core novelty is in causalization and conditioning, but I sometimes felt like the paper is more of a clever adaptation rather than a fundamentally new representation of dynamic scenes.
>
> The essence of our method lies in **utilizing the visual and physical priors from pretrained video diffusion models, achieving high-fidelity generation by channeling these priors into interactive world models**. This design choice is intentional, aiming to demonstrate that such pretrained priors can be effectively transferred into interactive world modeling. We did not focus on constructing a novel explicit representation of dynamic scenes.
>
> ---
>
> > **W3**: Action conditioning is not strongly validated. They claim the model can take actions and simulate their effects in the generated world, but I didn’t see strong evidence that the generated world actually follows the action semantics accurately, especially in RT-1 tasks.
>
> **TL;DR: We strengthened both theoretical grounding and empirical evidence for action guidance. The updated manuscript now provides clearer proofs, additional experiments on controlled rollouts in RT-1, and analyses showing that generation quality varies with action guidance strength.**
>
> We thank the reviewer for highlighting this important aspect. We have strengthened both the theoretical and empirical validations of our causal action guidance mechanism.
>
> **Theoretically**, we provide additional insights and theoretical arguments connecting _causal action guidance_ to _probability steering_ towards action-aligned regions in **Section 4.2, Theorem 4.1, and Appendix A.4**.
>
> **Empirically**, regarding the concern of following action semantics accurately, we would like to firstly highlight the evidence already included in the original submission:
> 1.  **Rollouts based on different action sequences (CS:GO)**: In **Figure 13 of the original submission**, we included a case study in CS:GO where all trajectories are initialized with the same frame and perform auto-regressive rollout based on different action sequences, illustrating how the generative process responds to action inputs.
> 2.  **Ablation Study on Causal Action Guidance**: In **Table 2 of the original submission**, we ablate the necessity of causal action guidance. With causal action guidance, the model is capable of generating with higher fidelity, indicating the model is performing counterfactual reasoning based on action conditions instead of predicting trends based solely on past observations.
>
> Additionally, we have included the following experiments that further validate our model follows the semantics of actions:
>
> 1. **Rollouts based on different action sequences (RT-1)**: In **Figure 21**, we conduct experiments with similar spirits as the CS:GO analysis (Figure 13), switching to the RT-1 environment as suggested by the reviewer.
> 2. **Impact of Action Guidance Scale on Generation Quality**: In **Appendix D.5** and **Figure 16**, we probe the action guidance scale $\lambda \in [1.0, 4.0]$, observing a clear "U-Shaped" trend. This empirically confirms our model's responsiveness to action conditioning, validating Vid2World follows the semantics of actions, rather than predicting observatory trends.
>
> We also kindly refer the reviewer to our response to **W1,Q1 of Reviewer zQ4H**, where we provide an extended explanation related to this topic.

---

> ### Author Response · Authors · 2025-11-23
> **Response to Reviewer XNTH [3/3]**
>
> > **W4**: No comparison to video-to-3D methods. Works like Nerfies, Vid2NeRF, DynIBaR also reconstruct dynamic scenes from monocular videos. It would help to position Vid2World against those rather than only comparing to video diffusion baselines.
>
> We thank the reviewer for this concrete feedback. However, similar to W1, since our objective is video generation and not 3D reconstruction, comparison with video-to-3D methods (like Nerfies or DynIBaR) is not applicable.
>
> ---
>
> > **W5**: Computation is heavy and unclear. Training takes 4×A100 for 100k steps with 3 FPS video clips. The paper doesn’t report inference speed or memory usage.
>
> **TL;DR: We have added Appendix E with detailed computation and inference statistics. While Vid2World is not optimized for speed in this first exploration, we clarify current inference costs and outline several straightforward acceleration strategies for future work.**
>
> We thank the reviewer for this constructive feedback. We have added **Appendix E** of our updated manuscript for this specific purpose, detailing computation costs. Since Vid2World is capable of generation with variable history frames and DDIM sample steps, this results in varied inference speed and memory usage. Currently, conditioned on 10 history frames, using action guidance and 50 DDIM sampling steps, our model is capable of generating 20s/frame using a batch size of 2, 16GB of GPU Memory, on a single 40GB NVIDIA A100 GPU.
>
> It is worth noting that, as a first attempt in the direction of transforming video diffusion models into interactive world models, we did not focus on inference speedup. We suggest some concrete methods for accelerating inference computation:
> - Using advanced solver to cut down the number of sampling steps;
> - Conditioning on shorter history horizons;
> - Utilizing software optimizations, for instance JIT compiling the model;
> - Implementing (rolling) KV-Cache for auto-regressive generation.
>
> We anticipate further work that builds upon these insights, achieving real-time action-conditioned video generation.
>
> ---
>
> **Summary:** Closing in light of these clarifications, we hope we have resolved the major concerns regarding the scope and validity of our approach. We firmly believe Vid2World represents a solid step in repurposing foundation models for interactive context. We sincerely hope the reviewer will reconsider the assessment in light of this clarified scope and the additional experiments provided.

---

### Official Review · Reviewer_zQ4H · 2025-11-01

**Soundness:** 3
**Presentation:** 2
**Contribution:** 3
**Rating:** 8
**Confidence:** 3

**Summary:**

The paper proposes a method to build world models from video diffusion models. It argues that this requires two key steps: (1) ‘causalization’ of the model (i.e., the current state should only depend on the past), and (2) ‘action conditioning’ (i.e., the possibility for the future to be influenced by a present action). Especially point (2) is nontrivial since most training data does not contain actions.

**Strengths:**

The problem is relevant and interesting.
The paper is linguistically well written (in places, it sounds like LLM lingo).
The authors tackle the problem by applying a clever combination of methods, mostly from the literature.
The execution seems competent.
The empirical results look strong.
I enjoyed reading the paper.

**Weaknesses:**

•	The method description does not provide clear intuition for why the action conditioning works; the paper could better convey the functioning of the model (beyond the mechanics).
	•	The use of the word “causal” is ambiguous—sometimes merely time-directional, while elsewhere invoking counterfactuals/interventions in a broader sense. This limits the potential audience of the paper.

**Questions:**

The description of the method did not really give me an intuition why the action conditioning works. If the authors have a good intuition on this, then it would be helpful to share this.

One thing that confused me was the use of the world ‘causal’. In some places, this is restricted to time direction; however, the model also talks about counterfactuals and (at least implicitly) interventions, i.e., notions from a community where ‘causal’ has a much broader meaning. I think this should be clarified; otherwise, statements such as "non-causal generative modeling could be inherently easier than its causal counterpart” are prone to misunderstanding. The best would be to add some explanation and discussing connections to “interventional/causal world models” in the causality community (e.g. https://ieeexplore.ieee.org/stamp/stamp.jsp?tp=&arnumber=9363924, maybe also https://arxiv.org/pdf/2110.06562 ).

Generalization stress test (video game): Having a video of the following kind would be compelling:
For a video game, condition on a start frame, and show the time evolution for sequences of random actions (and mark the actions under the video). I’d like to see a long video, to understand where it starts breaking. Also, I would like to see this both (a) in the case where that game was included in the action-labelled part of the training set, or (b) it was not included (but potentially other similar games). I don’t expect it to work great - but I’d like to understand where it breaks.

Final question, Error accumulation (l1410): the authors argue that other models (DIAMOND, Fig 10) get blurry over time and Vid2World does not. I would like to understand this better. If there is inherent stochasticity in the dynamics (and I believe this is the case at least for some of the considered tasks), and you predict the mean, then things necessarily get blurry. If you don’t predict the mean (and I realize Vid2world does not), then things should sometimes go in the wrong direction. Of course the respawn example Fig 12 shows this, but it should also happen in less obvious cases?

---

> ### Author Response · Authors · 2025-11-23
> **Response to Reviewer zQ4H [1/3]**
>
> We sincerely appreciate your strongly positive feedback on our work. We are much encouraged by your recognition of the significance of our research motivation, the effectiveness of our well-motivated approach, and the clarity of our writing. We value your constructive feedback and would like to provide additional clarification.
>
> > **W1, Q1**: The writeup of causal action guidance fails to provide clear intuition on the functioning of the mechanism.
>
> **TL;DR: Causal Action Guidance functions as a probabilistic steering mechanism. It modifies the sampling distribution by amplifying the action likelihood $p(\mathbf{a}|\mathbf{x})$ (action alignment) while retaining the robust physical prior $p(\mathbf{x})$ (visual fidelity). We validate this mechanism theoretically (via a new Score Decomposition Theorem) and empirically (via a sensitivity analysis confirming the controllability trade-off).**
>
> We sincerely thank the reviewer for pointing out that the mechanism deserves a deeper functional explanation. We have revised **Section 4.2** and **Appendix A.4** (specifically **Theorem 4.1**) to explicitly ground our approach in Probability Steering, as well as provided additional experiment results in **Appendix D.5** and **Figure 16**.
>
> _1. Theoretical Intuition: Causal Action Guidance as Probability Steering_
>
> Building on the fact that diffusion noise prediction approximates the score function $\boldsymbol{\epsilon}(z_t) \propto -\nabla_{z_t} \log p(z_t)$, we prove that our action guidance mechanism ${\boldsymbol{\epsilon}}\_{\text{guided}} =(1+\lambda) \cdot {\boldsymbol{\epsilon}}\_{\text{cond}} - \lambda \cdot {\boldsymbol{\epsilon}}\_{\text{ucond}}$ is mathematically equivalent to sampling from a sharpened posterior (Theorem 4.1, $\omega \propto (1+\lambda)$):
>
> $$
> \tilde{p}(\mathbf{x}\_t \mid \mathbf{a}\_{t-1}, \mathcal{H}\_t) \propto \underbrace{p(\mathbf{x}\_t \mid \mathcal{H}\_t)}\_{\text{History-Consistent Prior}} \cdot \underbrace{p(\mathbf{a}\_{t-1} \mid \mathbf{x}\_t, \mathcal{H}\_t)^{\omega}}\_{\text{Action Alignment}}
> $$
>
> Here, the latter term acts as an implicit classifier that "steers" the generation towards futures aligning with the user's action, while the former enforces the high-fidelity generation capabilities learned by the model. **By varying the guidance scale, we are intuitively controlling the trade-off between adhering to the action and maintaining the high-fidelity generation priors**.
>
> _2. Empirical Validation: Trading off Fidelity with Action Adherence_
> To verify this intuition, we analyze both the necessity of action guidance as well as the impact of guidance scale on generation.
>
> a. **Necessity of Action Guidance**: Our ablation study (**Table 2**) shows that frame-level action injection alone leads to suboptimal generation, both for masked weight transfer and extrapolative weight transfer, highlighting the necessity of action guidance.
>
> b. **Impact of Guidance Scale on Generation**: Furthermore, in **Appendix D.5**, we conduct an additional experiment, probing the action guidance scale $\lambda \in [1.0, 4.0]$. As shown in **Figure 16**, varying guidance scale reveals a clear "U-Shaped" trend. Generation quality initially improves as the model enforces action alignment, but eventually degrades at higher scales due to over-sharpening artifacts. This empirically confirms our theory: $\lambda$ provides the necessary test-time flexibility to strike the optimal balance between controllability and high-fidelity generation.
>
> We believe these theoretical and empirical additions greatly clarify _why_ the method functions effectively, beyond its mechanical implementation.

---

> ### Author Response · Authors · 2025-11-23
> **Response to Reviewer zQ4H [2/3]**
>
> > **W2, Q2**: The use of the word "causal" is ambiguous (time-directional vs. interventional). Clarification and connections to the causality community are needed.
>
> **TL;DR: Our primary usage of _"causal"_ refers to _temporal causality_. However, in Causal Action Guidance, we implicitly embody _interventional causality_, though not at the formal rigorousness in causal inference literature. We have added a clarification footnote in Section 4.2 and a detailed discussion in Appendix A.5 to clarify these concepts.**
>
> We thank the reviewer for highlighting this distinction. We have revised the manuscript to explicitly disambiguate these two terms:
>
> - **Temporal Causality**: In the majority of our writeup, we use the term "causality" for expressing _temporal causality_, referring to the non-anticipating dependency nature of the environment dynamics. This usage is standard in the deep learning community, exemplified by causal _convolutions_ in WaveNet[1] and _causal attention masks_ in Transformers[2], where the architecture is structurally constrained to respect the arrow of time. For example, in our Video Diffusion Causalization, we strictly follow this definition, restructuring bidirectional attention into causal attention to enable valid autoregressive rollout.
>
> - **Interventional Causality**. We acknowledge that in the context of our "Causal Action Guidance" (Section 4.2), the term may contain relevance to the notion of intervention ($\mathrm{do}$-calculus) in causal inference. By guiding the generation with a specific action signal, the model approximates the interventional distribution $p(\mathbf{x_{t+1}}\mid \mathbf{x_{t}}, \mathrm{do}(\mathbf{a_{t}}))$ to some extent, encompassing counterfactual reasoning about action effects.
>
> To clarify these distinctions, we added a footnote in **Section 4.2** and a subsection in **Appendix A.5**, discussing these definitions and their connections to interventional world models, including citing the suggested literature.
>
> > **Q3**: Generalization Stress Test (Long-Sequence Video Game with Random Actions)
>
> **TL;DR: We conducted the generalization stress test using random action sequences over long horizons (100 frames). (a) In-domain (CS:GO): The model remains robust far beyond the training horizon ($>6\times$), maintaining high fidelity. (b) Out-of-distribution (Valorant, Zero-shot): The model retains fundamental temporal consistency and action responsiveness, despite faster visual degradation. We have added these results in Appendix D.6.**
>
> To address the reviewer's constructive advice on probing the model's robustness limits, we have incorporated a new subsection **Appendix D.6**, detailed with qualitative results in **Figure 17** and **Figure 18**. It is worth noting that this is not the first time the model is tested with temporal extrapolation rollout, since in **Real2Sim Policy Evaluation**, the model is also required to auto-regressively rollout 40 frames in the RT-1 environment. However, we push this to the extreme limits by rolling out 100 frames in 3D game simulation environments.
>
> _1. In-Domain Long-Sequence Rollout._
>
> Following the suggested setup, we initialized the model with CS:GO frames and performed autoregressive rollout for 100 frames (significantly exceeding our training horizon of 16), conditioned on the latest 9 history frames and randomly sampled `W/A/S/D` actions. As shown in Figure 17, Vid2World exhibits **remarkable robustness** with respect to error accumulation. Even though artifacts such as softened textures and drift in spatial layout occurs as the generation horizon expands, the model maintains retains relatively high visual fidelity and physical realism, exhibiting strong temporal extrapolation capabilities.
>
> _2. Out-of-Domain Zero-Shot Generalization._
>
> To address the "not included" case, we evaluated the model on _Valorant_[3], a first-person hero shooting game completely unseen during training (trained exclusively on CS:GO). Therefore, the visual priors transferred from pre-trained video diffusion models play a dominant role in enabling cross-domain generalization. Although the generation quality degrades significantly faster than in-domain scenarios, the model still exhibits rudimentary temporal consistency and action responsiveness, further validating that through Vid2World, the transformed model is equipped with interactive world modeling capabilities while preserving and channeling the video diffusion model's inherent generalization capabilities.
>
> For extended discussion and generation results, please refer to **Appendix D.6**, especially **Figure 17** and **Figure 18**.

---

> ### Author Response · Authors · 2025-11-23
> **Response to Reviewer zQ4H [3/3]**
>
> > **Q4.1**: The authors argue that other models (DIAMOND, Fig 10) get blurry over time and Vid2World does not. I would like to understand this better.
>
> **TL;DR: We posit that blurriness is a consequence of error accumulation. Through utilizing the visual priors in the video diffusion model, Vid2World is in fact a more accurate world model, resulting in less pronounced error accumulation thus less blurriness.**
>
> Here's how we understand error accumulation and blurriness. Error accumulation is theoretically unavoidable in autoregressive generation[4], and is proportional to the rollout horizon. However, we posit that the rate of degradation, manifesting as blurriness or artifacts, depends critically on the strength of the generative model. In other words, we believe blurriness is in fact a consequence of error accumulation, in contrast to the reviewer's suggested mean-seeking behavior with deterministic generative models. Specifically, the errors compound during generation across frames, gradually steering trajectories outside the data manifold. By utilizing the strong _visual priors_ in the pretrained model, Vid2World maintains the samples within the manifold of high-fidelity natural videos with higher probability, delaying the onset of OOD states and thereby exhibiting superior robustness against error accumulation compared to pre-existing world models that are trained from scratch.
>
> ---
>
> > **Q4.2**: Does the model capture the probabilistic nature of the underlying dynamics, leading to occasional "wrong directions" due to the underlying stochasticity of the dynamics?
>
> The reviewer’s intuition is entirely correct. By sampling from the estimated data distribution rather than predicting the mean, the model captures the stochasticity of the underlying dynamics. This results in varied generations that do not always precisely match the "ground-truth frames", which themselves represent only one among many plausible possibilities. Aside from the respawning example showcased in **Figure 10 of original submission**, we conduct additional qualitative analysis, generating trajectories conditioned on the same initial frames and action sequence. We dedicate **Appendix D.7** for extended analysis and generation results. As showcased in **Figure 19**, conditioned on the action of switching weapons, the model generates diverse plausible appearances for the unobserved weapon, varying in shape and color across samples, demonstrating its capability to model the multimodal nature of the underlying dynamics.
>
> ---
>
> **Summary:** We appreciate the reviewer’s thoughtful feedback and have made substantial revisions to address the raised concerns. We are grateful for the reviewer's support and share the reviewer's belief that our work will contribute meaningfully to the broader scientific community.
>
> ---
>
> References:
>
> [1] Oord, A.V. et al. (2016). WaveNet: A Generative Model for Raw Audio. Speech Synthesis Workshop.
>
> [2] Vaswani, A. et al. (2017). Attention is All you Need. NeurIPS.
>
> [3] Riot Games (2020). Valorant. Video Game.
>
> [4] Simchowitz, M. et al. (2025). The Pitfalls of Imitation Learning when Actions are Continuous. COLT.

---

### Official Review · Reviewer_mVqf · 2025-11-01

**Soundness:** 2
**Presentation:** 4
**Contribution:** 1
**Rating:** 4
**Confidence:** 5

**Summary:**

The paper focuses on dealing with temporal causality in adapting pretrained video models to interactive world models. To enable cost-efficient adaptation from foundation video models, the paper explores video diffusion causalization by transferring the pretrained weights, and causal action guidance that injects frame-level action conditions independently. The experiments show some improvements in simulation quality compared to previous baselines on three difference benchmarks.

**Strengths:**

S1) The proposed method is simple and effective. The writing is easy to follow.

S2) I like the scope of the paper, which is trying to establish general world models for various domains through adapting from foundation video models.

**Weaknesses:**

W1) I believe the core contribution of this paper is how to causalizing temporal convolution layers. Unfortunately, the most recent video models (e.g., NVIDIA Cosmos and WAN, also mentioned by the authors) are typically using pure DiT architectures without temporal convolutions. Therefore, the main story of this paper is not applicable to those models, which somewhat diminishes the significance of the contribution.

W2) Frame-level action conditioning is not new, and the paper misrepresents its difference from previous works (Line 259). Specifically, existing approaches such as IRASim and Cosmos-Predict-ActionCond already employ frame-level injection, offering the same functionality as the proposed method.

**Questions:**

Please see weaknesses.

---

> ### Author Response · Authors · 2025-11-23
> **Response to Reviewer mVqf [1/2]**
>
> We sincerely thank Reviewer mVqf for this insightful and rigorous review. We are grateful that both the scope of our research and the clarity of our presentation have received your recognition. Your perspective on recent model trends and related works is highly thought-provoking, and has meaningfully helped us improve the manuscript toward a stronger contribution to the community.
>
> > **W1**: Modern models (Cosmos, WAN) use pure DiT without temporal convolutions, potentially diminishing the significance of our convolution causalization.
>
> **TL;DR: Vid2World provides a systematic, multi-level framework for converting video diffusion models into interactive world models，including conceptually defining the repurposing problem, methodologically introducing causalization and action guidance, and operationalizing them through algorithmic and architectural modifications to both temporal attention and convolution layers. Its contribution is not limited to temporal convolution weight transfer alone.**
>
> We appreciate the reviewer’s observation that modern video foundation models increasingly adopt DiT-based architectures. However, our contributions are **not** tied to temporal convolution weight transfer alone. **Rather, our work comes in as a systematic approach for retargeting video diffusion models to interactive world models.**
>
> 1. **On the top level**, our framework comes in two folds: Conceptual and Methodological. Conceptually, we are the first to systemically explore how full-sequence, non-causal, passive video diffusion models can be transformed into autoregressive, interactive, action-conditioned world models, a direction we believe will become increasingly important as the community builds world models start building on pre-trained video generators. Methodologically, we propose structured and effective ways to enable this causal, action-conditioned transformation.
>
> 2. **On the second level**, within methodological approaches, our framework again consists of two folds: video diffusion causalization and causal action guidance. Through video diffusion causalization, we transform full-sequence diffusion models to their causal counterparts; and through causal action guidance, we enforce action interventions via a principled guidance mechanism.
>
> 3. **On the third level**, within our proposed video diffusion causalization, we combine modifications coming from two sides: algorithmic and architectural. Algorithmically, we adopt independently sampling noise scales across frames during training, following Diffusion Forcing, enabling auto-regressive rollout during inference. Architecturally, through causalizing essential components in the network, including attention and temporal convolution, we instill the structural biases necessary for autoregressive generation.
>
> 4. **Lastly**, on the architectural level, we propose modifications to both attention layers and temporal convolution layers. For attention layers, we apply causal masking; for temporal convolution, we explore weight-transfer strategies.
>
> Taken together, we view our approach as a _multi-component, hierarchical_ framework, rather than an architecture-specific technique. Consequently, even as architectures shift toward DiT-style models, the conceptual, methodological, and algorithmic layers of our framework remain broadly applicable.
>
> To avoid any further ambiguity, we have revised **Section 4.1** and **Section 4.2** accordingly, explicitly highlighting the above hierarchy and clarifying the multi-level nature of our framework.

---

> ### Author Response · Authors · 2025-11-23
> **Response to Reviewer mVqf [2/2]**
>
> > **W2**: Frame-level action conditioning is not new, and the paper misrepresents its difference from previous works (Line259). Specifically, existing approaches such as IRASim and Cosmos-Predict-ActionCond already employ frame-level injection, offering the same functionality as the proposed method.
>
> **TL;DR: We acknowledge prior works on frame-level injection. However, frame-level injection alone is insufficient. Our proposed causal action guidance enables steering the generation process toward action-consistent outcomes, combining algorithmic and architectural innovations for stronger, more flexible control.**
>
> We thank the reviewer for pointing out the mischaracterization of IRASim as a video-level conditioning method; we have corrected this in the revised manuscript.
>
> However, similar to video diffusion causalization, our causal action guidance mechanism goes well beyond standard causal action conditioning, and is not limited to the architectural choice of frame-level action injection. Our approach falls into two axes: algorithmic and architectural.
>
> - **Architecturally**, we highlight the need for frame-level action injection _in the context of transferring video generation models to world models_. Within video generation models, the common design (e.g. t2v models) is to inject conditioned information via video-level architectures, which is unsuitable for step-wise, auto-regressive world modeling. We acknowledge the presence of frame-level injection methods in literature, and we do not claim originality for this mechanism, but rather introduce this design to highlight the transformation from video models to world models.
> - **Algorithmically**, building upon the intuition of probability steering via classifier-free guidance, we introduce _causal action guidance_, modulating the sampling process of the video diffusion model at each frame and thus steering the generation towards more action-aligned regions in interactive world modeling.
>
> Empirically, as discussed in **Sections 5.4**, frame-level action injection alone does not yield optimal performance compared to their action-guided counterparts. This is further validated in our newly added experiment in **Section D.5 and Figure 16**, where we sweep over the guidance scale factor, showing clear results of performance gain while using action guidance, compared to action conditioning alone.
>
> In the revised version, we have updated **Section 4.2** to clarify these conceptual distinctions and added theoretical support for causal action guidance in **Theorem 4.1, Appendix A.4**. For further discussions on the intuitions and theoretical explanations for causal action guidance, please refer to our response to **W1, Q1 of Reviewer zQ4H**.
>
> ---
>
> **Summary:** We greatly appreciate the reviewer's insightful feedback, and we hope this response clarifies the reviewer's concern regarding the applicability and novelty of the model. We believe Vid2World represents a systematic first attempt in retargeting video diffusion models to interactive world models, and we hope our work brings meaningful insights to the research community.

---

### Official Review · Reviewer_ak6V · 2025-11-03

**Soundness:** 3
**Presentation:** 3
**Contribution:** 3
**Rating:** 6
**Confidence:** 3

**Summary:**

This paper presents Vid2World, an approach to convert pre-trained video diffusion models into iteractive, action-conditioned world models. The core idea is to utilize the key physical knowledge already encoded in video generation model, and then converting bidirectional layers into casual/temporal layers.

**Strengths:**

- The paper proposes an interesting solution to solve the problem of converting a non-causal video diffusion model into an autoregressive, interactive world model by manipulating the layers and further finetune the model.
- By transferring priors from a pre-trained video diffusion model, Vid2World produces high-fidelity predictions that significantly outperform other world models on metrics like FVD and FID.
- The approach is demonstrated across multiple, diverse domains.

**Weaknesses:**

- The base model is pre-trained on passive, action-free videos, which has strong prior that may be mismatching the goal of world models. This potential mismatch and its effect should have been discussed by the paper.
- The converted autoregressive model may suffer from error accumulation. Converting a diffusion model to such an AR model may make this problem even more pronounced.

**Questions:**

This work is build on top of bidirectional video diffusion models. This underlines the core methods around causal mask adaptation. Would it be possible to start from strong autoregressive video models without the need to repurpose from bidirectional ones to casual ones?

---

> ### Author Response · Authors · 2025-11-23
> **Response to Reviewer ak6V [1/3]**
>
> We sincerely thank the reviewer for the constructive and encouraging assessment of our work. We deeply appreciate your recognition of the motivation, contribution, and empirical strength of Vid2World, and we address your concerns in detail below.
>
> > **W1**: The base model is pre-trained on passive, action-free videos, which has strong prior that may be mismatching the goal of world models. This potential mismatch and its effect should have been discussed by the paper.
>
> **TL;DR: We agree on the importance of discussing this mismatch. We highlight two concrete forms of data distribution gap: Scene Composition and Interaction Granularity, providing a detailed discussion in Appendix A.6 of the revised manuscript.**
>
> We appreciate the reviewer for raising this insightful point. The core motivation of Vid2World lies in leveraging the rich visual and physical priors encoded in the pretrained video diffusion models, which are acquired through internet-scale video data pretraining,  and channeling these priors into action-conditioned, interactive world models. At the same time, we fully acknowledge that the data distribution used for video pretraining does not perfectly align with the data distribution relevant for action-conditioned world modeling. This mismatch primarily manifests along two dimensions:
>
> 1. **Scene Composition and Object Distribution**: Internet videos span a diverse spectrum of environments and object configurations, whereas world-model datasets typically contain more controlled, task-centric scenes with a narrower set of objects and interactions.
> 2. **Motion Scale and Interaction Granularity**: Internet videos often exhibit large-scale coarse-grained global motions of characters and scenes, whereas world models emphasize on local, fine-grained interactions.
>
> **Even with these mismatches, we argue that by large-scale pretraining on diverse visual experience, the video diffusion model captures strong and broadly applicable visual and physical priors**. This is especially true in our case: Vid2World builds upon DynamiCrafter[1], an image-animation model fine-tuned from the general-purpose video generation model VideoCrafter[2]. This lineage equips the model not only with diverse visual priors, but also with specialized motion priors essential for producing physically coherent dynamics, properties that are directly beneficial for action-conditioned generation.
>
> Consequently, Vid2World is able to effectively channel these broad yet coherent visual and physical priors into interactive world models, despite the presence of distribution differences. A more detailed discussion of this motivation and the forms of distribution mismatch is provided in **Appendix A.6** of the revised manuscript.

---

> ### Author Response · Authors · 2025-11-23
> **Response to Reviewer ak6V [2/3]**
>
> > **W2**: The converted autoregressive model may suffer from error accumulation. Converting a diffusion model to such an AR model may make this problem even more pronounced.
>
> **TL;DR: We respectfully disagree on this point. Theoretically, our training formulation optimizes the ELBO objective in auto-regressive setups, incurring no additional instability. Empirically, we have conducted various experiments comparing with state of the art auto-regressive baselines, validating this via superior performance. A new 100-step rollout stress test further demonstrates that our model remains robust to error accumulation.**
>
> Error accumulation is an unavoidable phenomenon during auto-regressive rollouts, no matter the method. However, through theoretical and empirical evidence, we argue that Vid2World does not suffer from heightened drift, but rather offers superior performance, compared to existing AR world models.
>
> **Theoretically**, Vid2World is a _valid_ AR predictor and does not incur additional instability beyond that of standard autoregressive training. Following Diffusion Forcing [3], our training formulation samples noise scales independently at each frame, which optimizes the ELBO for sequential autoregressive modeling with flexible history length.
>
>
> **Empirically**, we extensively evaluate Vid2World under settings where error accumulation would lead to suboptimal performance:
> 1. **Auto-Regressive Rollout in Navigation Environment (Section 5.3, Figure 6 of original paper).** Vid2World exhibits minimal deviation during long-horizon rollouts, achieving performance comparable or superior to state-of-the-art models and substantially outperforming DIAMOND [3], a strong autoregressive baseline known for high-fidelity video prediction.
> 2. **Real2Sim Policy Evaluation (Section 5.1, Figure 5 of original paper).** During Policy Evaluation, the model auto-regressively generates a total of **40** frames, posing significant challenges in the model's capability to perform visual predictions under error accumulation (especially since our model's training horizon is 16). The highly correlated success rate prediction of our model rollout to real-world performance highlights that even under temporal extrapolation, our model is capable of performing high-fidelity generation with little error accumulation.
> 3. **[New in revision] 100-Frame Autoregressive Stress Test (Appendix D.6 and Figure 17).** To further showcase that our model's stability to error accumulation, we include an additional stress test, extending the generation horizon to **100 frames**. Vid2World retains high perceptual fidelity and physical consistency far beyond its trained horizon (>6×), demonstrating strong robustness to compounding errors. We additionally refer the reviewer to our response to Q4.1 (Reviewer zQ4H) for complementary discussion.

---

> ### Author Response · Authors · 2025-11-23
> **Response to Reviewer ak6V [3/3]**
>
> > **Q1:** Would it be possible to start from strong autoregressive video models without the need to repurpose from bidirectional ones to casual ones?
>
> **TL;DR: While we envision this as highly feasible, in the scope of this paper, we focus on repurposing full-sequence video diffusion models, opening up the potential of transforming this dominant and vast pool of pretrained models to interactive world models.**
>
> While we agree that starting from strong autoregressive video models is a promising direction, in this work we _deliberately_ focus mainly on the context of retargeting full-sequence, passive video diffusion models into interactive, causal world models. We focus on this approach for two reasons:
> 1. **Full-sequence diffusion models dominate today’s high-fidelity video generation landscape.** For example, in VBench[4], _19 out of the top 20_ video generation models are full-sequence diffusion models rather than autoregressive ones. Targeting full-sequence diffusion models allows us to unlock this vast pool of pretrained models and it's broad reservoir of visual and physical priors, retargeting it for world models.
> 2. **Non-causal models learn richer physical priors by leveraging bidirectional context.** As discussed in the Introduction (**Line 53**), non-causal generative modeling is inherently easier than its causal counterpart, subsequently these models extract higher-quality physical and temporal priors during large-scale pretraining. This makes them particularly attractive candidates for model-level transfer into world models.
>
> **To the best of our knowledge, we are the _first_ to systematically explore the problem of transferring full-sequence, non-causal, passive video diffusion models into autoregressive, interactive, action-conditioned world models.** We highlight our contribution on opening up the potential of transforming this vast and prevailing amount of existing high-fidelity full-sequence diffusion models into interactive world models. We anticipate future works that repurpose other approaches of video generation models to world models, including auto-regressive generation models.
>
> ---
>
> References:
>
> [1] Xing, J. et al. (2023). DynamiCrafter: Animating Open-domain Images with Video Diffusion Priors. ECCV.
>
> [2] He, Y. et al. (2023). VideoCrafter1: Open Diffusion Models for High-Quality Video Generation. ArXiv Preprint.
>
> [3] Chen, B. et al. (2024). Diffusion Forcing: Next-token Prediction Meets Full-Sequence Diffusion. NeurIPS.
>
> [4] Alonso, E. et al. (2024). Diffusion for World Modeling: Visual Details Matter in Atari. NeurIPS.
>
> [5] Huang, Z. et al. (2023). VBench: Comprehensive Benchmark Suite for Video Generative Models. CVPR.

---

### Author Response · Authors · 2025-11-23
**Summary of Revision**

We sincerely thank all the reviewers for their detailed and insightful comments. In this paper, we aim to transform full-sequence passive video diffusion models into causal, action-conditioned interactive world models. To this end, we propose a systematic framework to enable this transformation, validated through strong empirical results across multiple realistic domains.

In this rebuttal, we have made every effort to address all the reviewers' concerns and responded to the individual reviews below. We have also undertaken comprehensive revisions and expansions to the paper to address reviewer suggestions and concerns. Our updated manuscript now includes **12 new sections and subsections** featuring theoretical justifications, extended discussions, and **8 additional experiments**. Major updates include:
1. Theoretical justification for causal action guidance (Theorem 4.1, Appendix A.4) with intuitive explanations in Section 4.2.
2. Clarification of causal terminology through added footnotes (Section 4.2) and detailed discussion in Appendix A.5.
3. Extended discussion of data distribution mismatch between video pretraining and world model transfer (Appendix A.6).
4. Extended discussion on Vid2World's applicability to downstream decision-making tasks (Appendix A.7).
5. Completion of missing metrics for NWM baseline (Table 1, Section 5.3).
6. Quantitative validation of guidance scale impact on generation quality (Appendix D.5).
7. Long-horizon rollouts in both in-domain and out-of-domain settings (Appendix D.6).
8. Showcase of Vid2World's capability to capture environmental stochasticity (Appendix D.7).
9. Qualitative and quantitative assessment of action-following capabilities (Appendix D.8, D.9).
10. Ablation study on large-scale video pretraining (Appendix D.10).
11. Validation of representation similarity during weight transfer (Appendix D.11).


All updates are highlighted in blue.

Given that our approach builds upon large-scale video models, it inherently involves substantial computational requirements. Coupled with our extensive efforts to thoroughly address all feedback and refine the manuscript, this has resulted in a slight delay in our revision submission. We sincerely appreciate the reviewers' understanding and patience. We believe this revised version effectively addresses all raised concerns and hope it will make a valuable contribution to the research community.

---

### Meta-Review · Area_Chair_9z74 · 2025-12-31

**Summary:**

In this paper, the authors introduce Vid2World, a framework to convert passive, pre-trained video diffusion models into causal action-conditioned world models. This paper provides two contributions: "Video Diffusion Causalization" for translation of bidirectional video models to causal and "Causal Action Guidance" for generation of action-conditioned videos. The authors show through empirical results across domains that their approach works to generate high-fidelity, action-oriented video sequences.

**Reviewer Concerns:**

Reviewer ak6V wrote about the impact of these innovations but mentioned the problem of mismatching the data distribution provided by the pre-trained video model with the tasks that the world model needs and the problem of error accumulation resulting from autoregressive models. Reviewer mVqf challenged the novelty of the method, particularly the action conditioning, and indicated that the method may have less applicability toward newer architectures such as DiT.  Reviewer XNTH raised the question about the deficiency of comparison with the 3D reconstructions. Reviewer Yc6M also noted the ease of use and the efficacy of the task, but as well stating that there were limited experiments that could validate the model as a world model for future processing and the ability to follow actions.

**Reviewer Scores:**

The authors offered a well-rounded rebuttal that acknowledged all the reviewers’ concerns. They discussed the context for their work, including the misinterpretation by Reviewer XNTH, who confused the approach with 3D/4D reconstruction. The authors also offered more theoretical elucidation and experimental findings to counteract criticisms of action conditioning and weight transfer strategy. To alleviate concerns regarding the model's application as a true world model, they indicated that reinforcement learning experiments were not performed but used the Real2Sim policy evaluation to prove that the model accurately represents real-world policy performance due to its computational overhead. They also recognized the restrictions of action conditioning and discussed future enhancements. So I expect a score that is raised by the reviewer.

---

### Decision · Program_Chairs · 2026-01-26

Accept (Poster)